# A robust and scalable framework for hallucination detection in virtual tissue staining and digital pathology

Luzhe Huang [1,2,3,7], Yuzhu Li[1,2,3,7], Nir Pillar [1,2,3], Tal Keidar Haran [4], William Dean Wallace[5] & Aydogan Ozcan [1,2,3,6] ✉

Histopathological staining of human tissue is essential for disease diagnosis. Recent advances in virtual tissue staining technologies using artificial intelligence alleviate some of the costly and tedious steps involved in traditional histochemical staining processes, permitting multiplexed staining and tissue preservation. However, potential hallucinations and artefacts in these virtually stained tissue images pose concerns, especially for the clinical uses of these approaches. Quality assessment of histology images by experts can be subjective. Here we present an autonomous quality and hallucination assessment method, AQuA, for virtual tissue staining and digital pathology. AQuA autonomously achieves 99.8% accuracy when detecting acceptable and unacceptable virtually stained tissue images without access to histochemically stained ground truth and presents an agreement of 98.5% with the manual assessments made by board-certified pathologists, including identifying realistic-looking images that could mislead diagnosticians. We demonstrate the wide adaptability of AQuA across various virtually and histochemically stained human tissue images. This framework enhances the reliability of virtual tissue staining and provides autonomous quality assurance for image generation and transformation tasks in digital pathology and computational imaging.

Histopathological examination is considered the gold standard diagnostic procedure for most lesions of the human body. Before microscopic evaluation, the extracted tissue samples undergo multistep processing that includes fixation, grossing, embedding, sectioning, staining, mounting and cover-slipping. Artefacts, defined as abnormal structures or features produced by the processing of tissue samples[1], are common in histopathology. These artefacts alter tissue morphology, interfere with the diagnostic process and can even preclude reaching a definite diagnosis. To minimize tissue slides heavily affected by artefacts reaching the pathologist's table, histotechnicians and lab specialists constantly perform manual quality assurance after each processing step. These inspections include comparing the tissue slide gross morphology with the tissue block morphology, histochemical stain (HS) colour quality assessment and checking the presence of tissue folds, holes and other external material masking the tissue.

Over the last decade, pathology has undergone significant technological developments, with the inception of digital and computational pathology. The introduction of deep learning models permits

[1]Electrical and Computer Engineering Department, University of California, Los Angeles, CA, USA. [2]Bioengineering Department, University of California, Los Angeles, CA, USA. [3]California NanoSystems Institute, University of California, Los Angeles, CA, USA. [4]Department of Pathology, Hadassah Hebrew University Medical Center, Jerusalem, Israel. [5]Department of Pathology, Keck School of Medicine, University of Southern California, Los Angeles, CA, USA. [6]Department of Surgery, University of California, Los Angeles, CA, USA. [7]These authors contributed equally: Luzhe Huang, Yuzhu Li. ✉e-mail: ozcan@ucla.edu

the analysis and stratification of histopathology images in ways that were not available before[2,3]. Recently, artificial intelligence (AI)-based techniques have created new opportunities to transform traditional century-old HS methods. Deep learning models have been developed to virtually replicate the images of chemically stained slides using only the microscopic images of unlabelled/label-free tissue samples[4–17], eliminating the time-consuming, laborious and costly chemical staining procedures. These methods, which digitally generate histological stains using trained deep neural networks, are collectively termed as virtual staining (VS) and have been explored for both label-free staining[4–7,9,14–16,18–23] and stain-to-stain transformations[24–30]. Although these VS techniques offer major benefits such as reduced stain variability, high-speed, cost-effectiveness, reduced labour and multiplexing, they also introduce risks of potential hallucinations and artefacts in the generated VS images.

One known pitfall of generative models is their susceptibility to hallucinate by producing information/images that are not based on factual data or reality. These hallucinations can range from subtle structural and/or colour inconsistencies to entirely fabricated content. In the context of virtual histopathology and VS, these hallucinations can be categorized into two main areas: (1) 'technical and unrealistic' hallucinations and (2) 'realistic' hallucinations. Unrealistic hallucinations introduced by technical issues include blurred areas, folded tissue regions or aberrantly stained areas, which are similar to traditional artefacts that are frequently seen in glass slide-based pathology. These forms of hallucinations and artefacts would normally be spotted by an expert histotechnologist or pathologist examining the image. In the second category, however, such hallucinations would appear as 'realistic', such as, for example, the replacement of some tissue components with imaginary ones. These realistic hallucinations might mislead pathologists, deceiving them to diagnose features that do not appear in the actual tissue specimen, although looking realistic and believable from the perspective of tissue staining quality. These types of realistic hallucination may result in a misinterpretation of tissue features, misleading the diagnosis process (for example, hallucinated tumour cells appearing inside a benign tissue section), altering tumour grade (for example, hallucinated mitotic figures within a tumour) and affecting the predicted response to treatment (for example, hallucinated lymphocytes in the tumour microenvironment), among many other possibilities. While several deep learning-based tools have been developed for detecting technical or unrealistic artefacts in standard pathology slides[31–34] that are histochemically stained, their accuracy vary and constant human-in-the-loop inspections are required for quality assurance. Furthermore, none of these tools have been trained or evaluated specifically for their ability to detect hallucinations induced by VS models. Even more critically, there is currently no available tool to evaluate and detect VS tissue images that present realistic and believable hallucinations.

Here, we present AQuA, an automated image quality and hallucination assessment framework as computational tool that is specifically designed for recognizing morphological artefacts and hallucinations created by generative VS models (as depicted in Fig. 1a), including technical/unrealistic and, more importantly, realistic hallucinations. AQuA-Net was trained, validated and tested using a set of VS haematoxylin and eosin (H&E) tissue slides taken from human renal and lung biopsies. Using a data assembly methodology, a wide variety of morphological hallucinations and error modes within VS images were introduced into kidney and lung tissue datasets. Leveraging a novel architecture design (Fig. 1b), AQuA-Net automatically detects these morphological hallucinations and low-quality VS images without the need for any ground-truth information. By introducing a label-free autofluorescence (AF)-based virtual tissue staining model (that is, VS: AF → H&E) and its corresponding reverse image transformation, that is, virtual AF model (VAF: H&E → AF), we employed virtual iterations between the H&E and AF domains, which were used as the input of

AQuA-Net to autonomously detect artefacts and hallucinations in VS H&E images, without access to the ground-truth histochemical H&E images. Blindly tested on human kidney and lung tissue samples from new patients, AQuA successfully classified each VS image as having an acceptable or unacceptable stain quality with high specificity and sensitivity. It also showcased external generalization to new types and styles of hallucinations and artefacts generated by poorly trained VS models that were never used before. In a blinded comparison against a group of board-certified pathologists, AQuA scored superhuman accuracy, especially detecting numerous realistic hallucinations that were scored as good-stained images by expert pathologists, who would normally be misled into diagnosing patients using these hallucinated tissue images. In addition, we adapted AQuA to discriminate good and bad HS, standard H&E images and demonstrated its accuracy to detect staining artefacts that routinely appear in the traditional clinical workflow, also beating various analytical hand-crafted image quality assessment metrics.

Our analyses revealed AQuA's potential as a fully autonomous, widely generalizable virtual tissue staining quality and hallucination assessment framework with high performance. We believe that AQuA can be extended beyond the realm of virtual H&E staining to virtual immunohistochemistry and immunofluorescence staining, which pose similar stain-quality assessment challenges. This autonomous hallucination detection and uncertainty estimation tool for deep neural networks can substantially save time and professional labour conventionally induced by human supervision and simultaneously mitigate potential errors caused by subjective interpretations, which would be valuable for enhancing the reliability of a plethora of deep learning-based generative models and image translation methods.

## Results
### Image-level autonomous quality and hallucination assessment of kidney tissue VS
We first validated our method on virtual H&E staining of human kidney tissue samples. Starting with the establishment of good- and poor-staining models, we trained one VS model using paired AF-HS images (Methods) until it converged. With the convergence thresholds determined on both the validation loss and training epochs, early stopped checkpoints with fewer epochs and higher validation losses were defined as poor VS models, and the other checkpoints with more epochs and lower validation losses were regarded as good VS models. Figure 2a illustrates negative and positive VS images generated by good-staining models (free of artefacts and hallucinations—that is, 'negative') and poor-staining models (artefacts and hallucinations detected—that is, 'positive'), respectively. The registered HS images are also shown here, representing the ground truth—although they are not used in the inference phase since AQuA is an autonomous framework. As pointed by the black arrows, poor VS models produce hallucinations, which are hard to distinguish without the histochemical ground-truth reference. In fact, even with the ground-truth HS images, it is not easy to discriminate hallucinated images from high-quality VS images with sufficient accuracy as presented in Fig. 2b. Since VS models were trained under the generative adversarial network[35] framework, traditional structural quality assessment metrics are incapable of distinguishing semantic hallucination. In terms of commonly used structural image quality metrics, including mean square error (m.s.e.), Pearson correlation coefficient (PCC) and peak signal-to-noise ratio (PSNR), the distributions of the negative and positive VS images present strong overlap and cannot be well separated with simple thresholding. More importantly, in VS workflow, the HS images (the ground truth) would be unavailable, and therefore, such image quality metrics cannot be used in the deployment of the VS workflow.

By contrast, AQuA performs chemistry-free, unsupervised quality and hallucination assessment and better discriminates the two populations of VS images (positive versus negative) without the need for

**a**

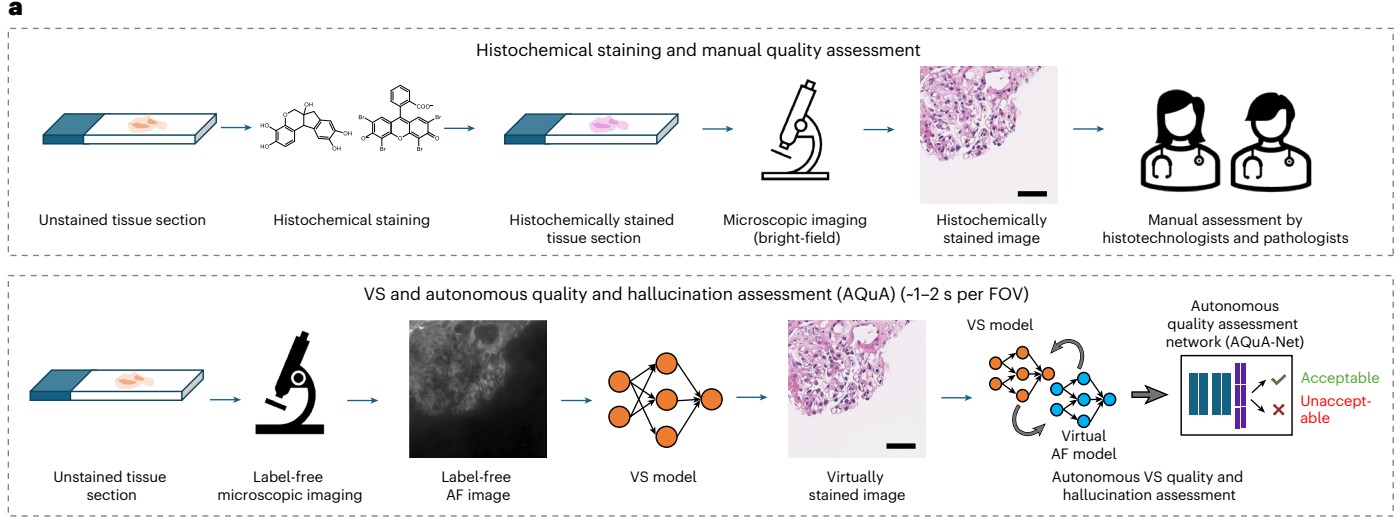

**b**

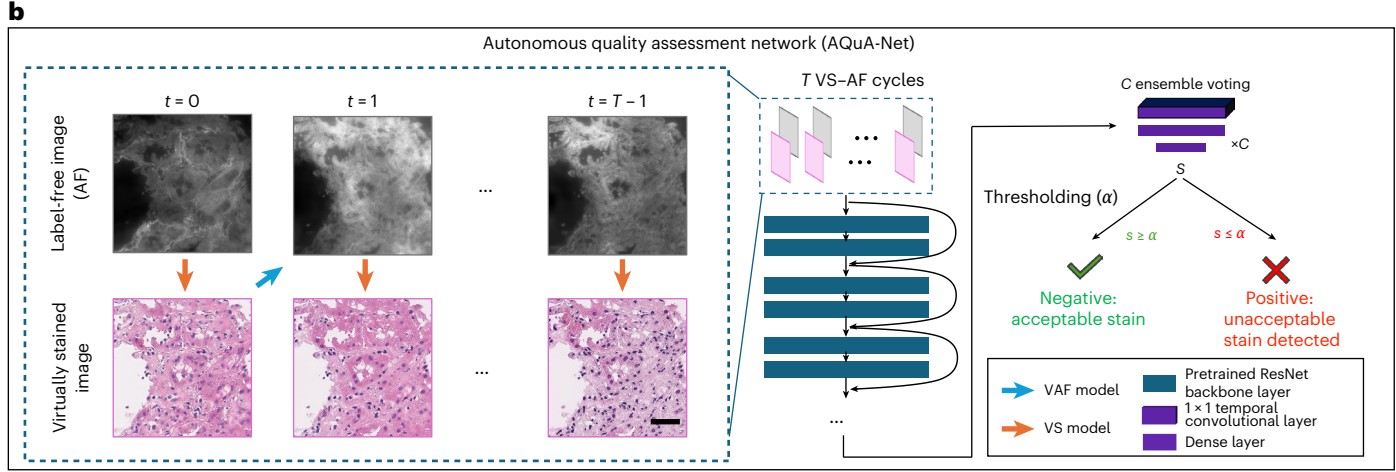

**Fig. 1 | Workflow of HS and VS of tissue and corresponding quality assessment. a**, A traditional HS workflow with the manual quality assessment by histotechnologists and pathologists and VS workflow with autonomous quality and hallucination assessment by AQuA at a throughput of ~1–2 s per image FOV. **b**, The details of AQuA-Net. The image sequences generated by T successive VS–AF cycles are fed into AQuA-Net, passing the pretrained neural network backbone. The C voting heads consisting of a temporal convolutional layer, and the dense layers generate the output confidence score to determine the quality of the VS image after thresholding ($\alpha$). Scale bar, 50 μm.

ground-truth HS images. Figure 2c showcases the confidence score distribution of the two populations and the confusion matrix with respect to the ground-truth labels. Based on a validation set, we selected the threshold $\alpha = 0.5253$ scoring 100% sensitivity and then blindly applied it to the test set (containing ~2,100 VS images) as shown by the green dashed line in Fig. 2c. Both the scatter plot and the confusion matrix confirm a very good classification performance provided by our method on the two populations, with an accuracy of 99.8% and a sensitivity of 99.8%. AQuA's performance at varying threshold values is demonstrated using the receiver operating characteristic curve and the precision–recall curve reported in Supplementary Fig. 1a. The area under curve scores ~1.0 for each curve, indicating the strong classification performance of AQuA. Figure 2d further compares the performances of our method and histochemical ground-truth-based quality assessment metrics, in terms of the absolute value of *t*-statistics and Kullback–Leibler (KL) divergence between the populations of positive and negative VS images. Our unsupervised quality and hallucination monitor provides significantly better positive/negative separation than all the supervised quality assessment metrics that demand access to the ground-truth HS images.

In addition to these comparisons against traditional structural HS-based metrics, we further assessed the performance of our method against the decisions of three board-certified pathologists, covering the condition that no histochemical reference images are accessible. Each pathologist independently and blindly evaluated the quality of a set of VS images of human kidney samples. A consensus was reached on $N = 127$ images, among which $N_G = 66$ images came from good-staining VS models and $N_P = 61$ images from poor-staining VS models. Specifically, the pathologists were asked to score each image on three metrics: the stain quality of the nuclei, cytoplasm and extracellular space from 1 to 4, with higher scores indicating better quality (refer to the evaluation details in the Methods section). Figure 3a reports the overall agreements between our method's confidence scores and the pathologists' scores on all $N = 127$ test images. Here, the range for the average score over three pathologists for an acceptable stain quality was set as (2, 4]. Figure 3a reveals that our method agrees well with the pathologists' opinion (average stain-quality score), corresponding to the blue and green areas in the confusion matrices shown in Fig. 3a. We further break down Fig. 3a into Figs. 3b,c, comparing our method against the pathologist scores on the good-staining and poor-staining VS models, respectively. As illustrated in Fig. 3b, AQuA achieves 100% acceptance rate without any human intervention or supervision; in other words, all VS images from good-staining VS models are accepted. As a reference, the pathologists agreed on 98.5%

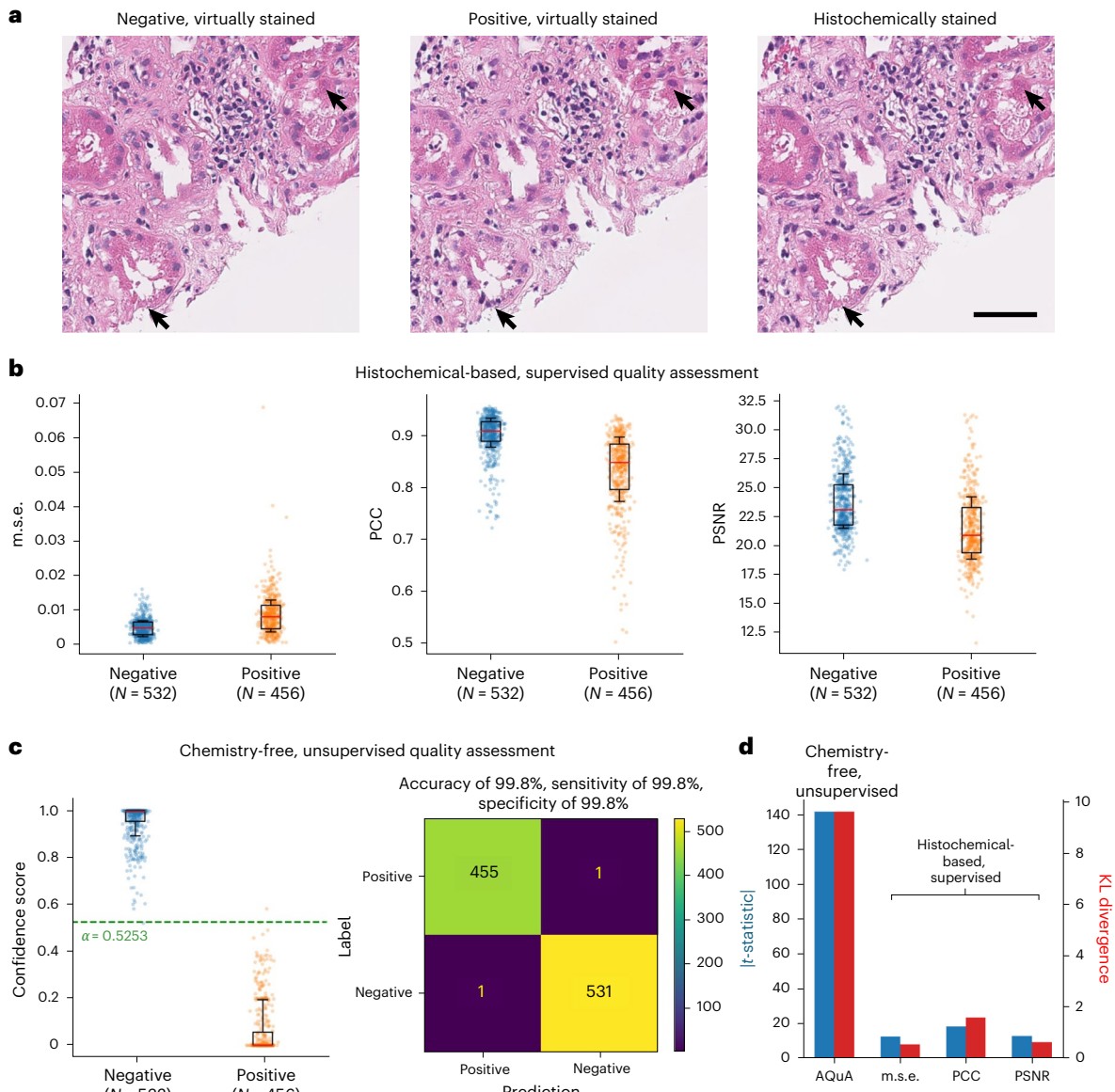

**Fig. 2 | Image-level autonomous quality and hallucination assessment of kidney tissue VS using AQuA. a**, Examples of a negative VS image generated by a good-staining model, a positive VS image generated by a poor-staining model and the corresponding HS H&E image for reference (ground truth). **b**, A performance of histochemical-based, supervised quality assessment metrics on the test set of VS images of human kidney samples. The m.s.e., PCC, PSNR are calculated between the VS images and the corresponding HS ground-truth images. **c**, A chemistry-free, unsupervised quality and hallucination assessment by AQuA on the same test set. The confidence score averaged from $C = 4$ ensemble voters with $T = 5$ VS–AF cycles. The green dashed line illustrates the 100% sensitivity threshold $\alpha$ determined on the validation set. **d**, Despite not using HS ground-truth images in its evaluation, AQuA provides better discrimination between the positive and negative VS image populations compared with histochemical-based, supervised image quality metrics. The box plot shows the 25th–75th percentiles; centre, median; whisker, 16th–84th percentiles. Scale bar, 50 µm.

of the classification of our method (green zones) over the three separate stain-quality scores and the average stain-quality score. In Fig. 3c, when detecting hallucinated VS images coming from poor-staining models, AQuA scores 100% rejection rate without any human supervision or access to HS ground-truth images. By contrast, when presented with these VS images featuring realistic hallucinations (yellow zones), pathologists failed to flag these hallucinated images and could not identify the mismatch between VS images and their HS counterparts when the access to HS ground truth was unavailable. Examples of these realistic hallucinations that mislead the pathologists are shown in Extended Data Fig. 1a, and further analyses of pathologists' failure on these images are detailed in the 'Discussion'. This comparison clearly demonstrated the superhuman level performance of AQuA as an

autonomous VS quality monitor and hallucination detector, rendering its potential in substituting human supervision in a VS workflow where HS is eliminated and not available.

Furthermore, AQuA is effective in detecting various failure patterns of VS models, including suboptimal convergence and generalization failure. In addition to testing our AQuA model on hallucinated images from early stopped checkpoints, we conducted an external generalization test to an overfitted VS model using the same classifier described above, which has never seen such overfitted VS models. As shown in Extended Data Fig. 2a, the overfitted VS model, well trained on a small subset of the training data, fails to generalize on testing data and exhibits a distinct staining failure pattern compared with the early stopped VS models, such as missing a large portion of nuclei and

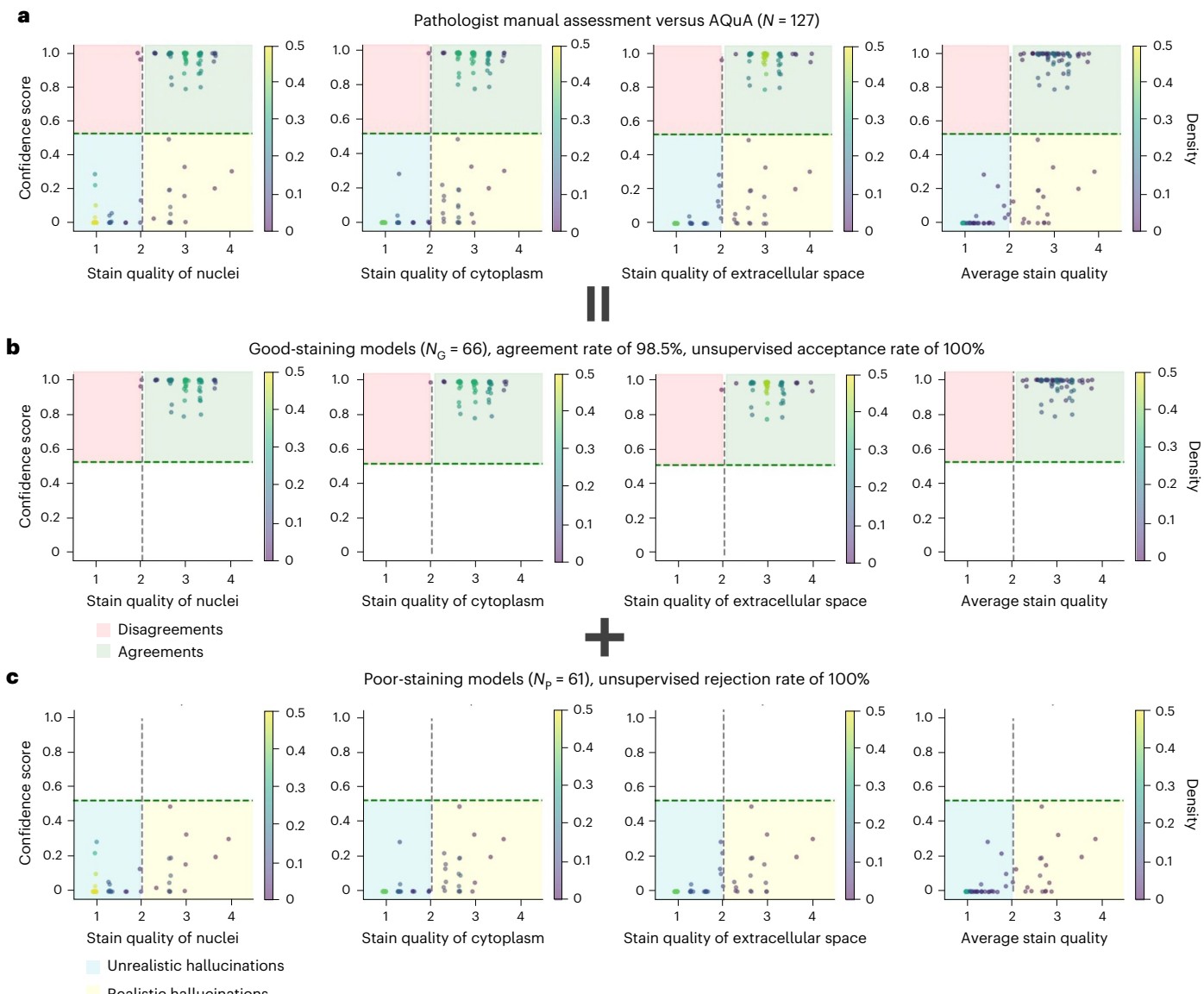

**Fig. 3 | Comparison between AQuA and a group of board-certified pathologists on VS image quality and hallucination assessment. a**, The overall agreement on a test set of $N = 127$ VS images of human kidney samples. Each scatter plot is further broken down into the case of good-staining and poor-staining VS models. **b**, The agreement on a test set of $N_G = 66$ VS images generated by good-staining VS models. AQuA achieves 100% accuracy on negative VS images while the pathologists agree on 98.5% of AQuA's results on average. **c**, The agreement on a test set of $N_P = 61$ VS images generated by poor-staining VS models. AQuA rejects 100% of the positive VS images generated by poor-staining models while the pathologists failed to distinguish realistic hallucinations in these positive images. Some of these realistic hallucination cases are further explored in Extended Data Fig. 1 and in the 'Discussion'.

generating blurry features. Despite the difficulty of detecting these staining failure patterns never seen before during the classifier training, AQuA successfully discriminated all VS images from the overfitted model and the early stopped model as illustrated in Extended Data Fig. 2b. This generalization test further demonstrates the robustness and generalizability of AQuA to various unseen, unknown staining failure patterns of VS models.

### Image-level autonomous quality and hallucination assessment of lung tissue VS

To showcase that AQuA's performance is not organ-dependent, we also applied AQuA to human lung tissue samples. Following a similar training strategy as detailed above, AQuA was trained on human lung VS images and tested on a set of ~2,400 VS images excluded from the training stage (see Methods for details). Figure 4 summarizes the performance of AQuA and compares it against histochemical-based,

supervised quality assessment metrics that used HS ground-truth images. Figure 4a showcases typical negative and positive VS images of human lung tissues, and the corresponding HS image is also shown as a reference. A poor VS model usually generates hallucinatory nuclei as pointed by the black arrows in Fig. 4a, which could be hard to distinguish without the HS image. Even with the supervision from HS ground truths, traditional quality assessment metrics such as m.s.e., PCC and PSNR cannot separate the populations of positive and negative VS images as depicted in Fig. 4b. On the contrary, AQuA successfully discriminates the two groups of VS images as shown in Fig. 4c and achieves a high accuracy of 97.8% and a sensitivity of 99.5% based on a threshold of $\alpha = 0.6263$. For lung tissue samples, AQuA's good classification performance at varying threshold values is also reported in Supplementary Fig. 1b. In terms of quantitative measures between the distributions of negative and positive VS images, our unsupervised method considerably outperforms supervised metrics and separates the two

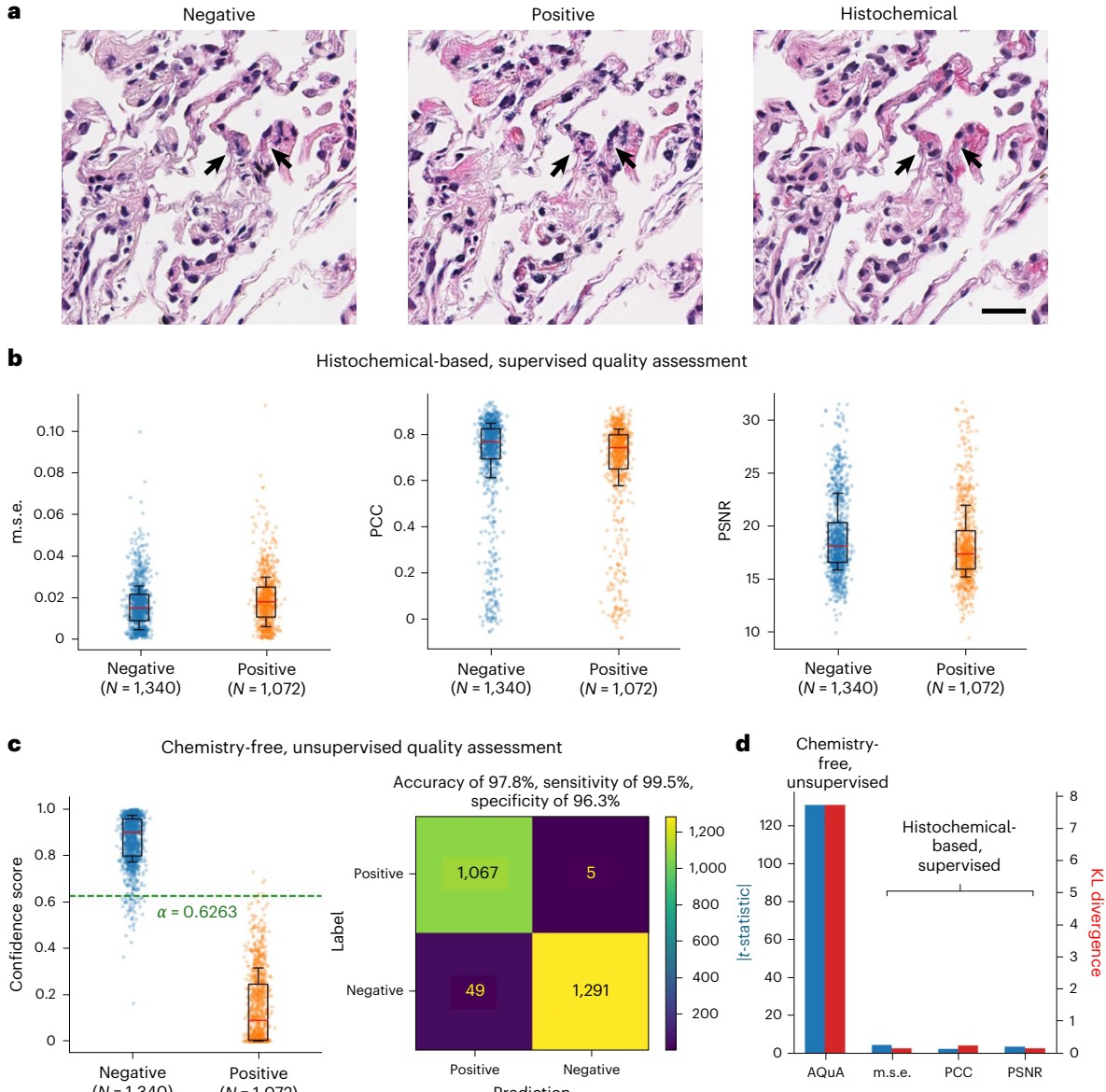

**Fig. 4 | Image-level autonomous quality and hallucination assessment of lung tissue VS using AQuA. a**, Examples of a negative VS image generated by a good-staining model, a positive VS image generated by a poor-staining model and the corresponding HS H&E image for reference (ground truth). **b**, The performance of histochemical-based, supervised quality assessment metrics on the test set of VS images of human lung samples. The m.s.e., PCC, PSNR are calculated between the VS images and the corresponding HS ground-truth images. **c**, A chemistry-free, unsupervised quality assessment by AQuA on the same test set.

The confidence score averaged from $C = 5$ ensemble voters with $T = 5$ VS–AF cycles. The green dashed line illustrates the 100% sensitivity threshold determined on the validation set. **d**, Despite not using HS ground-truth images in its evaluation, AQuA provides better discrimination between the positive and negative VS image populations compared with histochemical-based, supervised image quality metrics. The box plot shows the 25th–75th percentiles; centre, median; whisker, 16th–84th percentiles. Scale bar, 20 μm.

distributions much better, scoring significantly higher $t$-statistics and KL divergence values, even though it does not use ground-truth HS images.

A blinded comparison with pathologists' scores was also implemented on human lung tissue samples, following the previously outlined methodology. On a set of $N = 99$ testing VS images, where pathologists reached consensus, our method shows a good agreement with their scoring as shown in Fig. 5a. Figure 5b,c illustrates the breakdown of Fig. 5a into the two cases on good-staining and poor-staining VS models, respectively. As shown in Fig. 5b, AQuA correctly identifies 100% of the negative VS images generated by good-staining models ($N_G = 59$), on which board-certified pathologists also accepted 98.3% of the same VS images and rejected only one disagreement case

consisting almost entirely of red blood cells (RBCs). The disagreement case is further shown in Extended Data Fig. 1b and analysed in the 'Discussion'. On the poor-staining VS models with $N_P = 40$ images, AQuA successfully rejected all hallucinatory images generated by these poor-staining models, including unrealistic (blue zone) and realistic (yellow zone) hallucinations. By contrast, board-certified pathologists failed to identify the VS images generated by poor-staining models harbouring realistic hallucinations in the yellow zone. An example of these realistic hallucinations is shown in Extended Data Fig. 1a. An additional side-by-side comparison between these realistic hallucinations and the corresponding HS images is illustrated in Extended Data Fig. 3, showing a notable mismatch of nuclei in renal tubules and interstitium in the VS images that are inconspicuous

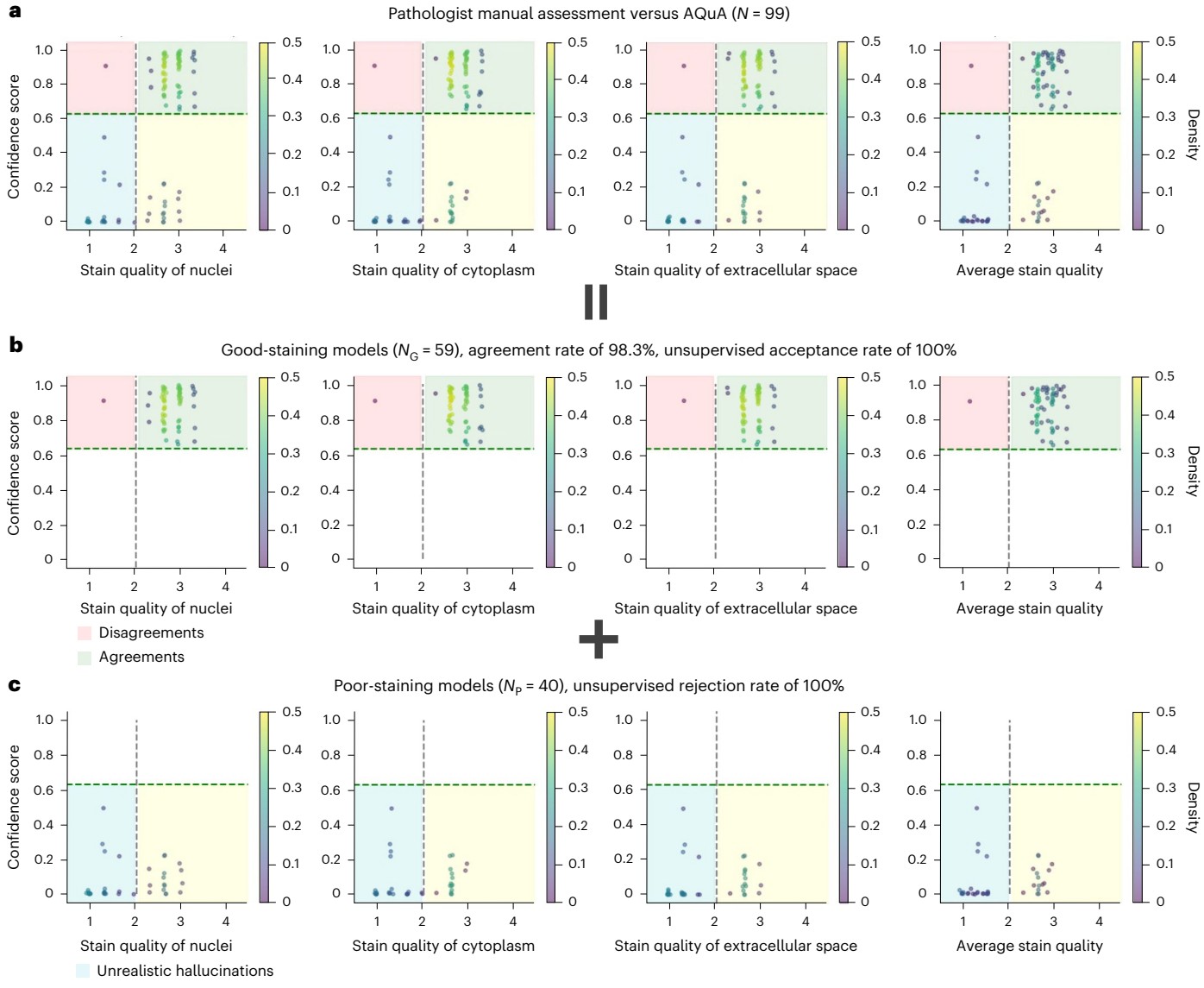

**Fig. 5 | Comparison between AQuA and a group of board-certified pathologists on VS image quality and hallucination assessment. a**, The overall agreement on a test set of $N = 99$ VS images of human lung samples. Each scatter plot is further broken down into the case on good-staining and poor-staining models. **b**, The agreement on a test set of $N_G = 66$ VS images generated by good-staining VS models. AQuA achieves 100% accuracy on negative images while the diagnosticians agree on 98.3% of AQuA's results on average. **c**, The agreement on a test set of $N_P = 40$ VS images generated by poor-staining VS models. AQuA rejects 100% of the positive VS images generated by poor-staining models while the pathologists failed to distinguish realistic hallucinations in these positive images. Some of these disagreement and realistic hallucination cases are further explored in Extended Data Figs. 1 and 3 and in the 'Discussion'.

without HS reference images. These comparisons further confirm the superhuman performance of AQuA's autonomous assessment of staining quality and hallucinations of VS models without access to ground-truth HS images.

To further elaborate on how the poor-quality VS models identified by our AQuA workflow can affect the diagnostic process, we conducted an additional analysis using a case study of a patient with lung transplant rejection. The results are demonstrated in Extended Data Fig. 4, where we compared the outputs from two poor-staining VS models and one good-staining VS model against their HS references. From the HS images (Extended Data Fig. 4d,h,l), a prominent mononuclear cell infiltrate was seen, suggesting acute cellular rejection. However, the first poor-staining VS model, as automatically identified by our AQuA model, generated images with artefacts, such as aberrant colour distributions and blurred nuclear boundaries (see Extended Data

Fig. 4a,e,i), making the images unsuitable for pathological review. Despite appearing to have good staining quality perceptually, the second poor-staining VS model introduced realistic hallucinations by missing many lymphocyte nuclei. This deficiency, shown in Extended Data Fig. 4b,f,j, could mislead pathologists to mistakenly assess the histological picture of this patient as 'non-rejection' rather than 'rejection'. By contrast, the images produced by the good-staining VS model, correctly identified by the AQuA workflow (illustrated in Extended Data Fig. 4c,g,k), aligned closely with the HS ground truths, leading to consistent diagnostic outcomes between the VS and HS images. We want to clarify that although some morphological differences can be observed between good VS images and their HS counterparts, as shown in Extended Data Fig. 4g,h, these differences are comparable to the natural variations seen within HS workflow and should not impact or alter downstream diagnostic results. For instance, the level of change

in nuclei count of VS images compared with their HS counterparts is smaller than the variability observed between two sequential cuts of the same tissue block, which arises from inherent tissue heterogeneity. Additional analysis and supporting information are provided in Supplementary Fig. 2 and Supplementary Note 2. These comparisons underscore the critical importance of AQuA in automatically identifying hallucinations in the VS process, which can potentially mislead the diagnostic process by presenting pathologists with incorrect tissue characteristics.

### Model-level autonomous quality assessment of VS

So far we have focused on implementing image-level autonomous quality monitoring of VS images; this scenario broadly applies to cases where already approved and rigorously validated VS models routinely and continuously generate VS images on new patient specimens, eliminating the standard HS workflow. Therefore, in these cases, AQuA serves as an autonomous and unsupervised 'watchdog' to detect when the VS images should not be used, even if the VS model has already been validated and approved for a given workflow.

Another important scenario that we consider here is the VS model preapproval process and we address the question of should we accept a given VS model into our workflow or not. To answer this question, we created an autonomous method for VS model quality assessment, termed $M$-AQuA—which is also based on the AQuA framework. In practice, the performance of VS models, even if they are preapproved, should be periodically evaluated, as part of a quality assurance routine, for generalization failures, which may result from a variety of causes including for example, changes in imaging and sample preparation protocols, replacement or aberrations of imaging hardware as well as variations in biological content and storage conditions of tissue, among other factors. Hence, the periodic assessment of each VS model is necessary to assure the quality of the whole VS workflow on top of autonomous quality monitoring of each VS image. Of course, the use of $M$-AQuA is supposed to be much less frequent compared with image-level VS quality assessments performed through AQuA, as they serve two different functions; the former ($M$-AQuA) decides on the quality of the VS model, whether, for example, it should be approved for use or trained with more data; the latter (AQuA), however, decides if the generated VS image should be used for expert examination or not.

The workflow of $M$-AQuA is established upon AQuA-based examination on a set of $N$ distinct VS images from the same VS model (that is, under investigation). In this workflow, the previously trained image-level AQuA is independently and simultaneously applied on each VS image in the set, giving the image-level confidence scores. Then, the set of $N$ confidence scores is compared with the confidence score distributions of good- and poor-staining VS models in the training set, to determine if the current VS model under test is generalizing well or not. Based on the theory of linear discriminant analysis (LDA), this is equivalent to comparing the average confidence score $\bar{s}$ (over $N$) to a threshold $\beta$. This threshold $\beta$ is determined on the training set following LDA such that the distributions of good- and poor-staining models have equal probability density at $\beta$ (see Methods for details). Figure 6a depicts the workflow of $M$-AQuA. As shown in Fig. 6a, $N$ VS images generated by the same VS model under test are fed into AQuA in parallel, and the resulting $\bar{s}$ is compared with $\beta$ to determine whether the VS model should be accepted or rejected.

We applied this VS model quality assessment workflow on $M = 10$ test VS models of human lung tissue, among which half were good-staining VS models and the remaining ones were poor-staining models. We randomly sampled $N$ images for each model $R = 100$ times, and summarized the repeatability of our test results (Fig. 6b). The mean and standard deviation of $\bar{s}$ over $R = 100$ repetitions are shown under various $N$ (from 2 to 20). The distribution of $\bar{s}$ squeezes and eventually converges over increasing $N$, indicating that more test images (larger

$N$) improve the discrimination accuracy of $M$-AQuA. Nevertheless, even for $N = 2$, the groups of good- and poor-staining models can be completely separated by a threshold of $\beta = 0.6159$. The accuracy and KL divergence between the $\bar{s}$ distributions of good- and poor-staining models are also reported in Fig. 6, confirming the benefits of increasing $N$. In Fig. 6c,d, we further conducted a scalability test to $M$-AQuA with increasing the number of VS models ($M$). Figure 6c presents the histograms of $\bar{s}$ of $M$ randomly and independently selected VS test models, where the good-staining and poor-staining VS models each form 50% of the models. In all these tests with various $M$, all the good- and poor-staining models were correctly discriminated, scoring 100% accuracy as reported to Fig. 6d. The KL divergence between the $\bar{s}$ distributions of good- and poor-staining models are also reported in Fig. 6d. The precision–recall curve and receiver operating characteristic curves of $M$-AQuA are illustrated in Supplementary Fig. 1c, where the high area under curve values for both curves confirm M-AQuA's successful performance at varying levels of threshold. These findings demonstrate the success and robustness of our model-level VS quality assessment method ($M$-AQuA).

We also tested the external generalizability of $M$-AQuA framework to poor-staining VS models that exhibit staining failure patterns never seen before, including, for instance, some VS models that overfitted to small training data. This might happen if HS staining (the ground truth needed for model training) is scarce, expensive or exhibit variations in quality, with a small number of tissue slides achieving acceptable staining quality, as usually encountered in, for example, immunohistochemistry and immunofluorescence tissue staining. Extended Data Fig. 2c summarizes the external generalization performance of $M$-AQuA on 20 overfitted VS models, 20 early stopped models as well as 20 good-staining VS models—all never seen before. The $M$-AQuA classifier and threshold established using only good-staining and early stopped VS models can successfully generalize to overfitted VS models, scoring 100% accuracy at the model assessment level. This experiment further confirms the robustness of $M$-AQuA to detecting various unseen types of VS model failure, including model suboptimality and generalization errors.

### Autonomous staining quality assessment on HS images

In addition to autonomous staining quality and hallucination assessment on VS images produced by generative AI models, AQuA framework can also be extended to assessing the quality of HS slides of tissue samples, automatically identifying artefacts existing in traditional chemically-stained histology images. Such artefacts can normally arise because of a variety of factors and exhibit different types of imperfection, including loss of image contrast due to erroneous chemical staining operations, reduced image quality because of defocusing and optical aberrations, among other factors. Figure 7a presents typical examples of HS images of good and bad staining quality labelled by board-certified pathologists, where the well-stained images demonstrate normal histological features, with inconspicuous numbers, sizes and distributions of cells, along with intratissue stain variability. By contrast, inadequately stained HS images show silhouettes of cells with lack of basophilia and loss of distinct nuclear contours. Depending on the clinical context, the poorly stained HS images could have been taken from a lung affected by conditions such as pulmonary infarction or organized pneumonia. Alternatively, they could result from poorly fixed, autolysed tissue or out-of-focus regions caused by autofocusing failure of the scanning optical microscope or thickness variations within the sectioned tissue. Extended Data Fig. 5 further illustrates additional examples of poorly stained HS images and highlights several regions with unacceptable staining quality. Figure 7b reveals that the use of well-defined, hand-crafted analytical metrics, such as normalized nuclei count and average nuclei area (see Methods for the definitions of these metrics), often employed in quality assurance in histopathology, cannot effectively differentiate issues

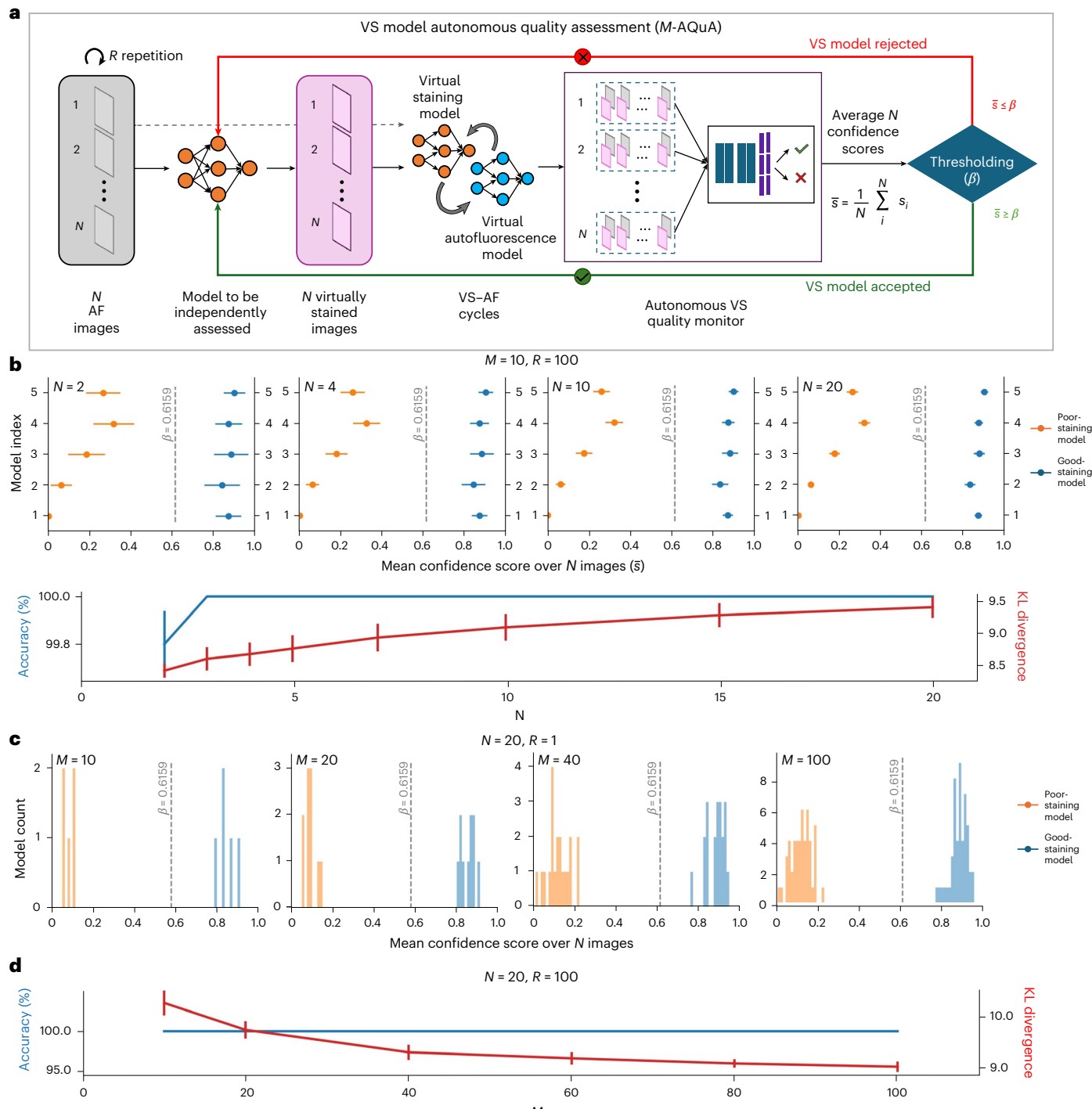

**Fig. 6 | VS model autonomous quality assessment (*M*-AQuA). a**, The average confidence score over a set of *N* test images from the same VS model is compared against the threshold *β* to accept or reject the VS model under investigation. *β* is determined by LDA on the training set of VS images of human lung tissue samples. **b**, *M*-AQuA using varying numbers (*N*) of random testing images on *M* = 10 VS models of human lung tissue. The mean confidence score of each VS model *s̄*, the overall model classification accuracy and the KL divergence between two model groups (poor versus good staining models) are shown. The mean ± standard deviation values are reported on *R* = 100 repetitions. **c**, *M*-AQuA using *N* = 20 random testing images on varying numbers (*M*) of VS models of human lung tissue. The histograms of the mean confidence scores of *M* models are shown for *R* = 1. **d**, The accuracy and KL divergence values are reported as a function of *M*. The mean ± standard deviation values are reported on *R* = 100 repetitions. In these analyses, the ratio of good-staining-to-poor-staining models is 1:1.

observed in poor versus good HS images. However, the application of autonomous stain-quality evaluation using AQuA framework leads to successful discrimination between the well-stained histology images and those laden with artefacts. Here, AQuA was trained on a set of HS images of human lung tissue samples excluded from the testing stage (Methods). This demonstration once again confirms the effectiveness of AQuA framework and its advantages over existing quality assessment methods, clearly highlighting the versatility of AQuA for autonomous assessment and detection of various artefacts, appearing in either VS or HS tissue images.

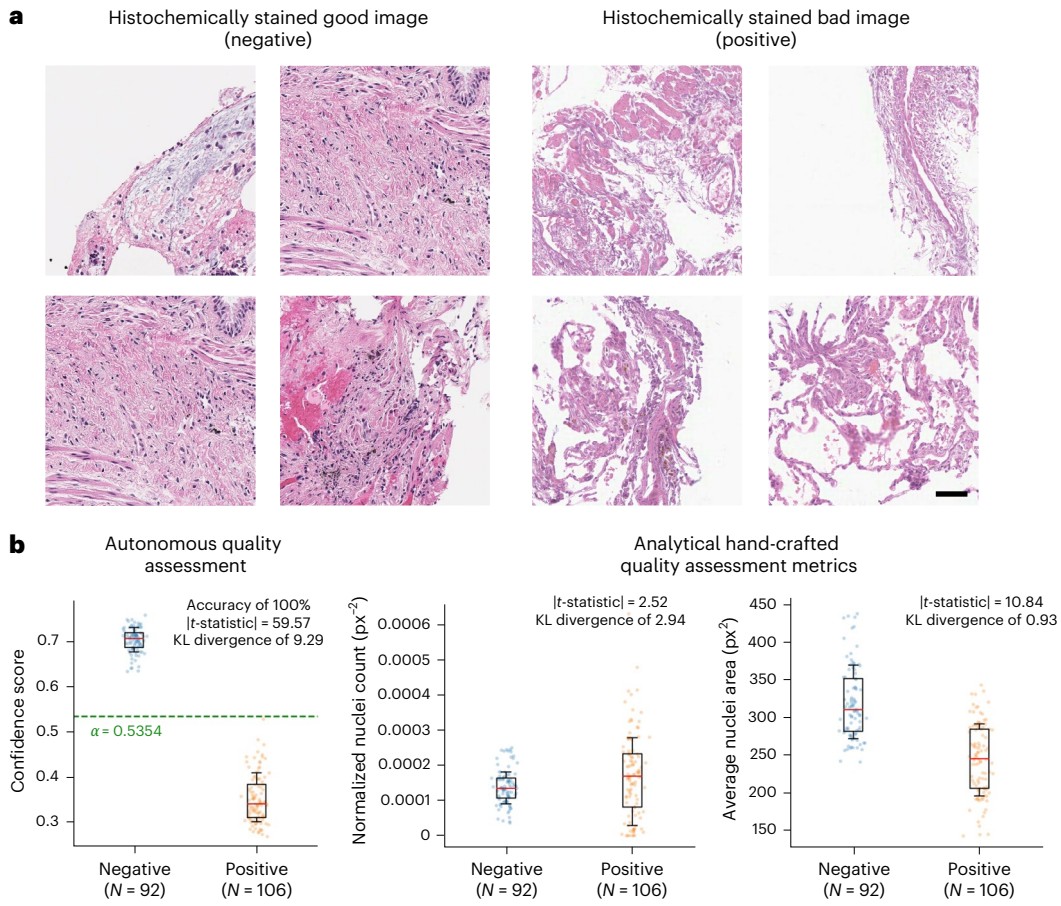

**Fig. 7 | AQuA performance on HS images. a**, Examples of good (negative) and bad (positive) HS images. **b**, Compared with conventional hand-crafted image quality metrics, AQuA provides better discrimination between the positive and negative HS image populations. The box plot show the 25th–75th percentiles; centre, median; whisker, 16th–84th percentiles. Scale bar, 50 μm.

To further demonstrate the adaptability of AQuA to handle an even larger variation of HS image distributions, we extended this pretrained AQuA model to The Cancer Genome Atlas Program (TCGA) image dataset[36]. A total of 461 whole slide images (WSIs) of human lung tissue collected by 49 distinct tissue processing sites within the TCGA dataset were labelled by a board-certified pathologist to identify the presence of artefacts. Using transfer learning, we fine-tuned the pretrained AQuA model using only 10 well-stained WSIs (8 for training and 2 for validation) and blindly tested it on a separate set consisting of 989 HS images (519 good and 470 bad HS images) sampled from the remaining 451 TCGA WSIs, which were excluded from the transfer learning (see Methods for details). Despite encountering various forms of new artefact modes in the TCGA dataset never seen before (Extended Data Fig. 6a), the AQuA model successfully discriminated bad-quality HS images, scoring an overall accuracy of 87.6% on the test set, as shown in Extended Data Fig. 6b,c. These results highlight the external generalization ability of AQuA to various unseen patterns of artefacts and its robustness on new testing data distributions from various tissue processing sites that exhibit several modes of HS artefacts never seen before. It is also worth noting that the VS and VAF models used in the cycle inference were not specifically fine tuned to accommodate the unseen data distribution of the TCGA dataset. Despite this, the AQuA pipeline still achieved a descent accuracy of 87.6%, demonstrating that even when the forward and backward models used in the cycle inference are not ideal, the AQuA pipeline maintains strong resilience to such imperfections.

We also compared AQuA against other HS slide quality control algorithms, including a transformer-based architecture, TransMIL[37], and an open-source quality control tool named HistoQC[38,39] designed for digital pathology slides. When comparing AQuA with TransMIL, both of the models shared the same pretrained ResNet-50 backbone[40] for feature extraction and were trained on the same dataset of human lung HS images employing the same majority voting mechanism ($C = 5$). Extended Data Fig. 7 summarizes the comparison of these two models on the same testing set of human lung HS images and reports the corresponding confusion matrices; the results show that TransMIL partially failed to separate the positive and negative populations with a low accuracy of 68.2%, whereas AQuA provides clear separation between the two populations and scored 100% accuracy. Moreover, the total number of trainable parameters of AQuA (~2.1 million) is less than TransMIL (~3.2 million), showcasing AQuA's efficiency. In addition to these, when comparing AQuA with HistoQC and TransMIL at the WSI-level, AQuA demonstrated a more robust concordance with pathologists' judgements in assessing the quality of the slide staining. Extended Data Fig. 8 illustrates the quality control detection results from the AQuA, HistoQC and TransMIL models for a poorly stained HS slide and a well-stained HS slide, as labelled by pathologists. As depicted in Extended Data Fig. 8a–f, AQuA consistently rejected all the regions from the poorly stained HS slide, while HistoQC and TransMIL only partially rejected some of these poorly stained regions. Although HistoQC and TransMIL successfully identified and rejected severely out-of-focus areas (Extended Data Fig. 8g,h), they both failed to accurately detect regions lacking haematoxylin staining, exhibiting small blurred areas, containing detached tissue fragments and obscured lung architecture due to the presence of excessive RBCs, as shown in Extended Data Fig. 8i–l. As for the well-stained HS slide shown in

Extended Data Fig. 8m–r, all three models—AQuA, HistoQC and Trans-MIL—agreed on accepting most of the tissue regions, as desired. However, HistoQC erroneously rejected some regions with high-quality tissue staining (Extended Data Fig. 8u,v), while AQuA accurately accepted and preserved these stained areas. TransMIL also erroneously rejected several regions with high-quality staining, as depicted in Extended Data Fig. 8w–x, whereas AQuA accurately accepted these regions. Additional examples from another poorly stained HS slide as well as another well-stained HS slide, labelled by pathologists, are illustrated in Supplementary Fig. 3. Here, AQuA was able to accurately reject poor-staining regions aligning with the pathologists' assessments as shown in Supplementary Fig. 3a–f. However, HistoQC incorrectly accepted regions that contained out-of-focus areas, staining artefacts and exogenous pigments, as shown in Supplementary Fig. 3i,j, and TransMIL inaccurately accepted regions with air bubbles in Supplementary Fig. 3k–l. Furthermore, both HistoQC and TransMIL erroneously rejected several regions that exhibited normal tissue structures and high staining quality, as depicted in Supplementary Fig. 3m–r, which AQuA correctly identified and accepted. A similar performance advantage of AQuA over HistoQC was also observed during testing on the TCGA dataset.

## Discussion

We reported AQuA, autonomous quality and hallucination assessment tool for VS tissue images, and showcased its ability in identifying morphological hallucinations formed by generative AI-based VS models. We also highlighted AQuA's ability at the model level by accurately detecting poor-quality VS models with types of staining failure never seen before. These demonstrate its role as a gatekeeper for AI-based VS of tissue in digital pathology.

The rapid rise and wide spread use of deep learning have caused concerns regarding the reliability and quality control of neural network outputs[41,42], especially in critical biomedical applications such as virtual tissue staining[17,43–45]. The generative nature of VS models not only brings up risks from new attack strategies[46–48] but also casts difficulty in detecting the failure modes of these models using traditional and supervised evaluation metrics. In fact, for VS of tissue, in the deployment phase of the VS model there would be no HS available, and therefore, supervised evaluation metrics based on ground-truth images cannot be used in the VS workflow. In VS-related initial studies, labourious manual quality assessments performed by pathologists on the basis of high-level semantic features and domain expertise were critical to assure the quality of VS models; this is not practical for the deployment of a VS model, which is expected to function autonomously. Therefore, AQuA provides a much needed tool for VS quality assessment and hallucination detection without access to HS ground-truth images. Through blind testing on human kidney and lung tissue samples, AQuA achieved 99.8% accuracy and 99.8% sensitivity based on its chemistry-free, unsupervised quality assessment, outperforming common structural, supervised quality assessment metrics that used HS ground-truth images. In comparison with a group of board-certified pathologists, the classification of AQuA reached 100% agreement on negative VS images, generated by good VS models, also manifesting a superhuman performance when detecting realistic-hallucinatory VS images that would normally mislead pathologists to diagnose realistic-looking VS tissue images that never existed in real life.

In addition to providing better detection accuracy, AQuA also presents considerable time savings compared with labourious manual assessment in the traditional HS workflow. H&E staining requires approximately 30–60 min using an automated slide stainer, which can take longer using manual staining protocols—especially for special stains. After the staining, human experts conduct quality control by examining the stained tissue slide under a microscope. A thorough examination, which involves assessing nuclear, cytoplasmic and extracellular morphology under high magnification, can take an additional hour, depending on the training level of the examiner. For virtually stained tissue images, the chemical staining process is eliminated; however, the manual examination of the image by experts still needs to be performed to ensure a high stain quality. As we have shown in this work, realistic hallucinations of VS models can deceive human experts and remain undetected in their manual examination process, potentially misleading them to diagnose features that do not appear in the actual specimen, although exhibiting a good stain quality. By contrast, AQuA achieves a throughput of ~1–2 s per image field-of-view (FOV) and can accurately detect staining quality failures and hallucinations without the need for ground-truth HS images of the samples. Parallelization of AQuA inference can further reduce this automated evaluation time as needed.

The success of AQuA is closely linked to the novel design of AQuA-Net. First, by leveraging successive iterations between the VS and VAF models, inference uncertainty within the VS image gradually accumulates and therefore becomes easier to accurately detect. The advantages of these VS–AF iterations are studied in additional ablation studies performed on the hyperparameter $T$, as summarized in Extended Data Fig. 9. Compared with the AQuA-Net performance without such VS–AF iterations ($T = 1$), which is equivalent to a ResNet baseline, the introduction of these iterations significantly improves both the classification accuracy and the KL divergence between the positive and negative VS image distributions. The effectiveness of VS–AF iterations can be further confirmed by the spatial-temporal Grad-CAM[49,50] activation maps, as illustrated in Extended Data Fig. 10 and Supplementary Figs. 4 and 5 and analysed in Supplementary Note 1. Leveraging the spatial-temporal features in the VS image sequence, the time-averaged activation map of AQuA effectively captures and locates modes of abnormality and inconsistency in the original VS images ($t = 0$), including missing tissue regions, areas with high background noise and residual tissue areas. As detailed in Supplementary Note 1, AQuA's activation maps successfully emphasize relatively large inconsistent regions that are also evidenced by the difference map between the VS and HS images (Extended Data Fig. 10d), as well as some of the smaller abnormal features that are undetectable by traditional pixel-wise difference maps (see Extended Data Fig. 10e,f). This comparison between AQuA's activation map and the HS reference image-based difference map further reveals the difficulty of VS image quality and hallucination assessment task even with the use of an HS reference image; this highlights the merits of AQuA as an autonomous hallucination and artefact assessment tool without the need for ground-truth HS images. Second, the use of majority voting mechanism considerably enhances the classification performance in terms of accuracy and sensitivity. As reported in Extended Data Table 1, compared with a single classifier ($C = 1$), majority voting mechanism with $C > 1$ effectively boosts both the accuracy and the sensitivity. The optimal hyperparameters $T$ and $C$ were determined through a grid scanning-based selection process and highlighted in Extended Data Table 1 for our experiments using human kidney and lung tissue samples. Furthermore, compared with drastically increasing the scale of the network model using complicated architectures and more trainable parameters, the majority voting mechanism provides an efficient and effective strategy to improve inference performance, as highlighted in Supplementary Fig. 6 and Extended Data Table 2.

Furthermore, AQuA exhibits robust external generalization to various unseen distributions during the blind testing stage. Extended Data Fig. 2 demonstrates AQuA's strong external generalization to unseen VS model failure patterns, where overfitted VS models are successfully detected by AQuA, which was trained only on failure modes of early stopped models. Moreover, AQuA's robustness can be further extended to generalization on unseen tissue types. We blindly tested two AQuA models trained with human kidney (and lung) samples on VS images of human lung (and kidney) samples, respectively. Despite the distinct tissue structures of these two organs and different imaging

parameters, including varying signal-to-noise ratio, AQuA successfully generalized and detected poor VS images with a sensitivity of 98.9%, as reported in Supplementary Fig. 7a,b. This strong generalization of AQuA effectively prevents low-quality VS images from a shifted data distribution passing through a pretrained AQuA model. In addition to these, we also used transfer learning on each new tissue type to show that AQuA could achieve further improved inference performance, comparable with an AQuA model trained from scratch solely on the testing tissue type (Supplementary Fig. 7c,d). This robustness of AQuA to new tissue types exemplifies its strong generalization, helping to avoid false negatives due to a shifted data distribution. As another example, Extended Data Fig. 6 demonstrates AQuA's external generalization on various unseen types of artefact that appeared in HS images with varying tissue sources, different sample preparation protocols and different imaging hardware from tens of different sites.

H&E staining has remained unchanged for over a century due to its reproducibility with a variety of fixatives, displaying a broad range of cytoplasmic, nuclear and extracellular matrix features. In a typical tissue sample, nuclei are stained blue/purple (with haematoxylin), while the cytoplasm and extracellular matrix exhibit varying degrees of pink staining (with eosin). Notably, cells display significant intranuclear details, featuring cell-type-specific patterns of nuclear condensation that hold diagnostic significance. The ability of virtual tissue staining technologies to replicate these intricate details using generative AI models harbours a concerning aspect of inducing hallucinations and generating visual artefacts. Many of these artefacts would classify as unrealistic hallucinations, resembling traditional histology artefacts that are easy to spot by experts—for example, morphological changes that may arise in any tissue processing step, such as unstained/over-stained areas, tissue folding, tears and holes. These unrealistic and easy-to-spot artefacts present a sharp contrast from well-stained areas. On the other hand, realistic hallucinations are much harder to detect due to their tendency to blend in the stained tissue. Hence, realistic hallucinations could lead to misinterpreting tissue characteristics, potentially misleading the diagnostic process by evaluating tissue images that never existed. To better highlight this, Extended Data Fig. 1a provides examples of realistic hallucinations that were predicted as having acceptable quality by a group of pathologists, which were correctly flagged by AQuA. Two pairs of images involved kidney tissue samples, wherein several nuclei in the glomeruli and renal tubules are missing on the VS images compared with their histochemical counterparts. Human practitioners/pathologists determined that these images were properly stained (ready to be diagnosed), while AQuA labelled these images as poor staining quality (that is, cannot be used for diagnostics). Another example originates from a lung tissue sample, where a number of type 2 alveolar cells and macrophages are under-stained in the VS images compared with traditional histology. Similar to the kidney case, these images were approved by pathologists but rejected by AQuA without access to HS ground-truth images. The only instance of a false negative evaluation (compared with pathologist consensus) that we observed is reported in Extended Data Fig. 1b; it is taken from another lung sample, where the entire patch is composed of RBCs and scattered lymphocytes. The under-representation of RBCs in the training data of normal lung tissue slides led to AQuA accepting this image FOV, whereas human pathologists, who are familiar with similar morphologies in lung specimens, disapproved its quality. Considering iatrogenic changes, including displaced epithelium and procedure-related bleeding, human diagnosticians typically disregard these areas when evaluating a tissue slide.

Besides detecting hallucinations and generalization failures of a trained VS neural network, AQuA can be potentially applied to provide another level of protection to the VS pipeline against adversarial attacks. Common adversarial attack techniques utilize the gradient of a trained neural network or its approximation to create imperceptible perturbations that mislead the network[51–53]. With the iterative inference of two separate neural networks (VS and VAF) between the two image domains, AQuA is intrinsically robust to gradient-based attacks and potentially applicable to detecting noise-corrupted inputs[54]. Moreover, AQuA framework can be prospectively beneficial when serving as an additional supervision metric to enhance the robustness of generative models[55–58]. In addition, although we did not provide results on this capability in this manuscript, the AQuA framework can potentially be extended to some of the recent advances in stain-to-stain transformations and stain normalization methods[25,59–61]. Through iterations between image domains of different stains or between image domains of AF and multiple types of stains, AQuA might provide autonomous quality and hallucination assessment for virtually transformed and normalized pathology images, which is especially important for cases where HS references are inaccessible.

As an autonomous tool, AQuA provides a robust, scalable and efficient quality assurance framework as a hallucination detector for VS images and provides a transformative advancement towards more reliable, trustworthy AI in VS-related pathology applications.

## Methods

### Sample preparation and standard histochemical H&E staining
The unlabelled kidney and lung biopsy tissue blocks utilized in this study were formalin-fixed, paraffin-embedded samples obtained from previously collected, deidentified specimens at the University of California Los Angeles (UCLA) Translational Pathology Core Laboratory, adhering to the ethical guidelines set by the UCLA Institutional Review Board (with approval from UCLA IRB 18-001029). The kidney samples originated from 10 unique patients who experienced a myriad of non-neoplastic kidney diseases (7 used for training and validating the VS/VAF and AQuA models and 3 for testing), while the lung samples came from 29 unique lung transplant recipients who underwent biopsy for transplant rejection evaluation. Among these 29 unique patients, 18 cases were allocated for training and validating the VS/VAF models, and 7 out of these 18 cases were chosen for training and validating the AQuA model. The rest of the 11 cases were left for testing the AQuA model (1 case for evaluating VS slides and 10 cases for assessing HS slides). Note that each unique patient corresponds to a unique WSI in our study. These tissue blocks were sectioned into ~4-µm-thin slices, deparaffinized and mounted onto standard glass slides for AF imaging. Following the AF image capture, these unlabelled tissue section slides were sent for standard histochemical H&E staining. This process was carried out by the Tissue Technology Shared Resource at UC San Diego Moores Cancer Center for kidney samples and by the UCLA Translational Pathology Core Laboratory for lung samples.

### Image acquisition
The AF images of unlabelled kidney tissue slides were captured by an Olympus IX-83 microscope (controlled by MetaMorph microscope automation software) with a 40×/0.95 NA (UPLSAPO, Olympus) objective lens, under the 4′,6-diamidino-2-phenylindole (DAPI) filter cube (Semrock OSFI3-DAPI5060C; excitation 377/50 nm, emission 447/60 nm). Similarly, for unlabelled lung tissue slides, the AF images were obtained using a Leica DMI8 microscope (controlled by Leica LAS X microscopy automation software) with a 40×/0.95 NA objective lens (Leica HC PL APO 40×/0.95 DRY), under the DAPI filter cube. After undergoing standard histochemical H&E staining, the stained tissue slides were digitized using a Leica Biosystems Aperio AT2 brightfield slide scanner.

### Image coregistration between paired AF and H&E images
To build label-free VS networks and their reversed VAF networks in a supervised approach, it is necessary to obtain well coaligned image pairs of AF images and their corresponding H&E stained images for the same FOVs. This alignment process entails a multistep process, beginning with broad matching and progressively refining to achieve

pixel-level precision. First, a rigid registration of the WSIs was conducted by calculating the maximum cross-correlation coefficient between the AF and H&E stained WSIs, correcting the relative rotation and translation between the two WSIs. Following this coarse alignment, these WSI pairs were segmented into smaller, localized FOV pairs (1,424 × 1,424 pixels for kidney FOVs and 1,024 × 1,024 pixels for lung FOVs) for multimodal affine registration[62] to correct any discrepancies in shift, scale and rotation between the AF and H&E image patches. Subsequently, to attain pixel-level precision in alignment, an elastic image registration algorithm based on local correlation matching[63,64] was employed. To facilitate this precise alignment, an initial, approximate VS network was trained and applied to the AF images. The resulting initially VS H&E images and their corresponding brightfield HS images were then used to calculate the transformation maps through a correlation-based, elastic pyramidal registration algorithm[63,64]. These transformation maps were then applied to correct for the local wrappings of the HS H&E images, resulting in a more accurate match with their AF counterparts. This training-registration cycle was repeated three to five times until the AF images and the corresponding brightfield H&E stained image patches were accurately matched at the single pixel-level.

### Dataset preparation for training VS and VAF networks

Following the image coregistration process, we populated 1,054 non-overlapping AF-H&E image patch pairs (1,424 × 1,424 pixels) for kidney samples and 1,068 pairs (1,024 × 1,024 pixels) for lung samples to train and validate the VS and VAF networks. The ratio for separating the training and validation datasets was set to 9:1. During each epoch of the training phase, these paired image FOVs were further subdivided into smaller 256 × 256 pixel patches, which were then normalized to a distribution with zero mean and unit variance. Before fed into the network, these normalized patches were further augmented by random flipping and rotation. Furthermore, to obtain the overfitted VS models for kidney samples, we selected a significantly smaller subset of the original training dataset, comprising just 66 AF-H&E image pairs, each with dimensions of 1,424 × 1,424 pixels.

### TCGA dataset

A total of 677 human lung tissue WSIs from 57 tissue source sites collected from the TCGA dataset[36] were labelled by a board-certified pathologist, and 395 good and 66 bad WSIs (that is, 461 WSIs in total) from 49 distinct source sites were determined suitable for this experiment, while the other WSIs contained artefacts not overlapping with large tissue areas and were therefore labelled 'ambiguous' and not used. Good WSIs had consistently good quality and no apparent artefacts. Bad WSIs contained artefacts overlapping with the tissue areas resulting from for example, defects of prepared slides, deblurring and artificial markers. Each WSI was randomly cropped into multiple 2,048 × 2,048 pixel patches, and each patch was then checked by a pathologist to have a consistent label (good/bad) with its source WSI. Since the WSIs in the TCGA database originated from 49 distinct tissue sources, they have significant H&E colour style variations that the initial AQuA model has never seen. Consequently, transfer learning is essential to maintain the high performance of the AQuA workflow. To highlight the importance of transfer learning, the testing results of the AQuA model with and without transfer learning are provided in Supplementary Fig. 8. During the transfer learning, two small subsets including 36 patches from eight good WSIs and 8 patches from two good WSIs were used as the negative samples for training and validation, respectively—that is, no bad WSIs were used in the transfer learning step. The rest of the 989 patches (519 good and 490 bad patches) generated from 385 good WSIs and 66 bad WSIs were used for blind testing. Since no bad WSIs from the TCGA dataset were used during transfer learning, we were able to test the external generalization ability of AQuA to various unseen patterns of HS artefacts from multiple tissue sites, as reported in Extended Data Fig. 6.

### Network architecture and training schedule of VS networks

To implement the VS neural networks for kidney and lung tissue samples, we used a generative adversarial network[35] architecture, detailed in Supplementary Fig. 9a,b. This architecture comprises two deep neural networks: a VS generator network (Supplementary Fig. 9a) and a discriminator network (Supplementary Fig. 9b). The generator network followed a five-level U-net[65] structure, which learned the statistical transformation from the input AF image to its corresponding brightfield H&E stained image. The discriminator network, based on a standard convolutional neural network classifier architecture, aimed to distinguish between VS H&E images produced by the generator and the actual HS H&E images. Through a competitive training process, the generator and discriminator were alternatively optimized, leading to improvements in their performance.

During the training phase, the VS generator network iteratively minimizes the loss function defined as

$$L_{G_{VS}} = \alpha_{VS} L_1 \{I_{VS}, I_{HS}\} - \beta_{VS} \log_{10} \left( \frac{1 + SSIM\{I_{VS}, I_{HS}\}}{2} \right) + \gamma_{VS} TV \{I_{VS}\} + \delta_{VS} BCE \{D(I_{VS}), 1\}, \tag{1}$$

where $I_{VS}$ denotes the VS H&E images outputted by the VS generator network, $I_{HS}$ denotes the HS H&E images (ground truth), $D(\cdot)$ represents the probability of being a HS H&E image predicted by the discriminator network. The coefficients $\alpha_{VS}$, $\beta_{VS}$, $\gamma_{VS}$ and $\delta_{VS}$ were empirically set as 1, 0.05, 0.01 and 0.01, respectively, for both the kidney VS network and the lung VS network. The $L_1\{\cdot\}$ stands for the mean absolute error loss, defined as

$$L_1 \{I_{VS}, I_{HS}\} = \frac{1}{X \cdot Y} \sum_{m,n} |I_{VS}(m, n) - I_{HS}(m, n)|, \tag{2}$$

where $X$ and $Y$ represents the pixel numbers at horizontal and vertical directions of $I_{VS}$ and $I_{HS}$, and $m$ and $n$ are pixel indices.

The SSIM$\{\cdot\}$ stands for the structural similarity index (SSIM)[66] between two images, which is defined as

$$SSIM(A, B) = \frac{(2\mu_A \mu_B + c_1)(2\sigma_{AB} + c_2)}{(\mu_A^2 + \mu_B^2 + c_1)(\sigma_A^2 + \sigma_B^2 + c_2)}, \tag{3}$$

where $A$ and $B$ represent $I_{VS}$ and $I_{HS}$ in our implementation, $\mu_A$ and $\mu_B$ correspond to the mean intensity values of images $A$ and $B$, respectively, and $\sigma_A$ and $\sigma_B$ denote the standard deviations of pixel intensity values within images $A$ and $B$, respectively. $\sigma_{AB}$ represents the intensity covariance between images $A$ and $B$. $c_1$ and $c_2$ were empirically selected as 0.01 and 0.03, respectively.

The TV$\{\cdot\}$ refers to the total variation (TV) loss, defined as

$$TV \{A\} = \sum_{m,n} (|A(m + 1, n) - A(m, n)| + |A(m, n + 1) - A(m, n)|), \tag{4}$$

where $A$ represents $I_{VS}$ in our implementation.

The BCE$\{\cdot\}$ stands for the binary cross entropy (BCE) loss, defined as

$$BCE \{p, q\} = -[q \cdot \log_{10}(p) + (1 - q) \cdot \log_{10}(1 - p)], \tag{5}$$

where $p$ represents the discriminator predictions and $q$ represents the actual labels (0 or 1).

The loss function for optimizing the VS discriminator network is defined as

$$L_{D_{VS}} = \frac{BCE \{D(I_{VS}), 0\} + BCE \{D(I_{HS}), 1\}}{2}. \tag{6}$$

The VS generator network and the discriminator network were updated with a frequency ratio of 3:1. The learning rate for optimizing VS generator network and discriminator network were set as $10^{-5}$. The Adam optimizer was used for the network training. The batch size was set as 16. A progressive strategy[67] was utilized in training process to improve VS performance. The training process for kidney samples lasted ~20 h, resulting in models that spanned 750 epochs. For lung samples, the training took ~90 h and yielded models encompassing 1,263 epochs. These models were then categorized into good staining and poor-staining VS models for the following training of AQuA model. The optimal VS generator network models for kidney and lung samples, utilized to facilitate the cycle of AF-H&E transformations within the AQuA model, were chosen on the basis of the lowest validation loss values and evaluations conducted through human visual inspection. More specifically, the VS images from the validation set generated by the models exhibiting the minimal validation loss were manually inspected and compared with their HS counterparts under the supervision of pathologists. The optimal VS model was then selected on the basis of its best perceptual similarity to the HS ground truth in terms of nuclear, cytoplasmic and extracellular details and the accurate representation of relevant diagnostic characteristics. The staining quality of the optimal VS models was assessed and validated against HS reference images in Supplementary Figs. 10 and 11 and Supplementary Note 3, where multiple staining quality evaluation metrics indicated no statistically significant difference between the VS and the corresponding HS images.

### Network architecture and training schedule of VAF networks

The network architecture of VAF networks (Supplementary Fig. 9c) for kidney and lung samples followed the same five-level U-net[65] structure as the VS generator network, except for the reversed input and target images. Mirroring the VS generator model, the VAF network takes the H&E stained image as input, converting it into its corresponding AF equivalent image. The loss function to optimize the VAF network is defined as

$$L_{G_{VAF}} = \alpha_{VAF} L_1\{I_{VAF}, I_{AAF}\} - \beta_{VAF}\log_{10}\left(\frac{1+\text{SSIM}\{I_{VAF}, I_{AAF}\}}{2}\right) + \gamma_{VAF}\text{TV}\{I_{VAF}\}$$
$$+ \delta_{VAF} L_1\{G_{VS}(I_{VAF}), I_{HS}\} - \varepsilon_{VAF}\log_{10}\left(\frac{1+\text{SSIM}\{G_{VS}(I_{VAF}), I_{HS}\}}{2}\right), \tag{7}$$

where $I_{VAF}$ denotes the virtual AF (DAPI channel) image generated by the VAF network model, $I_{AAF}$ denotes the physically recorded DAPI image (ground truth), $G_{VS}(\cdot)$ represents the selected best VS generator network model and $I_{HS}$ denotes the HS H&E image, which is the input of the VAF network model. The coefficients $\alpha_{VAF}$, $\beta_{VAF}$, $\gamma_{VAF}$, $\delta_{VAF}$ and $\varepsilon_{VAF}$ were empirically set as 1, 0.05, 0.01, 0.05 and 0.01, respectively, for both the kidney VAF network and the lung VAF network. The first three terms of the loss function were designed to ensure that the virtually generated AF DAPI image itself closely approximates the actual DAPI image. The last two terms of the loss function above were dedicated to preserving physical consistency of the VAF model, which means when reconverting the virtually generated AF image ($I_{VAF}$) back to the brightfield H&E stained domain through the optimal VS generator model, it should closely match the actual HS H&E image with minimal discrepancies.

The learning rate for optimizing VAF network was set as $10^{-5}$. The Adam optimizer was used for the network training. The batch size was set as 16. Similar to the VS generator model, a progressive strategy[67] was also used during the training process of VAF models. Once the networks converge, the best VAF network models for kidney and lung samples were selected on the basis of the lowest validation loss and expert visual evaluations.

### Definition of good-staining and poor-staining models for VS

During the training of the VS model, the checkpoint after each epoch $e$ was saved with its corresponding validation loss $l$. The validation loss has the same form as the training loss but calculated on the validation set. A complete convergence is determined with no considerable change of validation loss for 100 or more epochs, and an empirical threshold ($e_0, l_0$), corresponding to a picked starting point of convergence, is determined on the basis of the fluctuation of the validation loss curve, where $e_0, l_0$ stand for the thresholds of epoch number and validation loss, respectively. Then, a grey zone is marked to avoid the ambiguity of checkpoints in this zone. For the VS model checkpoints of human kidney tissue, we set the grey zone as $e \in [e_0 - 50, e_0 + 50]$ and $l \in [0.97l_0, 1.03l_0]$, and for the VS model checkpoints of human lung tissue, the corresponding grey zone is $e \in [e_0 - 100, e_0 + 100]$ and $l \in [0.97l_0, 1.03l_0]$. The checkpoints outside of the grey zone satisfying $e < e_0$ and $l > l_0$ are labelled as poor-staining models, and those with $e > e_0$ and $l < l_0$ are labelled as good-staining models. Our model selection methodology relies on the validation loss metrics, including the mean absolute error, SSIM and TV, and is frequently utilized in the field of image generation and translation[68,69]. In addition, a board-certified pathologist further verified the correctness of the model definitions. Supplementary Fig. 12 illustrates the thresholding and checkpoint selections for both human kidney and lung tissue VS models.

### Division of good-staining and poor-staining histochemical WSIs

To determine whether a histochemically stained WSI is of good quality or bad quality, three board-certified pathologists reviewed and assessed each WSI from three aspects: nuclear detail, cytoplasmic detail and extracellular detail. For each aspect, the pathologists were requested to assign a score to every WSI on a scale from 1 and 4, where 4 indicates 'perfect' quality, 3 denotes 'very good', 2 corresponds to 'acceptable' and 1 represents 'unacceptable' quality. The final quality score for each WSI was calculated by averaging the scores across all three aspects from all three pathologists. WSIs with an average score above 3 are classified as high-quality (good-staining) HSs, whereas those with an average score below 2 are classified as low quality (bad staining).

### Dataset preparation and splitting of AQuA

For human kidney samples, a set of 1,054 non-overlapping AF FOVs (each 1,424 × 1,424 pixels) were collected from seven individuals for training and validation, and a set of 76 non-overlapping AF FOVs collected from another three subjects were used for testing. The kidney VS image datasets were generated by a set of VS models on the corresponding AF image sets. The training and validation dataset was generated by five good-staining and five poor-staining VS models, and each model randomly picked and stained 20% of the full set of AF images, producing ~2,100 VS images. Then, the dataset was randomly divided into training and validation sets at a ratio of 9:1. The testing dataset was generated by seven good-staining and six poor-staining VS models excluded from training, consisting of 988 VS images (456 positives and 532 negatives). Forward–backward cycles were inferred starting from these VS images and using another selected optimal VS model and VAF model to implement the following AF-H&E cycles.

For human lung samples, we collected 572 non-overlapping AF FOVs (each 1,024 × 1,024 pixels) from seven individuals for training and validation (a subset of the training and validation data of VS models) and 268 non-overlapping AF FOVs from another subject for testing. The training and validation dataset of VS images was generated by eight good-staining and seven poor-staining VS models, and each model randomly picked and stained 15% of the full set of AF images, generating ~2,300 VS images. Then, the dataset was randomly divided into training and validation sets at a ratio of 9:1. The testing dataset was generated by five good-staining and five poor-staining VS models excluded from training and contains 2,412 VS images (1,072 positives and 1,340 negatives). Forward–backward cycles were inferred starting from these VS images and using the selected optimal VS model and VAF model to implement the subsequent AF-H&E cycles.

The histochemically stained H&E image dataset of human lung tissue sections consists of a subset of 100 non-overlapping FOVs (each 2,048 × 2,048 pixels) for training, 16 non-overlapping FOVs for validation and 220 non-overlapping FOVs for testing (110 positives and 110 negatives). The testing FOVs came from subjects excluded from the training and validation subsets. Each subset has equal number of positive and negative images. Forward–backward cycles were inferred starting from the HS images and using the same VS and VAF models as in the experiment of lung VS images to implement subsequent AF-H&E cycles.

### Architecture and training schedule of AQuA-Net

AQuA-Net starts with the $T$ VS–AF cycles between histological stain and AF domains. Let us denote the initial AF measurement as $x_0$ and the given VS image as $y_0$, the input sequence $(x_0, y_0, \ldots, x_{T-1}, y_{T-1})$ are formulated by iteratively passing $x_t$ and $y_t$ through VS and VAF networks $G_{VS}$ and $G_{VAF}$, respectively,

$$x_{t+1} = G_{VAF}(y_t), t = 0, \ldots T - 2$$

$$y_t = G_{VS}(x_t), t = 1, \ldots, T - 1, \tag{8}$$

Here, $G_{VS}$ and $G_{VAF}$ always use the predetermined, fixed best checkpoints with the lowest validation loss, which are independent of $(x_0, y_0)$. It is worth noting that specially for the HS image datasets of human lung tissue sections, actual AF measurement is not accessible. Therefore, we shift the VAF sequence one time step backward and substitute the AF measurement with $G_{VAF}(y_0)$. In other words, AQuA-Net for HS images takes in the input sequence $(y_0, x_0, \ldots, y_{T-1}, x_{T-1})$, defined as

$$x_t = G_{VAF}(y_t), t = 0, \ldots T - 1$$

$$y_{t+1} = G_{VS}(x_t), t = 0, \ldots, T - 2. \tag{9}$$

The pretrained backbone includes all the residual blocks and the 2D average pooling layer in a ResNet-50 model trained on ImageNet-1K[50]. A sequence of $T$ VS and a sequence of $T$ AF images (including virtual AF images and the actual AF measurement) are separately passed through this backbone to extract spatial features $f_s \in R^{B \times 2T \times C}$, where $B$ is the batch size and $C$ denotes the channel number of the feature after the 2D average pooling layer. Then, two temporal convolution layers with $1 \times 1$ kernels and ReLU activations integrate the temporal information over the cycles and generate a spatial-temporal feature $f_{st} \in R^{B \times C}$ squeezed along the temporal axis. At the end, two dense layers connected by a ReLU activation function project $f_{st}$ onto the predicted logits for positive and negative image classes.

The loss function for the classifier was the BCE between the predicted confidence scores $s \in R$ and the labels $y \in \{0,1\}$, as defined in equation (5), where $s$ corresponds to $p$ in and $y$ refers to $q$ in equation (5).

An Adam optimizer with learning rate $10^{-4}$ to optimize the classifier while the pretrained backbone remains fixed throughout the training, validation and testing stages. All classifiers were trained for 75 epochs, and the checkpoints with the best validation accuracy were picked for testing.

During the transfer learning on VS images of unseen organ types, the pretrained AQuA models were fine tuned on the full training dataset of the target tissue with a learning rate of $10^{-4}$ for 75 epochs. Later, the checkpoints with the best validation accuracy were picked for testing. For transfer learning on the TCGA dataset, the pretrained AQuA model was fine tuned with a learning rate of $10^{-5}$ for 10 epochs, and the checkpoint with the best validation accuracy was used for blind testing.

All models in this study were implemented using standard Python and PyTorch[70] libraries on a computer with Intel Core i9-12900F, 64GB RAM and a Nvidia RTX 3090 graphic card. Training for a classifier with

the pretrained backbone takes ~7 h, and its inference (batch size of 30, $T = 5$) takes 1 ms for a single 712 × 712 pixel FOV on average. The average inference time of VS and VAF models (batch size of 8) is ~2.6 ms for a single 1,024 × 1,024 pixel FOV.

### Implementation of TransMIL

Following the work of Shao et al.[47], TransMIL was trained, validated and tested on the HS image dataset of human lung tissue samples. Each full-resolution HS image (2,048 × 2,048 pixel) was first cropped into a 9 × 9 grid consisting of 224 × 224 patches. The same pretrained ResNet-50 backbone was utilized to encode the sequence of patches, producing an encoding matrix of 81 × 2,048 for further processing by the transformer. An AdamW optimizer with an initial learning rate of $10^{-4}$ and a BCE loss function were employed for optimization. All Trans-MIL models were trained for 100 epochs from scratch, and the checkpoints with the best validation accuracy were picked for testing.

### Comparison between HistoQC and AQuA

The quality control detection results for the HistoQC model were generated using the open-source HistoQC model with its default configuration (version 2.1)[48,49]. We optimized the parameter 'blur_radius' to 5 to improve the detection accuracy. In our comparative analysis, the acceptance probability maps predicted by the AQuA model were obtained by cropping the test HS WSIs into 2,048 × 2,048 pixel patches without overlapping, which were then independently fed through the AQuA model to obtain the predicted acceptance scores.

### Majority voting

Each ensemble was trained with the same training and validation datasets and hyperparameters except for the random seeds. We initially trained ten ensembles for each sample type and selected the best combination of $C$ ensembles out of 10 on the basis of the validation set. During transfer learning, only the $C$ ensembles of the best combination on the pretrained tissue type were transferred and then tested on the target tissue type.

Soft voting is applied throughout this work for the classifier ensembles[71]. Given $C$ ensembles, the confidence score after voting is defined as

$$s_{(C)} = \frac{1}{C} \sum_{i=1}^{C} s_{(i)}, \tag{10}$$

where $s_{(i)}, i = 1, \ldots, C$ is the confidence score from $i$th ensemble.

### Evaluation metrics

In this study, histochemical-based supervised image quality assessment involves three metrics. Given a VS image $x$ and the corresponding HS image $y$ (ground truth), the m.s.e., PCC and PSNR are calculated using

$$\text{m.s.e.}(x, y) = \frac{1}{N^2} \sum_{i,j} (x_{ij} - y_{ij})^2, \tag{11}$$

$$\text{PCC}(x, y) = \frac{\sum_{i,j} (x_{ij} - \bar{x})(y_{ij} - \bar{y})}{\sqrt{\sum_{i,j}(x_{ij} - \bar{x})^2 \sum_{i,j}(y_{ij} - \bar{y})^2}}, \tag{12}$$

$$\text{PSNR}(x, y) = 20\log_{10} \frac{255}{\sqrt{\text{m.s.e.}(x, y)}}. \tag{13}$$

Here $i, j$ are two dimensional indices of pixels, and $\bar{x}, \bar{y}$ are the mean values of $x, y$, respectively. We assume $x, y$ are $N \times N$ 8-bit images.

In addition to these, $t$-statistics and KL divergence are leveraged in our work to quantify the distinction between two distributions. All the $t$-tests in this study are conducted with the null hypothesis

$H_0 : \mu_1 = \mu_2$, where $\mu_1, \mu_2$ stand for the mean values of the two populations in comparison. The two-sided $t$-statistics are calculated as

$$t = \frac{\mu_1 - \mu_2}{s_p \times \sqrt{\frac{1}{n_1} + \frac{1}{n_2}}} \qquad (14)$$

$$s_p = \sqrt{\frac{(n_1 - 1) s_1^2 + (n_2 - 1) s_2^2}{n_1 + n_2 - 2}}, \qquad (15)$$

where $s_1^2, s_2^2$ are the variances of the two populations, and $n_1, n_2$ are the numbers of samples drown from each population.

KL divergence is calculated between the distribution of the confidence scores of the negative samples (or the good-staining models) and that of the positive samples (or the poor-staining models). For this, we first computed the sample probability density function between 0 to 1 with 0.01 bin width. Let us denote the probability density function of negative and positive samples as $p_n$ and $p_p$ respectively, and the bins as $b_i, i = 1, 2, \cdots, 100$, the KL divergence between them can be written as

$$D_{KL} (p_n \parallel p_p) = \sum_i \log \left( \frac{p_n (b_i) + \epsilon}{p_p (b_i) + \epsilon} \right) p_n (b_i), \qquad (16)$$

where $\epsilon = 0.001$ to avoid infinite values.

For classification tasks involved in this work, we denote the number of true positives, true negatives, actual positives and actual negatives as TP, TN, P and N, respectively. The accuracy, sensitivity and specificity are respectively defined as

$$\text{Accuracy} = \frac{TP + TN}{P + N}, \quad \text{sensitivity} = \frac{TP}{P}, \quad \text{specificity} = \frac{TN}{N}. \qquad (17)$$

The true positive rate (TPR) and false positive rate (FPR) are defined as

$$\text{TPR} = \text{sensitivity}, \quad \text{FPR} = 1 - \text{specificity}.$$

The precision and recall are defined as

$$\text{Precision} = \frac{TP}{TP + FP}, \quad \text{recall} = \text{TPR}.$$

### Pathologist evaluations
Three board-certified pathologists participated in the evaluation process. For each of human kidney and lung sample types, 200 VS images were randomly selected from the test set with 1:1 ratio of positive and negative images. Each pathologist independently scored each VS image according to three metrics: the stain quality of nuclei, cytoplasm and extracellular space. Each metric score belongs to $\{1, 2, 3, 4\}$, where the 4 refers to the best quality, and each pathologist was asked to give a pass/fail label, indicating whether the image has acceptable or unacceptable stain quality. Consensus among pathologists were established on the basis of the pass/fail labels such that on the FOVs with consensus, the diagnosticians unanimously agreed on the overall pass/fail label assessment. Pathologist consensus was reached on $N = 127$ FOVs for kidney samples and $N = 99$ FOVs for lung samples. Then, the scores from three pathologists on each metric are averaged to form their final score on the corresponding stain-quality metric. The average stain-quality score is the average of the three metrics. For average scores over three pathologists, we choose the interval $(2, 4]$ as the acceptable stain-quality range.

### LDA
In the VS model quality assessment using $M$-AQuA, we first obtained the logits of confidence scores for all positive and negative images in the training dataset. Then, two Gaussian distributions were fitted to the population of positive and negative images, denoted as $N(\mu_1, \sigma_1^2)$ and $N(\mu_2, \sigma_2^2)$, respectively. By the central limit theorem, the parameters for these distributions can also be approximated by

$$\mu_k \cong \frac{1}{N_k} \sum_{i=1}^{N_k} t_{i,k}, \quad \sigma_k^2 \cong \frac{1}{N_k - 1} \sum_{i=1}^{N_k} (t_{i,k} - \mu_k)^2, \; k = 1, 2. \qquad (18)$$

Here, $t_{i,1}, i = 1, \cdots, N_1$ and $t_{i,2}, i = 1, \cdots, N_2$ represent the logits of confidence score for $i$th positive and negative image, respectively. $N_1, N_2$ are the numbers of positive and negative images in the training set, respectively. The LDA determines the discriminant threshold $\beta$ by[72]

$$\log (\sigma_1) + \frac{(\beta - \mu_1)^2}{2\sigma_1^2} = \log (\sigma_2) + \frac{(\beta - \mu_2)^2}{2\sigma_2^2}. \qquad (19)$$

### Hand-crafted analytical metrics evaluating histochemically stained H&E slides
To evaluate HS slides, we used two hand-crafted analytical metrics, that is, normalized nuclei count and average nuclei area; for this, a nuclei segmentation algorithm based on stain vector decomposition[73] was employed. Each HS image to be evaluated is separated into the nuclei channel (purple, corresponding to haematoxylin staining) and the cytoplasmic-extracellular matrix channel (pink, corresponding to eosin staining). Then, Otsu's thresholding[74] method and a series of morphological operations including image erosion and dilation were used to segment the binary nuclei map based on the separated nuclei channel. The normalized nuclei count (in the unit of $px^{-2}$) is defined as the number of connected components divided by the total number of pixels in the image, and the average nuclei area is determined by calculating the mean area of these connected components (in the unit of $px^2$).

### Grad-CAM visualization
Grad-CAM heat maps were calculated on the basis of the activation maps and gradients of the last residual block of the ResNet-50 backbone and averaged over the three convolutional layers within that block[45]. The raw activation maps were upsampled to match the resolution of the VS image. For each time step, Grad-CAM heat maps of $C = 5$ classifier ensembles were averaged and displayed. Gradient-based weights were applied to the heat map at each time step $t$ for the visualization and the generation of the averaged heat map.

### Quantitative evaluation of the VS model used in VS–AF iterations
To quantitatively assess the match between the VS images and their HS counterparts, we analysed 76 pairs of VS and HS kidney images, each with $1,424 \times 1,424$ pixels. First, to compare the colour distributions of the VS and HS images, they were converted from the RGB (red, green, blue) colour space into the YCbCr colour space. The intensity histograms for all 76 test FOVs were plotted for the Y, Cb and Cr channels, as shown in Supplementary Fig. 10b. Moreover, using the stain vector decomposition, we separated the RGB (red, green, blue) images into channels corresponding to eosin and haematoxylin staining, and their respective intensity histograms were presented in Supplementary Fig. 10c. To further evaluate the similarity between VS and HS images, the PSNR and SSIM values were calculated for the same test FOVs based on equations (13) and (3), respectively, and reported in Supplementary Fig. 11b,c. Moreover, the nuclei segmentation and the quantification of different features—the number of nuclei per FOV and average nuclei area—were performed using the same methodologies outlined in the previous section. The box plots used to illustrate the distribution of the number of nuclei per FOV and average nuclei area of VS and HS images

are shown in Supplementary Fig. 11d,e. Then, the Hellinger distances[75] for each feature were calculated as follows

$$D_{\text{Hellinger}} = \sqrt{1 - \sum_{i=1}^{k} \sqrt{p_i q_i}}, \qquad (20)$$

where $P = (p_1, p_2, \ldots, p_k)$ represents the discrete probability distribution of the histograms for each feature in VS images, $Q = (q_1, q_2, \ldots, q_k)$ in HS images and $k$ is the number of bins of histograms, which is 7 in measuring the number of nuclei per FOV and 6 for the average nuclei area. Lastly, the $G$-tests[76] were employed to statistically compare the distribution differences in the number of nuclei per FOV and average nuclei area between the VS and HS images, with a statistical significance level of 0.05 was adopted for both $G$-tests.

### Reporting summary

Further information on research design is available in the Nature Portfolio Reporting Summary linked to this article.

### Data availability

A portion of the testing dataset is shared and referenced in the code repository without any links or identifiers to the patients, available via Zenodo at https://doi.org/10.5281/zenodo.15107104 (ref. 77). The TCGA dataset labels on human lung tissue WSIs are also shared in the same repository[77].

### Code availability

The codes for the deep learning models used in this work[78] (written in Python 3.9.16 and PyTorch 1.13.0) are available via Zenodo at https://doi.org/10.5281/zenodo.15122854 (ref. 78). The code for analysing the results was written in Python using standard, open-source Python libraries.

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

## Acknowledgements

We acknowledge K.D. Hann and T. Liu of UCLA for their assistance with data, S. Pan and X. Wang of UCLA for their assistance with dataset preprocessing and X. Yang and H. Chen of UCLA for valuable discussions. A.O. acknowledges the support of the Volgenau Chair at UCLA and the V.M. Watanabe Award.

## Author contributions

A.O. and L.H. conceived the research. L.H. and Y.L. conducted the experiments and analysed the results. Y.L. collected and preprocessed the experimental data. N.P., T.K.H. and W.D.W. performed histological evaluation on the results. L.H., Y.L., N.P. and A.O. contributed to the preparation of the manuscript. A.O. supervised the research.

## Competing interests

A.O., L.H. and N.P. have a pending patent application on the presented technology.

## Additional information

**Extended data** is available for this paper at https://doi.org/10.1038/s41551-025-01421-9.

**Correspondence and requests for materials** should be addressed to Aydogan Ozcan.

**Extended Data Table 1 | Study on the hyperparameters T and C**

| | Lung sample | | | Kidney sample | |
|---|---|---|---|---|---|
| | **Acc. @ 100% sensitivity threshold ($\alpha$)** | **Sen. @ 100% sensitivity threshold ($\alpha$)** | | **Acc. @ 100% sensitivity threshold ($\alpha$)** | **Sen. @ 100% sensitivity threshold ($\alpha$)** |
| T=1, C=1 | 0.8856 | 0.8284 | T=1, C=1 | 0.9302 | 0.9846 |
| T=1, C=3 | 0.9403 | 0.9897 | T=1, C=3 | 0.9666 | 0.9496 |
| T=1, C=5 | 0.9655 | 0.9869 | T=1, C=5 | 0.9838 | 0.9846 |
| T=2, C=1 | 0.9283 | 0.8965 | T=2, C=1 | 0.6903 | 0.9956 |
| T=2, C=3 | 0.9664 | 0.9701 | T=2, C=3 | 0.9929 | 0.9978 |
| T=2, C=5 | 0.9789 | 0.9869 | T=2, C=5 | 0.9970 | 0.9956 |
| T=3, C=1 | 0.8955 | 0.8853 | T=3, C=1 | 0.9443 | 0.9934 |
| T=3, C=3 | 0.9565 | 0.9869 | T=3, C=3 | 0.9919 | 0.9912 |
| T=3, C=5 | 0.9776 | 0.9953 | T=3, C=5 | 0.9980 | 0.9978 |
| T=4, C=1 | 0.8914 | 0.8834 | T=4, C=1 | 0.9636 | 0.9715 |
| T=4, C=3 | 0.9565 | 0.9254 | T=4, C=3 | 0.9949 | 0.9978 |
| T=4, C=5 | 0.9789 | 0.9841 | **T=4, C=5** | **0.9980** | **0.9978** |
| T=5, C=1 | 0.9005 | 0.8694 | T=5, C=1 | 0.4615 | 1.0000 |
| T=5, C=3 | 0.9129 | 0.9935 | T=5, C=3 | 0.9899 | 0.9912 |
| **T=5, C=5** | **0.9805** | **0.9953** | T=5, C=5 | 0.9960 | 0.9934 |

The best combinations were chosen from 10 ensembles. Thresholds were determined on the validation set. The optimal hyperparameters and the corresponding scores are highlighted.

**Extended Data Table 2 | Training consumption reduction using a pre-trained ResNet-50 backbone**

|  | # trainable param. (million) | Training time (min/epoch) |
|---|---|---|
| Pre-trained backbone | 2.1 | 5 |
| From scratch | 25.6 | 60 |

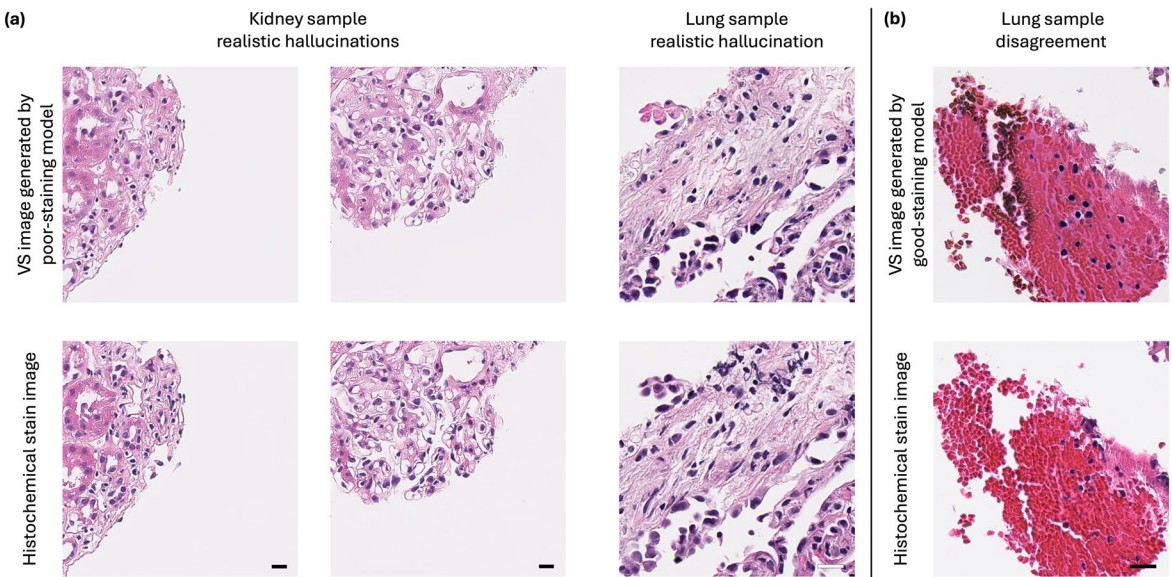

**Extended Data Fig. 1 | Examples of the images where AQuA disagrees with a group of pathologists.** (**a**) AQuA identifies VS images generated by poor-stain models, avoiding realistic hallucinations to be diagnosed by experts. On the other hand, pathologists were deceived by the VS images with realistic hallucinations. (**b**) Pathologists rejected this image from a good-staining VS

model on red blood cells (RBCs) while AQuA accepted it. The disagreement between AQuA and board-certified pathologists in this case is due to the under-representation of RBCs in the training data of AQuA. Also, refer to the Discussion section for a discussion of this. Scale bar: 20 μm.

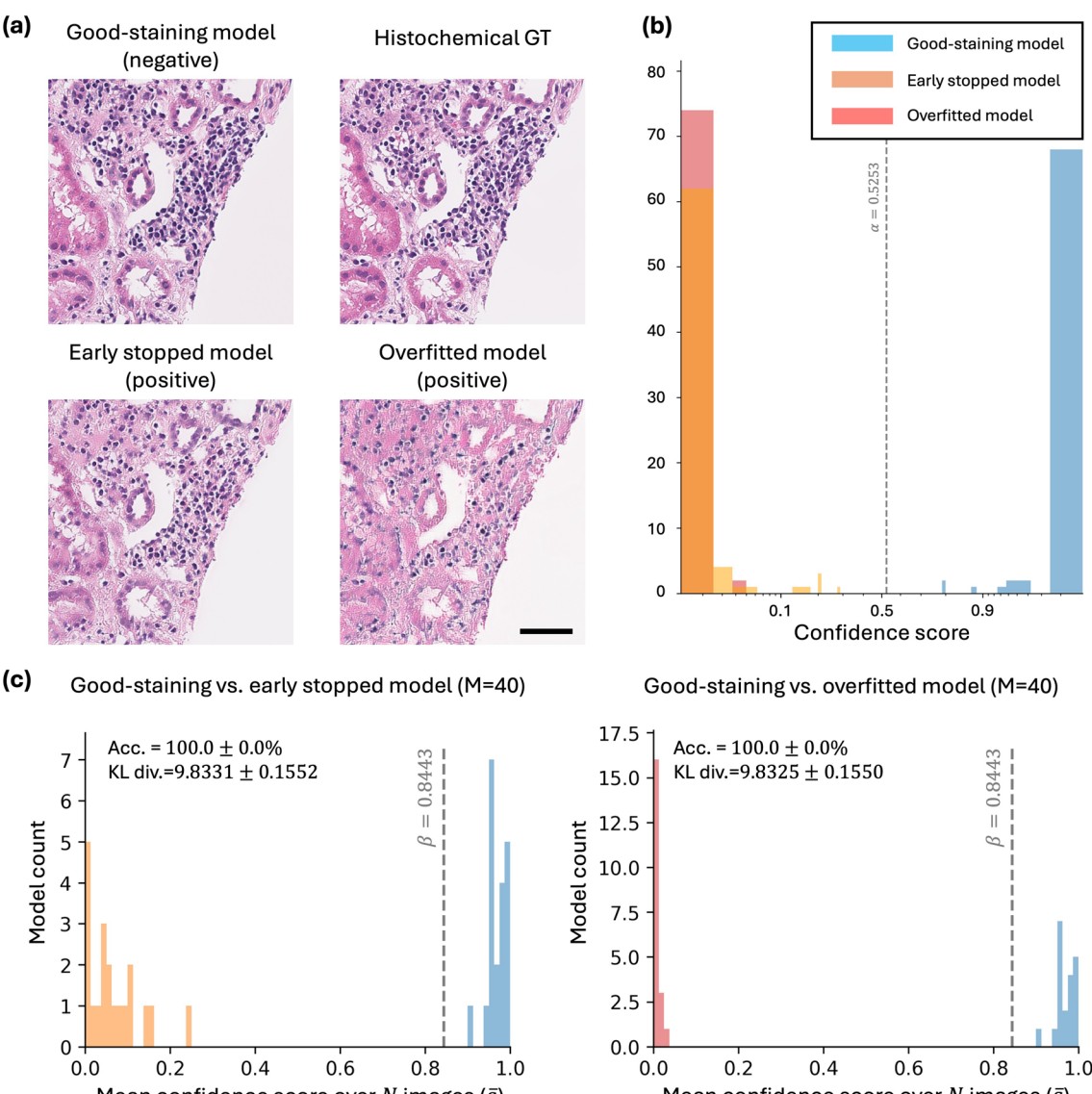

**Extended Data Fig. 2 | Successful external generalization of our method on overfitted models.** (**a**) Positive and negative images were generated by various VS models on the same FOV of a human kidney sample. The negative image was generated by a good-staining model, and the two positive images were generated by an early-stopped model and an overfitted model. The histochemical H&E image was registered as the reference (ground truth image). (**b**) Discrimination of

AQuA on the three models. During the training, the classifier only saw positive images from early stopped models, whereas it successfully generalized to positive images from the overfitted model. The threshold $\beta$ is determined on the training set. The histogram is plotted on the test set of $N = 76$ human kidney FOVs. (**c**) M-AQuA classification on $M = 40$ virtual staining models of human kidney samples. The ratio of negative and positive models is 1:1.

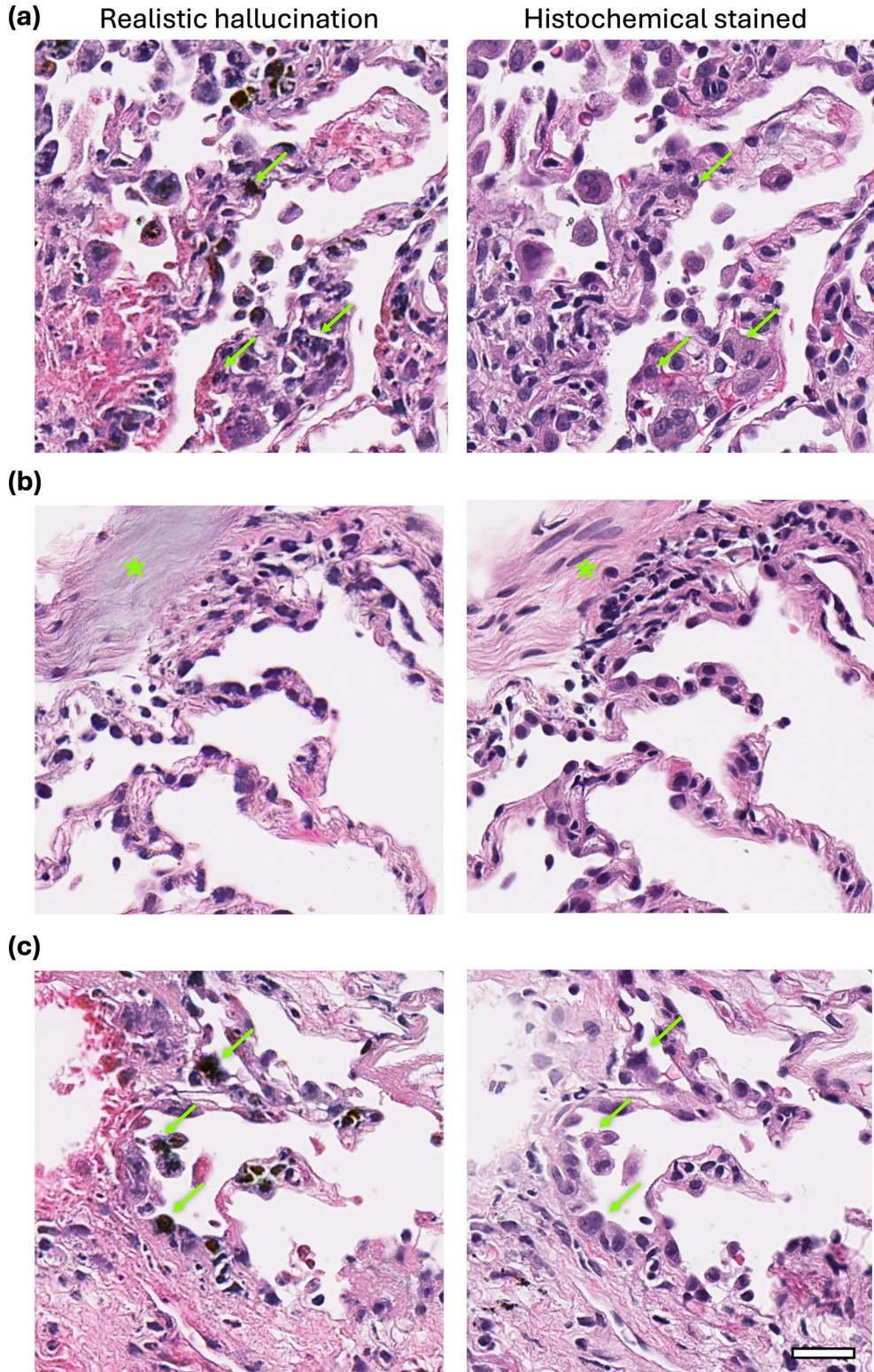

**Extended Data Fig. 3 | Examples of realistic hallucinations generated by VS on human lung tissue samples, in comparison with the corresponding HS reference images.** (**a**) Cellular hallucinations are observed in several locations (green arrows), mimicking atypical mitoses in the VS slide (left). HS slide demonstrates normal appearing cells. (**b**) Myxoid change in the VS slide (green asterisks) can raise a differential diagnosis of several benign and malignant lesions. HS slide demonstrates normal appearing stroma. (**c**) Abundance of anthracotic pigment in alveolar macrophages (green arrows) in the VS slide, suggesting potential smoking exposure. HS slide demonstrates a minimal amount of pigment. Scale bar: 20 µm.

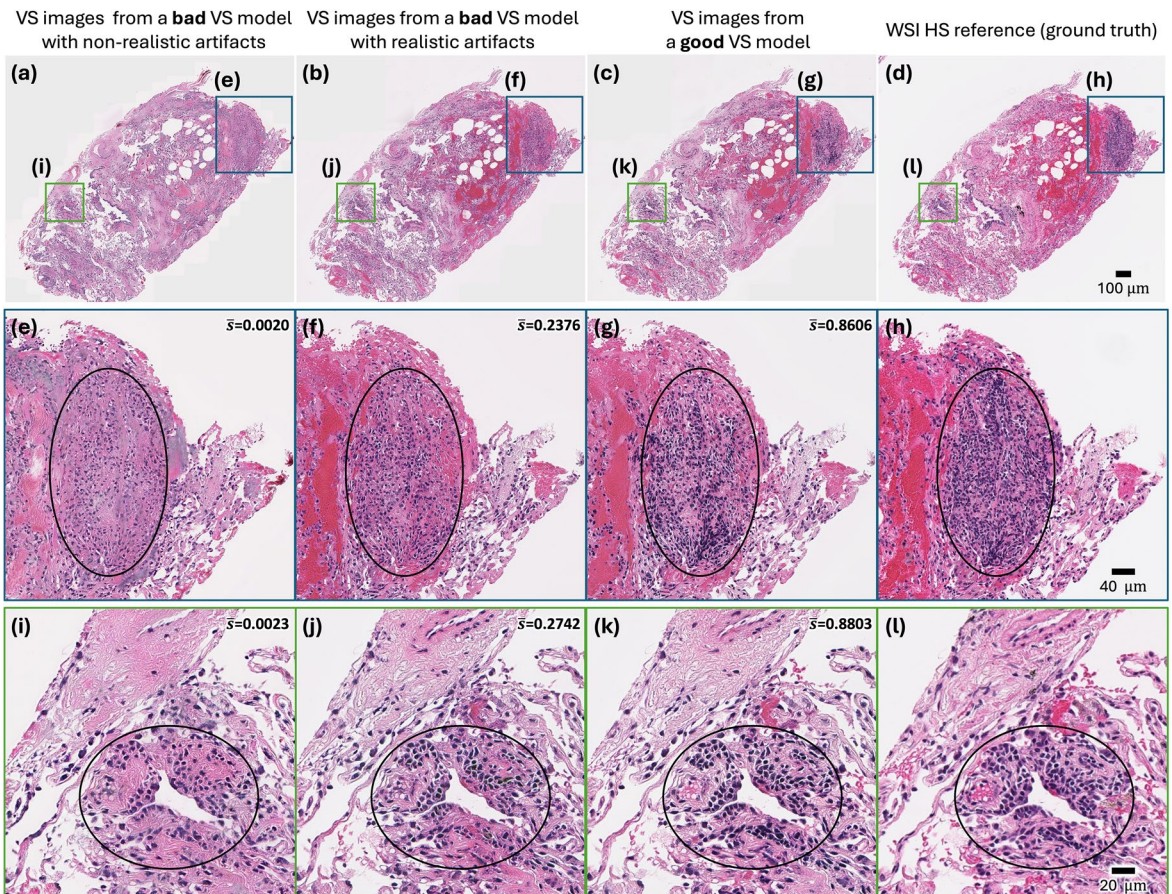

**Extended Data Fig. 4 | Comparisons of images from two poor-staining and one good-staining VS models, identified by the AQuA workflow, against their HS references in a lung transplant rejection case. (a-d)** Large region images from a poor-staining VS model with non-realistic hallucinations; a poor-staining VS model with realistic hallucinations; a good-staining VS model; and the corresponding HS reference. **(e-h)** Zoomed-in images from the same large regions - from the poor-staining VS models, the good-staining model and the HS reference. The circled region corresponds to cell infiltrate, indicative of acute transplant rejection. The confidence scores $\bar{s}$ predicted by the AQuA model are reported for the VS images of **(e-g)**. **(i-l)** Zoomed-in images from another diagnostically relevant region, maintaining the same comparison format as in **(e-h)**. The circled region corresponds to cell infiltrate, indicative of acute transplant rejection. The confidence scores predicted by the AQuA model are reported for the VS images of **(i-k)**.

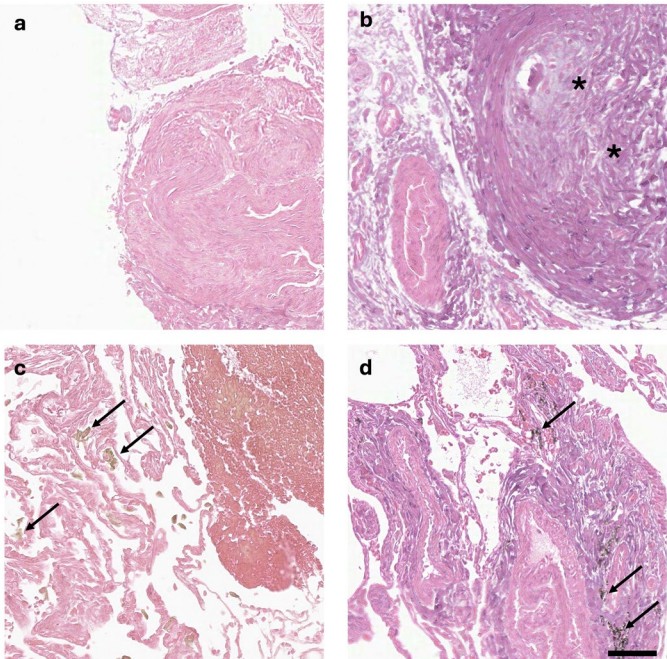

**Extended Data Fig. 5 | Examples of poorly stained HS slides.** (**a**) Complete lack of nuclear staining; (**b**) out of focus, blurry regions (black asterisks); (**c**) lack of nuclear staining and artefactual greenish pigment (black arrows); (**d**) appearance of anthracotic-like black pigment in abnormal shape and location (black arrows). Scale bar: 50 μm.

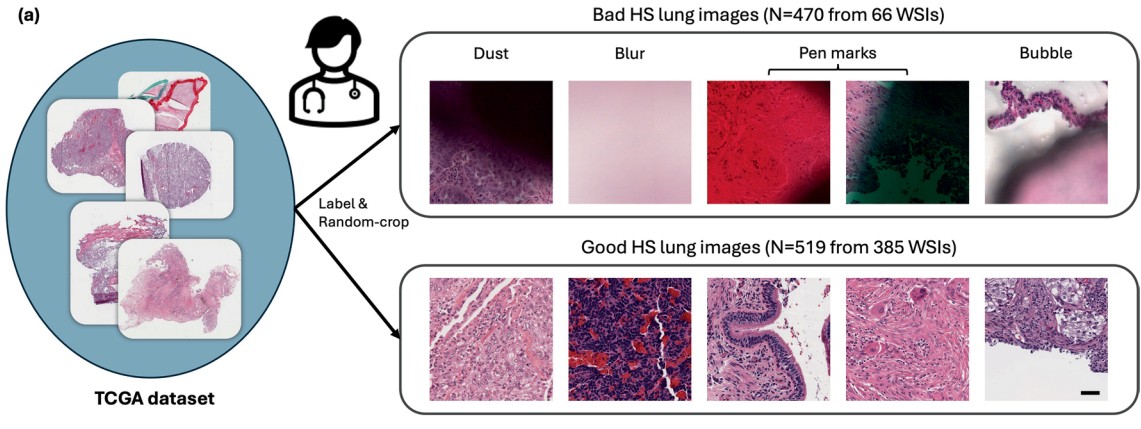

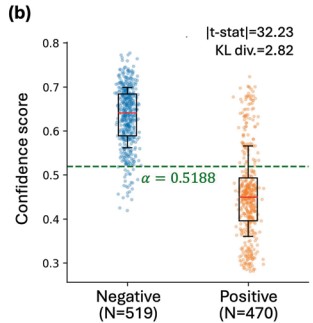

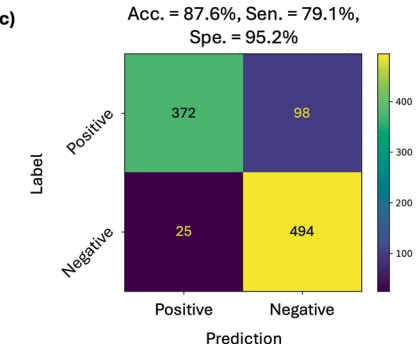

**Extended Data Fig. 6 | Autonomous quality assessment on TCGA image dataset.** (**a**) 519 good and 470 bad HS images were randomly cropped from the selected WSIs for testing. (**b**) Boxplot of confidence scores given by AQuA. AQuA model used transfer learning on a small subset (10 slides) of only well-stained TCGA WSIs. This blind testing reveals the external generalization capability of AQuA on bad TCGA WSIs from various tissue sites, staining protocols and imaging hardware since no bad HS images from the TCGA dataset were used in the training phase. (**c**) Confusion matrix of AQuA's testing results on the TCGA dataset. See the Methods for details. Scale bar: 50 μm.

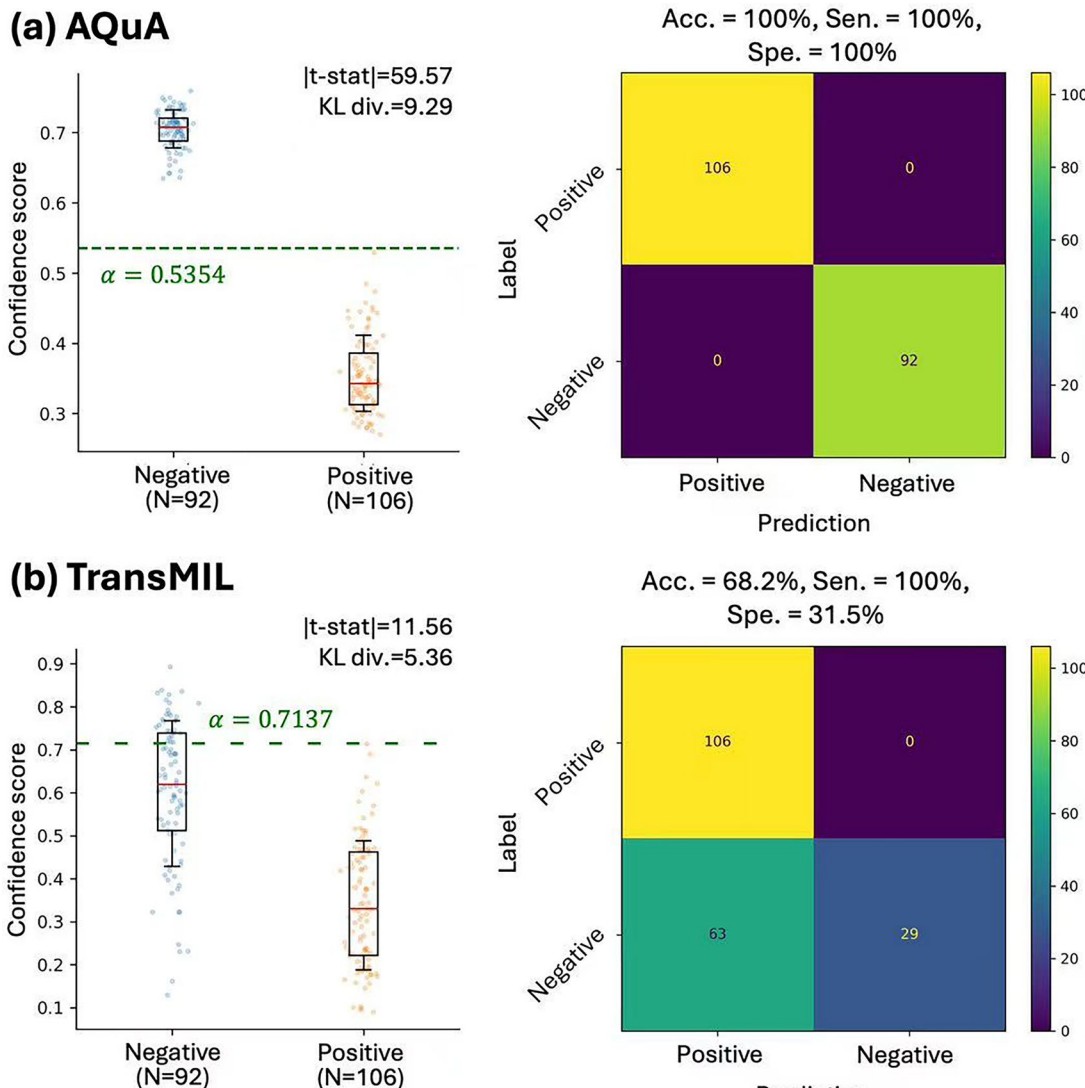

**Extended Data Fig. 7 | Classification performance comparison between (a) AQuA and (b) TransMIL on detecting good and bad HS images of human lung tissue samples.** Both models utilized the best 5 out of 10 ensembles and 100% sensitivity threshold ($\alpha$) acquired on the validation set. Box plot, 25th-75th percentiles; center, median; whisker, 16th-84th percentiles. See the Methods section for details on TransMIL implementation.

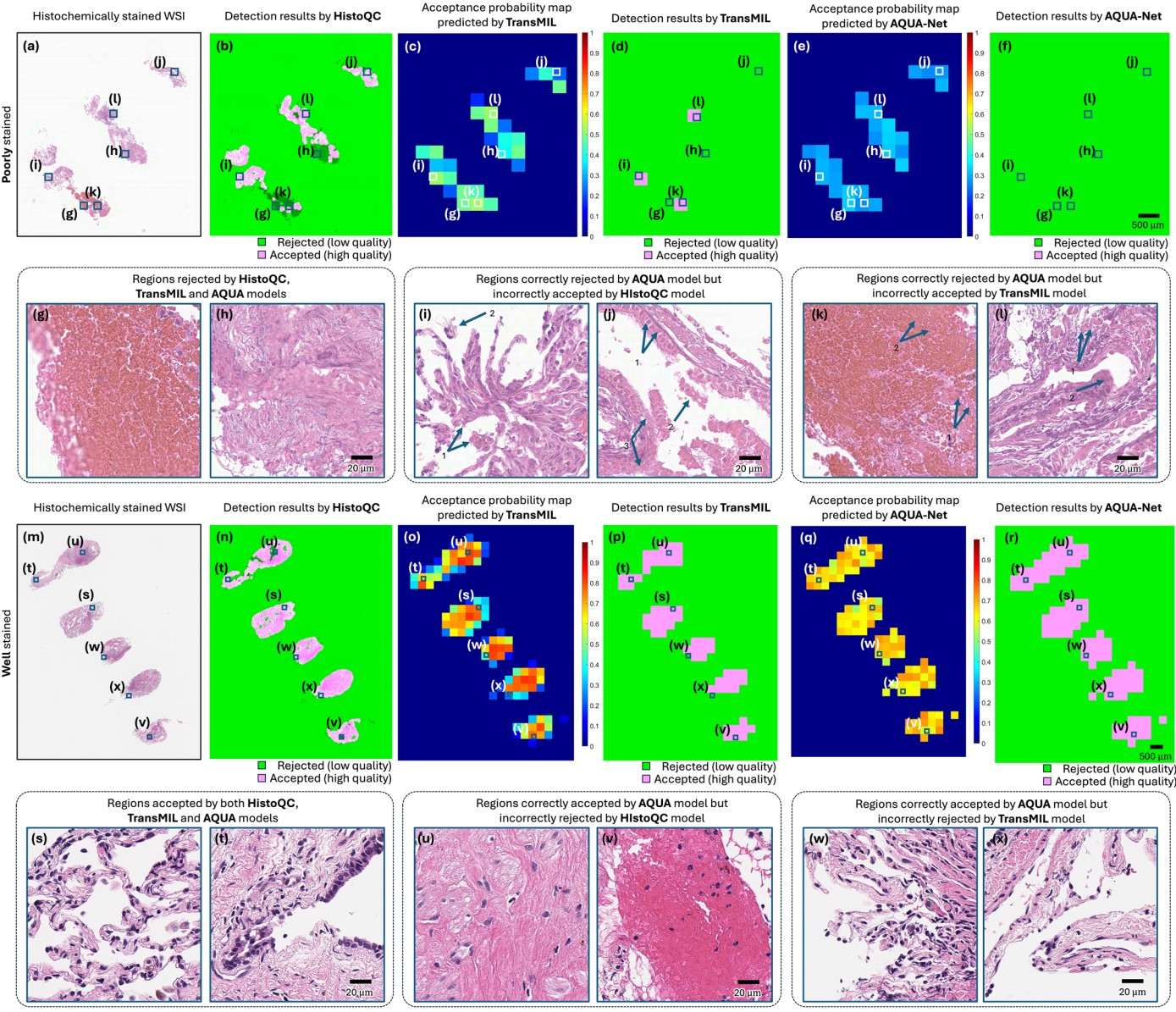

**Extended Data Fig. 8 | Quality control detection results from the HistoQC model, the TransMIL model and the AQuA model.** (**a**) The WSI of a poorly stained HS slide labelled by pathologists. (**b**) The detection results of (**a**) using the HistoQC model, where pink regions indicate high-quality stained areas accepted by the model, and green regions correspond to poorly stained areas rejected by the quality control model. (**c**) The acceptance probability map of (**a**) predicted by the TransMIL model, where values close to 1 indicate acceptance, while values near 0 signify rejection. (**d**) The detection results of (**a**) using the TransMIL model, which was obtained by thresholding (**c**) by 0.5. (**e**) The acceptance probability map of (**a**) predicted by the AQuA model. (**f**) The detection results of (**a**) using the AQuA model, which was obtained by thresholding (**e**) by 0.5. (**g**) One example zoomed-in tissue region rejected by the HistoQC model, the TransMIL model and AQuA model, which contains blurred red blood cells. (**h**) Another example zoomed-in tissue region rejected by the HistoQC model, the TransMIL model and AQuA model, which contains blurred out-of-focus areas. (**ii-j**) One example zoomed-in region with insufficient/missing hematoxylin staining, which was incorrectly accepted by the HistoQC model but correctly rejected by the AQuA model. Arrow 1 points to faded nuclei without haematoxylin staining, and arrow 2 corresponds to a detached tissue fragment. (**j**) Another example zoomed-in region incorrectly accepted by the HistoQC model but correctly rejected by the AQuA model. Arrow 1 also points to faded nuclei without haematoxylin staining. Arrow 2 corresponds to a detached tissue fragment, and arrow 3 indicates out-of-focus areas. (**k**) One example zoomed-in tissue region with red

blood cells obscuring the lung structure, which is correctly rejected by the AQuA model, but incorrectly accepted by the TransMIL model. Arrow 1 points to faded nuclei without haematoxylin staining, and arrow 2 corresponds to low-quality RBC staining. (**l**) Another zoomed-in tissue region with small areas of blurriness, which is correctly rejected by the AQuA model, but incorrectly accepted by the TransMIL model. Arrow 1 points to faded nuclei without haematoxylin staining, and arrow 2 corresponds to out-of-focus regions. (**m**) The WSI of a well-stained HS slide labelled by pathologists. (**n**) The detection results of (**m**) using the HistoQC model. (**o**) The acceptance probability map of (**m**) predicted by the TransMIL model. (**p**) The detection results of (**m**) using the TransMIL model, which was obtained by thresholding (**o**) by 0.5. (**q**) The acceptance probability map of (**m**) predicted by the AQuA model. (**r**) The detection results of (**m**) using the AQuA model, which was obtained by thresholding (**q**) by 0.5. (**s**) One example zoomed-in tissue region accepted by the HistoQC model, the TransMIL model and the AQuA model, which corresponds to well-stained alveolated lung tissue with scattered macrophage. (**t**) Another example zoomed-in tissue region accepted by the HistoQC model, the TransMIL model and the AQuA model, which corresponds to well-stained bronchiole lining with ciliated cells. (**u-v**) Two example zoomed-in regions with good staining quality which were incorrectly rejected by the HistoQC model but correctly accepted by the AQuA model. (**w-x**) Another two examples of zoomed-in regions with good staining quality that were incorrectly rejected by the TransMIL model but correctly accepted by the AQuA model.

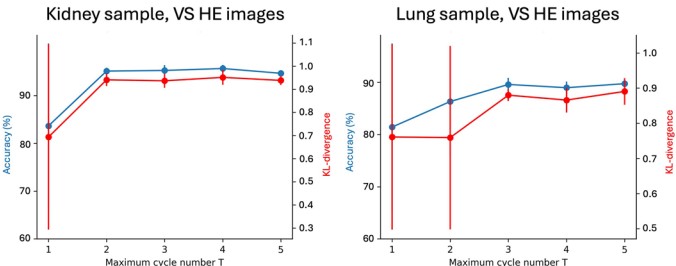

**Extended Data Fig. 9 | Study on hyperparameter T.** Mean ± standard deviation is calculated on 10 ensembles and the test sets of each sample type. KL divergence was calculated between the distributions of the confidence scores of positive and negative VS images.

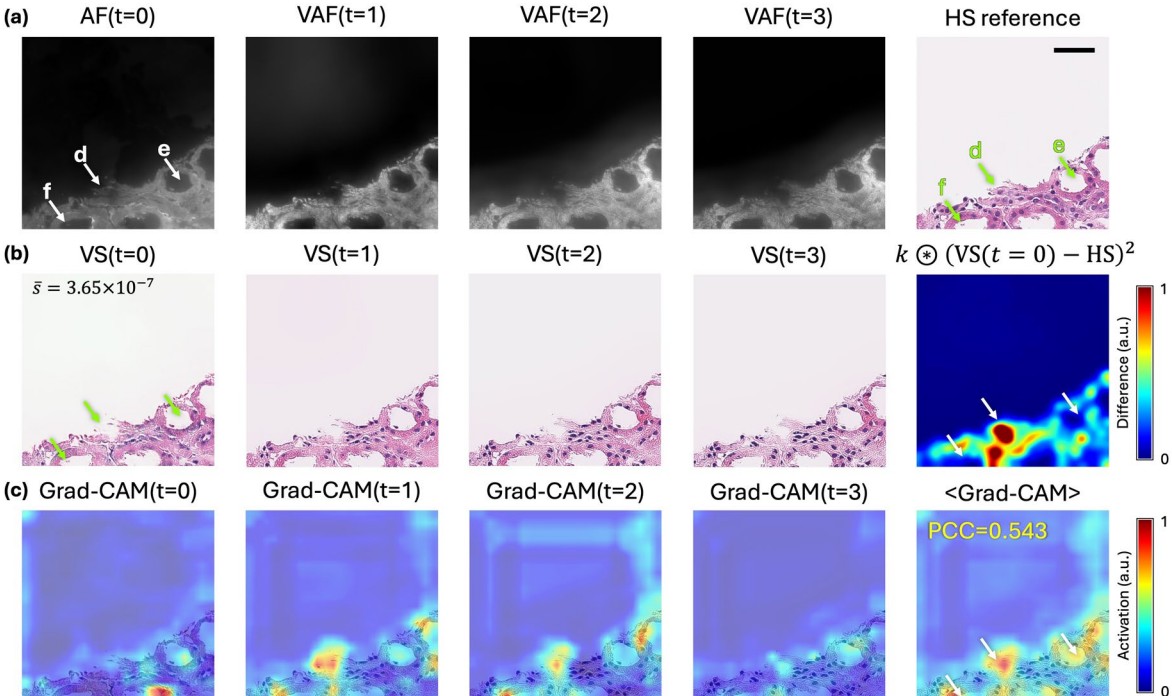

**Extended Data Fig. 10 | Spatial-temporal Grad-CAM visualization of the activation maps during the AQuA inference on poor VS images of human kidney sample. (a)** AF and VAF images in the VS-AF iterations, and the HS reference image (ground truth). **(b)** VS images in the VS-AF iterations, and the difference map between the VS(t = 0) and the HS reference image is smoothed by a Gaussian kernel $k$ with $\sigma = 15.5$px to match the resolution of Grad-CAM. ⊛ stands for 2D convolution. **(c)** Grad-CAM heatmaps for VS(t) and the averaged heatmap. $< \cdot >$ stands for averaging over time. PCC values were calculated between the difference map and the averaged heatmap. **(d)** A broken tissue area after virtual staining was successfully identified by AQuA without using an HS reference image. **(e)** A hollow area with strong AF background noise, marked by AQuA. **(f)** A tiny residual tissue not cleared in sample preparation was successfully detected by AQuA. **(d, e, f)** are pointed by arrows in **(a)**. Scale bar: 50 μm.

# Reporting Summary

## Statistics

For all statistical analyses, confirm that the following items are present in the figure legend, table legend, main text, or Methods section.

| n/a | Confirmed | |
|---|---|---|
| ☐ | ☒ | The exact sample size (*n*) for each experimental group/condition, given as a discrete number and unit of measurement |
| ☐ | ☒ | A statement on whether measurements were taken from distinct samples or whether the same sample was measured repeatedly |
| ☐ | ☒ | The statistical test(s) used AND whether they are one- or two-sided<br>*Only common tests should be described solely by name; describe more complex techniques in the Methods section.* |
| ☒ | ☐ | A description of all covariates tested |
| ☒ | ☐ | A description of any assumptions or corrections, such as tests of normality and adjustment for multiple comparisons |
| ☐ | ☒ | A full description of the statistical parameters including central tendency (e.g. means) or other basic estimates (e.g. regression coefficient) AND variation (e.g. standard deviation) or associated estimates of uncertainty (e.g. confidence intervals) |
| ☐ | ☒ | For null hypothesis testing, the test statistic (e.g. *F*, *t*, *r*) with confidence intervals, effect sizes, degrees of freedom and *P* value noted<br>*Give P values as exact values whenever suitable.* |
| ☒ | ☐ | For Bayesian analysis, information on the choice of priors and Markov chain Monte Carlo settings |
| ☒ | ☐ | For hierarchical and complex designs, identification of the appropriate level for tests and full reporting of outcomes |
| ☒ | ☐ | Estimates of effect sizes (e.g. Cohen's *d*, Pearson's *r*), indicating how they were calculated |

*Our web collection on statistics for biologists contains articles on many of the points above.*

## Software and code

Policy information about availability of computer code

| Data collection | AF (autofluorescence) images of unlabeled kidney tissue slides were captured by an Olympus IX-83 microscope (controlled by MetaMorph microscope automation software) with a 40 × /0.95NA (UPLSAPO, Olympus) objective lens, under the DAPI filter cube (Semrock OSFI3-DAPI5060C, EX 377/50 nm EM 447/60 nm). Similarly, for unlabeled lung tissue slides, the AF images were obtained using a Leica DMI8 microscope (controlled by Leica LAS X microscopy automation software) with a 40×/0.95 NA objective lens (Leica HC PL APO 40×/0.95 DRY), under the DAPI filter cube. After undergoing standard histochemical H&E staining, the stained tissue slides were digitized using a Leica Biosystems Aperio AT2 brightfield slide scanner. |
|---|---|
| Data analysis | The codes for the deep learning models used in this work (written in Python 3.9.16 and PyTorch 1.13.0) can be accessed through:<br>https://drive.google.com/drive/folders/1ztfS6hTkyU-mUrXuHI6LAgUFJ1-hcEyY?usp=share_link<br>Upon acceptance of the manuscript, it will also be placed in GitHub for public release. The trained model and demo data are uploaded and available in the same code repository. The code for analyzing the results was written in Python using standard, open-source Python libraries. |

For manuscripts utilizing custom algorithms or software that are central to the research but not yet described in published literature, software must be made available to editors and reviewers. We strongly encourage code deposition in a community repository (e.g. GitHub). See the Nature Portfolio guidelines for submitting code & software for further information.

# Data

Policy information about availability of data

All manuscripts must include a data availability statement. This statement should provide the following information, where applicable:

- Accession codes, unique identifiers, or web links for publicly available datasets
- A description of any restrictions on data availability
- For clinical datasets or third party data, please ensure that the statement adheres to our policy

A portion of the testing dataset is also shared and referenced in the code repository without any links or identifiers to the patients, which is made available at: https://drive.google.com/drive/folders/1ztfS6hTkyU-mUrXuHI6LAgUFJ1-hcEyY?usp=share_link. The TCGA dataset (an open-sourced and publicly available dataset) labels on human lung tissue WSIs are also shared in the same repository.

# Research involving human participants, their data, or biological material

Policy information about studies with human participants or human data. See also policy information about sex, gender (identity/presentation), and sexual orientation and race, ethnicity and racism.

| | |
|---|---|
| Reporting on sex and gender | *Use the terms sex (biological attribute) and gender (shaped by social and cultural circumstances) carefully in order to avoid confusing both terms. Indicate if findings apply to only one sex or gender; describe whether sex and gender were considered in study design; whether sex and/or gender was determined based on self-reporting or assigned and methods used. Provide in the source data disaggregated sex and gender data, where this information has been collected, and if consent has been obtained for sharing of individual-level data; provide overall numbers in this Reporting Summary. Please state if this information has not been collected. Report sex- and gender-based analyses where performed, justify reasons for lack of sex- and gender-based analysis.* |
| Reporting on race, ethnicity, or other socially relevant groupings | *Please specify the socially constructed or socially relevant categorization variable(s) used in your manuscript and explain why they were used. Please note that such variables should not be used as proxies for other socially constructed/relevant variables (for example, race or ethnicity should not be used as a proxy for socioeconomic status). Provide clear definitions of the relevant terms used, how they were provided (by the participants/respondents, the researchers, or third parties), and the method(s) used to classify people into the different categories (e.g. self-report, census or administrative data, social media data, etc.) Please provide details about how you controlled for confounding variables in your analyses.* |
| Population characteristics | *Describe the covariate-relevant population characteristics of the human research participants (e.g. age, genotypic information, past and current diagnosis and treatment categories). If you filled out the behavioural & social sciences study design questions and have nothing to add here, write "See above."* |
| Recruitment | *Describe how participants were recruited. Outline any potential self-selection bias or other biases that may be present and how these are likely to impact results.* |
| Ethics oversight | *Identify the organization(s) that approved the study protocol.* |

Note that full information on the approval of the study protocol must also be provided in the manuscript.

# Field-specific reporting

Please select the one below that is the best fit for your research. If you are not sure, read the appropriate sections before making your selection.

☒ Life sciences  ☐ Behavioural & social sciences  ☐ Ecological, evolutionary & environmental sciences

For a reference copy of the document with all sections, see nature.com/documents/nr-reporting-summary-flat.pdf

# Life sciences study design

All studies must disclose on these points even when the disclosure is negative.

| | |
|---|---|
| Sample size | After data preprocessing, we obtained 1054 non-overlapping AF-H&E image patch pairs (1424×1424 pixels) for kidney samples, and 1068 pairs (1024×1024 pixels) for lung samples, to train and validate the VS and VAF networks. The kidney samples originated from 10 unique patients (7 used for training and validating the VS/VAF and AQuA models, and 3 for testing), while the lung samples came from 29 unique patients. Among these 29 unique patients, 18 cases were allocated for training and validating the VS/VAF models, 7 out of these 18 cases were chosen for training and validating the AQuA model. The rest of the 11 cases were left for testing the AQuA model. For AQuA on human kidney samples, a set of 1054 non-overlapping AF FOVs (1424×1424 pixels) were collected from 7 individuals for training and validation, and a set of 76 non-overlapping AF FOVs collected from another 3 subjects were used for testing. For AQuA on human lung samples, we collected 572 non-overlapping AF FOVs (1024×1024 pixels) from 7 individuals for training and validation (a subset of the training and validation data of VS models), and 268 non-overlapping AF FOVs from another subject for testing. Additionally, the histochemical-stained H&E image dataset of human lung tissue sections consists of a subset of 100 non-overlapping FOVs (2048×2048 pixels) for training, 16 non-overlapping FOVs for validation, and 220 non-overlapping FOVs for testing of AQuA models. Moreover, for the external generalization test using TCGA dataset, two small subsets including 36 patches from 8 good whole slide images (WSIs), and 8 patches from 2 good WSIs were used as the negative |

samples for training and validation, respectively – i.e., no bad WSIs were used in the transfer learning step. The rest of the 989 patches (519 good and 470 bad patches) generated from 385 good WSIs and 66 bad WSIs were used for blind testing.

Data exclusions | The training FOVs and testing FOVs were exclusive, and the testing FOVs were strictly different from the training slides.

Replication | VS and VAF models were trained on two datasets of human kidney and lung tissue samples, and then blindly tested on corresponding test sets. 10 ensembles of AQuA model were repeated for each training dataset and combination hyperparameters. After training, each ensemble was blindly tested on the corresponding test set.

Randomization | The training, validation and testing image datasets were randomly partitioned. Ensembles used randomly generated seeds and initializations.

Blinding | All the testing results generated by the trained neural networks were blindly performed on new FOVs excluded from the training dataset. The blind testing FOVs were also captured on new slides from new patients that did not appear in the training dataset.

# Reporting for specific materials, systems and methods

We require information from authors about some types of materials, experimental systems and methods used in many studies. Here, indicate whether each material, system or method listed is relevant to your study. If you are not sure if a list item applies to your research, read the appropriate section before selecting a response.

## Materials & experimental systems

| n/a | Involved in the study |
|-----|------------------------|
| ☒ ☐ | Antibodies |
| ☒ ☐ | Eukaryotic cell lines |
| ☒ ☐ | Palaeontology and archaeology |
| ☒ ☐ | Animals and other organisms |
| ☒ ☐ | Clinical data |
| ☒ ☐ | Dual use research of concern |
| ☒ ☐ | Plants |

## Methods

| n/a | Involved in the study |
|-----|------------------------|
| ☒ ☐ | ChIP-seq |
| ☒ ☐ | Flow cytometry |
| ☒ ☐ | MRI-based neuroimaging |

## Plants

Seed stocks | *Report on the source of all seed stocks or other plant material used. If applicable, state the seed stock centre and catalogue number. If plant specimens were collected from the field, describe the collection location, date and sampling procedures.*

Novel plant genotypes | *Describe the methods by which all novel plant genotypes were produced. This includes those generated by transgenic approaches, gene editing, chemical/radiation-based mutagenesis and hybridization. For transgenic lines, describe the transformation method, the number of independent lines analyzed and the generation upon which experiments were performed. For gene-edited lines, describe the editor used, the endogenous sequence targeted for editing, the targeting guide RNA sequence (if applicable) and how the editor was applied.*

Authentication | *Describe any authentication procedures for each seed stock used or novel genotype generated. Describe any experiments used to assess the effect of a mutation and, where applicable, how potential secondary effects (e.g. second site T-DNA insertions, mosiacism, off-target gene editing) were examined.*

