## [Peer Review File · Nature Biomedical Engineering]

A robust and scalable framework for hallucination detection in virtual tissue staining and digital pathology

Corresponding Author: Dr Aydogan Ozcan

Version 0:

Decision Letter:

Dear Aydogan,

Thank you again for submitting to *Nature Biomedical Engineering* your manuscript, "Autonomous Quality and Hallucination Assessment for Virtual Tissue Staining and Digital Pathology". The manuscript has been seen by three experts, whose reports you will find at the end of this message.

You will see that the reviewers appreciate the work. However, they express substantial concerns about the potential clinical utility of the approach, its interpretability, the size of the dataset, and the apparently insufficient validation and benchmarking efforts. We hope that with substantial further work you can address the criticisms and convince the reviewers of the merits of the study.

When you are ready to resubmit your manuscript, please upload the revised files, a point-by-point rebuttal to the comments from all reviewers, the [reporting summary](https://www.nature.com/authors/policies/ReportingSummary.pdf), and a cover letter that explains the main improvements included in the revision and responds to any points highlighted in this decision.

Please follow the following recommendations:

- * Clearly highlight any amendments to the text and figures to help the reviewers and editors find and understand the changes (yet keep in mind that excessive marking can hinder readability).
- * If you and your co-authors disagree with a criticism, provide the arguments to the reviewer (optionally, indicate the relevant points in the cover letter).
- * If a criticism or suggestion is not addressed, please indicate so in the rebuttal to the reviewer comments and explain the reason(s).
- * Consider including responses to any criticisms raised by more than one reviewer at the beginning of the rebuttal, in a section addressed to all reviewers.
- * The rebuttal should include the reviewer comments in point-by-point format (please note that we provide all reviewers will the reports as they appear at the end of this message).
- * Provide the rebuttal to the reviewer comments and the cover letter as separate files.

We hope that you will be able to resubmit the manuscript within 12 weeks from the receipt of this message. If this is the case, you will be protected against potential scooping. Otherwise, we will be happy to consider a revised manuscript as long as the significance of the work is not compromised by work published elsewhere or accepted for publication at *Nature Biomedical Engineering*.

We hope that you will find the referee reports helpful when revising the work. Please do not hesitate to contact me should you have any questions.

Best wishes,

Pep

Reviewer #1 (Report for the authors (Required)):

The authors present a novel automated framework called AQuA for quality assessment and detection of hallucinations in virtually stained histology images. The authors have developed a robust data assembly methodology to introduce a wide variety of morphological hallucinations and error modes into the kidney and lung tissue datasets. The reported accuracy is 99.8% in detecting acceptable and unacceptable virtually stained images, along with 98.5% agreement with manual expert assessments. The ability of AQuA to detect both "technical/unrealistic" and "realistic" hallucinations is a unique feature. AQuA outperforms human experts in identifying realistic-looking hallucinations, demonstrating the potential of this tool for clinical applications.

In my view, this is an important development, as virtual staining techniques have limitations in terms of potential artifacts and hallucinations, which can significantly impact clinical decision-making. The training, validation, and testing of the AQuA-Net using virtual staining data, as well as the comparison to manual expert assessments, are well-executed. The reported accuracy without the involvement of ground truth (i.e., unsupervised learning) is also highly satisfactory. The authors have evaluated the adaptability and generalization of AQuA across various virtually and histochemically stained human tissue images, which strengthens the claims of the framework's versatility. The development of AQuA has the potential to enhance the reliability and clinical utility of virtual staining techniques, which is crucial for the widespread adoption of these methods in digital pathology.

The results look promising with a clear goal. However, there are some major concerns in the manuscript that should be addressed. I would encourage the authors to revise their manuscript based on the following comments:

1. Many of the quality assessments and detection of hallucinations in this work are based on the images that are generated with poor-staining models. However, those poor-staining models are being created with not enough number of epochs or with sub-optimal validation loss. That means the poor-staining models do not exist naturally, unlike the artifacts in histochemical staining. The authors may have to justify more on when will we have the poor-staining models in practice as we always choose the lowest validation loss values.
2. Apart from being an automated framework, the authors should also mention how much time is saved for their proposed framework. For example, in Fig. 1 (a), the authors can also compare the time for the two workflows so as to further show the strength of their AQuA.
3. The authors should provide a more in-depth analysis of why AQuA is superior to pathologists, especially in distinguishing realistic hallucinations. I would suggest the authors look into data similar to Extended Data Fig. 1 (a) for more details. For example, what are the features in the images for the lung/kidney samples that make pathologists believe that the images are properly stained or what are the features that make AQuA able to catch those cases are realistic hallucinations? A follow-up interesting investigation is if the pathologists feel that it is ready for diagnosis, whether those features are important or not for diagnosis.
4. For Fig. 6(d), why the red curve can exceed 100% when M is 10?
5. I am afraid that equation (17) is incorrect. The denominators of sensitivity and specificity are not all negatives and positives, respectively. According to the equation (17), it should be PPV and NPV instead. Furthermore, the Specificity is also spelled wrongly in the equation. The authors should double check.
6. Following what the authors mentioned in the second last paragraph of the discussion, the lung sample is composed of red blood cells (RBCs) and scattered lymphocytes, which are under-representation in the training data, which causes AQuA accepting that image in Extended Data Fig. 1b. The authors should further address the data imbalance issues' methodology as this is one of the common problems in deep learning, which already shows as a problem in the presented case.
7. Given the number of patients with limited amounts, how confident are the authors in supporting their claims on strong external generalization? This is also related to question 6.

Although the manuscript is pretty well written, there are still some writing issues. For example,

1. Once the acronym is defined, the authors should use it throughout. For example, histochemical stain/staining (HS) is defined in the introduction. However, I do see many other places using histochemical stain/staining and HS interchangeably. It is also the case for autofluorescence vs AF. Please double check.

Reviewer #2 (Report for the authors (Required)):

The manuscript presents an autonomous quality and hallucination assessment method (AQuA) for virtual tissue staining. Specifically, AQuA performs T successive virtual staining-autofluorescence cycles to obtain pairs of autofluorescence images and H&E stained images. Then, the generated image pairs are fed into AQuA-Net to classify whether the input virtual stain is acceptable or unacceptable. The proposed framework presents a very high agreement (98.5%) with the manual assessments of pathologists. While the topic is novel and the initial premise promising, there are several fundamental concerns.

Major:

1. **Clinical significance:** It's crucial to acknowledge that while virtual staining is advancing rapidly, its validation and acceptance in clinical settings are still controversial. Consequently, the assessment of virtual stains remains of limited clinical significance.
2. **Insufficient interpretability:** Highlighting areas that contribute to the prediction of classification models is crucial to understanding the decision-making process. However, the proposed framework only provides binary classification results instead of heatmap within the images, hence the potential clinical application is constrained.
3. **The construction of a virtual stain dataset:** AQuA-Net is trained with a virtual stain dataset generated using a set of poor-staining and good-staining models. The label of a virtual stain is determined by the model instead of the hue within the stain. Note that even a good-staining model may produce some unacceptable stains, and vice versa, a poor-staining model may generate acceptable stains. The authors assume that all stains from a good-staining model are acceptable, yet theoretically, it may lead to degraded quality of the dataset.
4. **Size of dataset:** The dataset comprises images from merely 10 patients for kidney tissue and images from 29 patients for lung tissue, which is notably tiny dataset considering the complexity and heterogeneity of human pathology. This small-scale sample may hinder the generalizability and robustness of the developed framework, not to mention the wide application.
5. **The performance of the virtual stain model:** In Fig. 4(a), examples of negative and positive virtual stains are compared with the corresponding histochemically stained H&E image. It's evident that both virtual stains fail to present nuclei well, as indicated by the region to the left of the black arrow.

Minor:

1. **The model selection:** The framework relies solely on Generative Adversarial Networks (GANs), and no experiments are conducted with other generative models, e.g. diffusion models. This lack of experimentation with alternative models raises questions about the comprehensiveness of the approach. Also, the approach to evaluate the quality of virtual stains from a different generative model is absent.
2. **Iterations between different domains:** Several iterations between the H&E and AF domains are performed to generate image pairs as the input for AQuA-Net. It is of high risk that one model is not properly trained, then the whole pipeline fails.
3. **The definition of good-staining models and poor-staining models:** Good-staining and poor-staining models are defined based on a simple threshold applied to the number of epochs and loss value. However, this threshold-based approach lacks explanatory details and context.

Missing details:

1. **Details of dataset.** The article lacks precise descriptions of the dataset used in the study. The specific types of diseases and the number of WSIs were missing.
2. **How to perform majority voting.** The article should elaborate how majority voting is conducted within the framework. For example, how were the different heads trained?

Suggestions for improvement:

1. **Addressing artifacts in chemical staining:** In addition to tackling the issue of virtual staining artifacts, it is important to focus on addressing natural artifacts in chemical staining, such as tissue folding or uneven staining. Excluding both virtual and chemical staining artifacts will improve the overall reliability and accuracy of the staining process.
2. **Explainability:** To prove the effectiveness of rejecting a certain virtual stain, it is essential to visualize the regions of interest that the network focuses on.

Reviewer #3 (Report for the authors (Required)):

In this study, the authors present an autonomous quality and hallucination assessment method to validate virtual stained images generated by generative models. While the topic is of interest, several issues would need substantial attention.

1. In general, the clinical use of the proposed method is unclear. Once a section is prepared, the physical staining can be

performed very fast and highly cost-efficiently. Thereby, the exact rationale for virtually staining the sections of unstained slides, particularly since these need to be scanned using fluorescent microscopes, is not completely clear. In which setting this should be used? What would be the added computational requirements and time needed to run the model compared to the physical staining?

2. It is not possible for humans to accurately find inconsistencies and differences between the few shown exemplary virtually and physically stained slides. How could it be excluded that the "hallucinations" that are not detected by the proposed model wouldn't change other diagnoses that were not tested?

3. To validate the model, it is important to look at quantifiable metrics between the virtually and physically stained sections. Unfortunately, the authors didn't use such hand-crafted features to validate their model. As a simple example, this could allow checking if the number and shape of nuclei inside different structures are consistent. E.g., does Glomerulus number 1 in virtual stain have the same number of nuclei as the Glomerulus number 1 in the physically stained section? This would be particularly interesting since the authors showed some of the artifacts as being fake or missing nuclei. This would be diagnostically relevant. Numerous such features would be relevant.

4. The authors calculated metrics such as MSE, PCC, or PSNR for quality assessment between the statistical approaches and the virtually stained slides. The results are, unsurprisingly, in favor of the proposed model. This might be because these metrics are too simple to capture the hallucinations. The authors should compare their model to a supervised model, e.g., such as TransMIL.

5. There are many methods available to detect tissue artifacts, the most prominent probably being HistoQC. A benchmark against such available methods/tools is missing.

6. The authors mention attack strategies. Please discuss or evaluate how integrating AQUA into the virtual staining model, e.g., as part of the discriminator that needs to be fooled, would change the study results. Please also evaluate in a classification setting whether AQUA can be used to detect malicious changes in images (adversarial attacks).

Minor points:

1. Several parts of the manuscript are repetitive and could be shortened. Most of the methods section is rather common knowledge and could be moved to the appendix section.

2. Figures 3 and 5 - the second rows of both figures are redundant and add no extra information.

3. Some text sections are printed bold, please omit such modifications.

Version 1:

Decision Letter:

Dear Aydogan,

Thank you for your revised manuscript, "Autonomous Quality and Hallucination Assessment for Virtual Tissue Staining and Digital Pathology", which has been seen by the original reviewers. In their reports, which you will find at the end of this message, you will see that the reviewers acknowledge the improvements to the work and that they raise recurrent and additional technical criticisms that I am hoping you will be able to address.

As before, when you are ready to resubmit your manuscript, please upload the revised files, a point-by-point rebuttal to the comments from all reviewers, and the [reporting summary](https://www.nature.com/authors/policies/ReportingSummary.pdf).

As a reminder, please follow the following recommendations:

* Clearly highlight any amendments to the text and figures to help the reviewers and editors find and understand the changes (yet keep in mind that excessive marking can hinder readability).

* If you and your co-authors disagree with a criticism, provide the arguments to the reviewer (optionally, indicate the relevant points in the cover letter).

* If a criticism or suggestion is not addressed, please indicate so in the rebuttal to the reviewer comments and explain the reason(s).

* Consider including responses to any criticisms raised by more than one reviewer at the beginning of the rebuttal, in a section addressed to all reviewers.

* The rebuttal should include the reviewer comments in point-by-point format (please note that we provide all reviewers will the reports as they appear at the end of this message).

* Provide the rebuttal to the reviewer comments and the cover letter as separate files.

We look forward to receive a further revised version of the work. Please do not hesitate to contact me should you have any questions.

Best wishes,

Pep

Pep Pàmies

Chief Editor, Nature Biomedical Engineering

Reviewer #1 (Report for the authors (Required)):

I am glad to see the improvement of the manuscript based on the comments from the last round. The revision is of high quality. Most of the comments have been appropriately addressed by the authors.

The only outstanding question I still have is related to comment 3 from the last round, which was not fully addressed. Although I understand the focus of the work is on the quality check of the VS or HS images, it is still meaningful to investigate whether the hallucinated images cause any wrong diagnoses. I suggest the authors perform one additional quality assessment with the board-certified pathologists, confirming whether they will make the correct/incorrect diagnostic outcome based on the hallucinated images (with respect to the HS ground truth). If the hallucinated images cause wrong diagnoses by board-certified pathologists, which can be detected by the AQuA, the value of the work will be further strengthened.

Reviewer #2 (Report for the authors (Required)):

Here are comments for the revised manuscript:

1. Generally, the evaluation dataset is quite limited. The inclusion of an external validation set, such as images from TCGA, would be beneficial to validate the generalizability of the proposed framework.
2. More experiments are needed on the selection of thresholds used to differentiate between good-stain and bad-stain models. It's worth investigating the impact of different thresholds on classification accuracy.
3. The manuscript lacks detailed explanation on how histochemically stained images are classified as 'positive' or 'bad'.
4. In the task of histochemical stain (HS) evaluation, additional comparisons between the use of ResNet and Transformer would be beneficial.
5. Given the varying significance of different iterations, a straightforward average of all heatmaps (from various iterations) could potentially generate inaccurate attention maps.
6. The manuscript could provide more detailed information about the number of positive and negative stains in the test sets for the evaluation of VS and HS.
7. Generalization and Robustness with Limited Data. The issue of generalization and robustness due dataset has not been adequately addressed. Experiments conducted with such a small sample size (not pixel count) make the test results for kidneys and lungs less meaningful for assessing robustness and generalization of the method. While transfer-based validation suggests some level of generalization, overfitting is also a potential concern due to the limited amount of data.
8. Human Visual Inspection in H&E and AF Domain Iterations. The requirement for human visual inspection of models in each iteration between the H&E and AF domains raises questions about the stability of training. Does this imply that training stability is prone to issues, or does it serve as a form of supervisory information in some regard?
9. Explanation of Good vs. Poor Staining Models. The explanation of good and poor staining models only describes how specific models are selected, without addressing why these choices are made or why they are appropriate. More detailed justification for the selection criteria and their correctness is needed.
10. Transfer of Staining Techniques Using AQuA-Net. AQuA-Net iterates images between the H&E and AF domains. Is it possible to apply staining transfers using this method? Images with different staining styles should map consistently to the AF domain. Since there is substantial validation data available for staining normalization issues, this could be a way to

enhance the clinical application of the method.

11. Virtual stain may miss or mis-stain some diagnostic important information, i.e., nuclei. The authors should quantify the results on this essential metric.

12. There is still issue with the evaluation method, "each pathologist was asked to give a binary flag on the overall stain quality of each VS image, indicating whether the image has acceptable or unacceptable stain quality." The pathologists cannot judge the overall stain quality. They can hardly tell if only 8 out of 10 nuclei are properly stained, or real stain is unnecessary. Careful comparison between virtual stained and real HE stained on large-scale histology images by pathologists is almost impossible unless the dataset is small.

Reviewer #3 (Report for the authors (Required)):

Thank you for addressing several of my previous issues raised. Still, some ambiguity remains that would need further clarification:

1) Even if the study is designed to only target kidney and lung histology, it is essential to demonstrate the diagnosis of the cases. The authors need to provide data about the WSIs that were used to train and test the model and their respective diagnoses to see the generalizability of the AQUA results in different disease geometries. It is also important to describe how the patches were selected, were these healthy, diseased, etc.

2) Thinking about the pathology workflow – if the diagnosis for a slide is "disease X" on the chemically stained tissue, does this align with the pathologist's diagnosis made on the virtual stain? This would be more relevant on the slide level. It is also interesting if the artificial stains from the so-called "low-quality artificial stain generator" output would also have the same diagnosis.

3) If according to AQUA a slide includes patches with hallucinations, how do these hallucinations affect the diagnosis of a pathologist from the artificial stain?

4) How can the information from image patch labels received from AQUA be transferred to the slide label? If the recommended approach is to exclude the patches that involve hallucinations, it could make the diagnosis impossible or inaccurate (i.e., what if the part that is essential for the diagnosis of the WSI lies in the patches that AQUA suggests excluding?)

How would such limitation affect AQUA and virtual stains in routine pathology?

5) The patch that is used for grad cam is heavily background dominant, please add more images, especially high-resolution images with full tissue.

6) Regarding the comparison of TransMIL vs AQUA, please provide more confusion matrices to show in which types of hallucinations AQUA outperforms TransMIL.

7) Regarding the labeling of the slide, TransMIL can have the WSI as one bag and therefore identify if the WSI is acceptable or not. If AQUA would be able to provide a method for a decision on slide level, would it still outperform TransMIL in giving a label (good quality or bad quality stain) to the WSI?

Minor:

1) Names of figures are missing from Extended Data Figure 6 where are parts e, f, etc?

2) The workflow depends on fluorescent scanning of the unstained tissue, this needs to be considered when judging the overall suitability and efficacy.

Version 2:

Decision Letter:

Dear Aydogan,

Thank you for your revised manuscript, "Autonomous Quality and Hallucination Assessment for Virtual Tissue Staining and Digital Pathology", which has been seen by the original reviewers. In their reports, which you will find at the end of this message, you will see that the reviewers acknowledge the improvements to the work and that they raise a few additional technical criticisms that I am hoping you will be able to suitably address.

As before, when you are ready to resubmit your manuscript, please upload the revised files, a point-by-point rebuttal to the comments from all reviewers, and the [reporting summary](https://www.nature.com/authors/policies/ReportingSummary.pdf).

We look forward to receive a further revised version of the work. Please do not hesitate to contact me should you have any questions.

Best wishes,

Pep

Pep Pàmies

Chief Editor, Nature Biomedical Engineering

Reviewer #1 (Report for the authors (Required)):

I am glad to see the authors have improved their manuscript further by taking my suggestions and showing the comparison of images from the poor-staining VS model and the good-staining VS model with the ground truth (newly added Extended Data Figure 12). The results show that the AQuA workflow can differentiate the quality of the virtual staining models, further showing the value of the workflow.

However, this detailed comparison shows that even the good VS model is not as good by comparing Fig. 12(g) and (h). The good VS model still misses lots of cancer cells/lymphocytes (cannot be confirmed with the current magnification). Therefore, additional effort has to be made in the virtual staining algorithm to minimize this digital staining accuracy. This additional study shows the weakness of the current work (even the good ones identified by AQuA are not good enough). The authors should comment more on the difference between Fig. 12(g) and (h), and how to improve the virtual staining performance.

Reviewer #2 (Report for the authors (Required)):

Most comments have been appropriately addressed. The remaining issue is just clarity of certain sections, for example, the following ones.

Human visual inspection in Methods section:

'The optimal VS generator network models for kidney and lung samples, utilized to facilitate the cycle of AF-H&E transformations within the AQuA model, were chosen based on the lowest validation loss values and evaluations conducted through human visual inspection.'

'2. Iterations between different domains: Several iterations between the H&E and AF domains are performed to generate image pairs as the input for AQuA-Net. It is of high risk that one model is not properly trained, then the whole pipeline fails.'

Reviewer #3 (Report for the authors (Required)):

Thanks for addressing most of the issues. There are only few points remaining.

One aspect concerns the newly added Extended Data Figure 12 and paragraph, that addressed my previous points 2 and 3, regarding the effect of hallucinations on diagnosis.

- Would it be possible for the authors to add the grad cam visualization of the ROIs for the samples that are illustrated in the Extended Data Fig. 12? The authors stated: "the second poor-staining VS model introduced realistic hallucinations by missing many lymphocyte nuclei". It would be beneficial to show that AQUA is rejecting patches due to the hallucination of lymphocyte nuclei.

- Could the authors provide the output of TransMil on these patches to show whether TransMil also identifies such hallucinations?

- Finally, to comprehensively show AQUA's capabilities, adding a similar case study on hallucinations that can alter the diagnosis in a good VS model and can be detected by AQUA would also be interesting.

Regarding the newly added study on the TCGA data, it would be interesting to compare the results with the output of HistoQC. Reporting the accuracy without transfer learning would also be helpful since the samples were HE.

Would suggest to either support the following statement with some experiments or leave it out: "Additionally, the AQuA framework can be extended to some of the recent advances in stain-to-stain transformations and stain normalization methods^{25,59–61}. Through iterations between image domains of different stains, or between image domains of AF and

multiple types of stains, AQuA can potentially provide autonomous quality and hallucination assessment for virtually transformed and normalized pathology images, which is especially important for cases where HS references are inaccessible."

Version 3:

Decision Letter:

Dear Aydogan,

Thank you for your revised manuscript, "Autonomous Quality and Hallucination Assessment for Virtual Tissue Staining and Digital Pathology". Having consulted with Reviewers #1 and #3 (whose comments you will find at the end of this message), I am pleased to write that we shall be happy to publish the manuscript in *Nature Biomedical Engineering*.

We will be performing detailed checks on your manuscript, and in due course will send you a checklist detailing our editorial and formatting requirements. You will need to follow these instructions before you upload the final manuscript files.

Best wishes,

Pep

Pep Pàmies

Chief Editor, *Nature Biomedical Engineering*

Reviewer #1 (Report for the authors (Required)):

The authors have appropriately addressed my concerns. I have no further questions regarding the manuscript. I support the acceptance of the manuscript for publication in *Nature Biomedical Engineering*.

Reviewer #3 (Report for the authors (Required)):

Thanks for addressing all the remain issues!

Version 4:

Decision Letter:

Dear Dr Ozcan,

I am happy to inform you that your manuscript, "A robust and scalable framework for hallucination detection in virtual tissue staining and digital pathology", has now been accepted for publication in *Nature Biomedical Engineering*.

Over the next few weeks, the figures will be checked for production quality, the text edited to ensure that it conforms to house style, and the manuscript typeset.

Our Articles are published about 40 days after the acceptance date (we recommend that you inform your institutional press office of this timeframe), and you will be notified of the actual publication date a few days in advance. Articles can be published any working day of the week, and are pushed live shortly after 10 am London time.

Publishing agreement. You will be asked to digitally sign a publishing agreement (grant of rights). After the signed publishing agreement has been received, the proofs of the article will be sent to you for review. If you have any queries during the production process, or you cannot meet the requested deadline for returning the proofs, please contact rjsproduction@springernature.com.

Nature Biomedical Engineering is a Transformative Journal. Authors may publish their research with us through the

traditional subscription access route, or make their paper immediately open access through payment of an article-processing charge. More [information about publication options](https://www.springernature.com/gp/open-research/transformative-journals) is available.

You may need to take specific actions to [comply](https://www.springernature.com/gp/open-research/funding/policy-compliance-faqs) with funder and institutional open-access mandates. If the work described in the accepted manuscript is supported by a funder that requires immediate open access (as outlined, for example, by [Plan S](https://www.springernature.com/gp/open-research/plan-s-compliance)) and your manuscript was originally submitted on or after January 1st 2021, then you should select the gold OA route. Authors selecting subscription publication will need to accept our standard licensing terms (including our [self-archiving policies](https://www.springernature.com/gp/open-research/policies/journal-policies)), and these will supersede any other terms that the author or any third party may assert apply to any version of the manuscript.

Acceptance of your manuscript is conditional on agreement, by all authors, with both our [media embargo](http://www.nature.com/authors/policies/embargo.html) and [confidentiality and pre-publicity](http://www.nature.com/authors/policies/confidentiality.html) policies. In particular, you may arrange your own publicity of the Article (for instance, through your institutional press office), as long as you ensure that journalists strictly adhere to the media embargo.

To assist you in disseminating the work, as soon as the Article is published you will be able to take advantage of the Springer Nature [SharedIt](https://www.springernature.com/gp/researchers/sharedit) initiative to [generate a unique shareable link to the Article](http://authors.springernature.com/share) that will allow anyone (with or without a subscription) to read it. Recipients of the link who are subscribers will also be able to download and print the PDF.

Thank you for having submitted this work to *Nature Biomedical Engineering*.

Best wishes,

Barbara Cheifet
Editor
Nature Biomedical Engineering

As detailed in this Response Letter, we have added substantial additional data and results/analyses to our revised manuscript in response to the referees' questions/comments, which have significantly improved the quality and clarity of our manuscript.

We briefly summarize below the main additions made in the revised manuscript, following the reviewers' feedback. We then address, point by point, the specific comments that were raised by our reviewers. Throughout this letter, the original referee comments are shown in black color, whereas for ease of communication, our answers are provided in blue. Our revisions have also been marked in the main text and supplementary information files using yellow highlighting.

Summary of the revisions

- Detailed assessment and comparisons between VS (virtually stained) images generated by the optimal VS model shown in this study and the corresponding HS (histochemical) references were added to our revision. Multiple quantitative evaluation metrics are employed to support the advantages and clinical utility of VS technologies over traditional HS workflow without sacrificing the staining quality. The following new results and analyses related to this topic are added to our revised manuscript:
 - Revision to the Methods sub-section: “Network architecture and training schedule of VS networks”
 - **New Methods sub-section:** “Quantitative evaluation of the virtual staining model used in VS-AF iterations”
 - Revision to Figure 1: “Workflow of HS and VS of tissue and corresponding quality assessment”
 - **New Supplementary Note 2:** “Comparison between VS and HS quality”
 - **New Supplementary Figure 1:** “Color distribution comparisons of VS and HS kidney tissue images”
 - **New Supplementary Figure 2:** “PSNR, SSIM and structural feature comparisons of VS and HS kidney tissue images”
- New results of AQuA-Net inference and detailed analysis of the contributing factors to its superior performance. Leveraging the Grad-CAM visualization technique, we showcase that the novel VS-AF iterations in AQuA-Net architecture effectively utilize spatial-temporal image features and very well capture abnormalities and inconsistencies in VS images without any HS reference image. Board-certified pathologists further provided comments from clinical and diagnostic perspectives on realistic-hallucinatory VS images when HS references were provided and compared side-by-side to better understand VS hallucination and the advantages of AQuA. Regarding this new experiment and analysis, we have added the following new figures and sections to our revised manuscript:
 - Revision to Results sub-section: “Image-level autonomous quality and hallucination assessment of lung tissue virtual staining”
 - Revision to the Discussion section

- **New Extended Data Figure 3:** “Examples of realistic hallucinations generated by VS on human lung tissue samples, in comparison with the corresponding HS reference images.”
 - **New Extended Data Figure 4:** “Examples of poorly stained HS slides”
 - **New Supplementary Note 1:** “Grad-CAM visualization analysis”
 - **New Extended Data Figure 6:** “Spatial-temporal Grad-CAM visualization of the activation maps during the AQuA inference on poor VS images of human kidney sample.”
 - **New Methods sub-section:** “Grad-CAM visualization”
- On top of the generalization tests on unseen VS model failure modes reported in our original manuscript, we extended our tests to evaluate AQuA’s performance on unseen tissue types. When blindly tested on unseen types of organs, AQuA is able to reject poor VS images, achieving a high sensitivity; and after transfer learning, it achieves equivalent performance to an organ-specific model trained from scratch. This new experiment and related analysis have been summarized into the following new figures and sections in our revised manuscript:
 - **New paragraph in the Discussion section:** Generalization of AQuA on unseen tissue types
 - **New Extended Data Figure 10:** “External generalization of AQuA on unseen types of tissue samples”
 - Revision to Methods sub-section: “Architecture and training schedule of AQuA-Net”
- **Extensive benchmarking results and analyses** against some of the state-of-the-art competitive approaches are implemented, including comparison with an open-source stain quality control tool HistoQC and a transformer-based network TransMIL. These comparisons strongly support the effectiveness and efficiency of AQuA, which consistently demonstrates better classification accuracy than the existing competitive approaches. These new results and discussion have been summarized in the following new figures and sub-sections:
 - **New paragraph in the Discussion section:** Comparison between AQuA and other competitive methods
 - **New Extended Data Figure 8:** “Classification performance comparison between (a) AQuA and (b) TransMIL on detecting good and bad HS images of human lung tissue samples”
 - **New Extended Data Figure 9:** “Quality control detection results from the HistoQC model and the AQuA model”

- **New Methods sub-sections:** “Implementation of TransMIL” and “Comparison between HistoQC and AQuA”
- In addition to these, we have also added the following new figure and discussion to our revised manuscript:
 - **New Extended Data Figure 12:** “Thresholding and selection of the VS network model checkpoints”
 - **New paragraph in the Discussion section:** potential applications of AQuA framework on protecting against adversarial attacks and improving network robustness

In summary, the following new figures, new supplementary notes and new sub-sections are added to our revised manuscript:

New Figures Added

- **New Extended Data Figure 3:** Examples of realistic hallucinations generated by VS on human lung tissue samples, in comparison with the corresponding HS reference images. (a) Cellular hallucinations are observed in several locations (green arrows), mimicking atypical mitoses in the VS slide (left). HS slide demonstrates normal appearing cells. (b) Myxoid change in the VS slide (green asterisks) can raise a differential diagnosis of several benign and malignant lesions. HS slide demonstrates normal appearing stroma. (c) Abundance of anthracotic pigment in alveolar macrophages (green arrows) in the VS slide, suggesting potential smoking exposure. HS slide demonstrates a minimal amount of pigment. Scale bar: 20 μ m.
- **New Extended Data Figure 4:** Examples of poorly stained HS slides. (a) Complete lack of nuclear staining; (b) out of focus, blurry regions (black asterisks); (c) lack of nuclear staining and artefactual greenish pigment (black arrows); (d) appearance of anthracotic-like black pigment in abnormal shape and location (black arrows). Scale bar: 50 μ m.
- **New Extended Data Figure 6:** Spatial-temporal Grad-CAM visualization of the activation maps during the AQuA inference on poor VS images of human kidney sample. (a) AF and VAF images in the VS-AF iterations, and the HS reference image (ground truth). (b) VS images in the VS-AF iterations, and the difference map between the VS($t=0$) and the HS reference image is smoothed by a Gaussian kernel k with $\sigma = 15.5$ px to match the resolution of Grad-CAM. \otimes stands for 2D convolution. (c) Grad-CAM heatmaps for VS(t) and the averaged heatmap. $\langle \cdot \rangle$ stands for averaging over time. PCC values were calculated between the difference map and the averaged heatmap. (d) A broken tissue area after virtual staining was successfully identified by AQuA without using an HS reference image. (e) A hollow area with strong AF background noise, marked by AQuA. (f) A tiny residual tissue not cleared in sample preparation was successfully detected by AQuA. Scale bar: 50 μ m.

- New Extended Data Figure 8:** Classification performance comparison between (a) AQuA and (b) TransMIL on detecting good and bad HS images of human lung tissue samples. Both models utilized the best 5 out of 10 ensembles and 100% sensitivity threshold (α) acquired on the validation set. Box plot, 25th-75th percentiles; center, median; whisker, 16th-84th percentiles. See the Methods section for details on TransMIL implementation.
- New Extended Data Figure 9:** Quality control detection results from the HistoQC model and the AQuA model. (a) The WSI of a poorly stained HS slide labelled by pathologists. (b) The detection results of (a) using the HistoQC model, where pink regions indicate high-quality stained areas accepted by the model, and green regions correspond to poorly stained areas rejected by the quality control model. (c) The acceptance probability map of (a) predicted by the AQuA model, where values close to 1 indicate acceptance, while values near 0 signify rejection. (d) The detection results of (a) using the AQuA model, which was obtained by thresholding (c) by 0.5. (e) One example zoomed-in tissue region rejected by both the HistoQC model and AQuA model, which contains red blood cells effacing the lung architecture. (f) Another example zoomed-in tissue region rejected by both the HistoQC model and AQuA model, which contains blurred out-of-focus areas. (g-h) Two example zoomed-in regions with insufficient/missing hematoxylin staining, which were accepted by the HistoQC model but rejected by the AQuA model. (i) The WSI of a well stained HS slide labelled by pathologists. (j) The detection results of (i) using the HistoQC model. (k) The acceptance probability map of (i) predicted by the AQuA model. (l) The detection results of (i) using the AQuA model, which was obtained by thresholding (k) by 0.5. (m) One example zoomed-in tissue region accepted by both the HistoQC model and AQuA model, which corresponds to well-stained alveolated lung tissue with scattered macrophage. (n) Another example zoomed-in tissue region accepted by both the HistoQC model and AQuA model, which corresponds to well-stained bronchiole lining with ciliated cells. (o-p) Two example zoomed-in regions with good staining quality which were rejected by the HistoQC model but accepted by the AQuA model.
- New Extended Data Figure 10:** External generalization of AQuA on unseen types of tissue samples. Classification performance of (a) AQuA trained on human kidney samples and tested on human lung samples to test external generalization; (b) AQuA trained on human lung samples and tested on human kidney samples to test external generalization; (c) AQuA trained on human kidney samples, then transfer-learned and tested on human lung samples; (d) AQuA trained on human lung samples, then transfer-learned and tested on human kidney samples. Threshold α was selected as the 100% sensitivity threshold on the validation set. Box plot, 25th-75th percentiles; center, median; whisker, 16th-84th percentiles. Refer to the Methods section for transfer learning details.

- **New Extended Data Figure 12:** Thresholding and selection of the VS network model checkpoints. The gray zones exclude checkpoints with ambiguity so that the selected checkpoints are well separated in both the validation loss and training epoch number. Refer to the Methods section for more details on thresholding. The legends denote three types of VS checkpoints selected for the training, validation and testing of AQuA.
- **New Supplementary Figure 1:** Color distribution comparisons of VS and HS kidney tissue images.
- **New Supplementary Figure 2:** PSNR, SSIM and structural feature comparisons of VS and HS kidney tissue images.

New Supplementary Notes Added

- **New Supplementary Note 1:** Grad-CAM visualization analysis
- **New Supplementary Note 2:** Comparison between VS and HS quality

New Sub-sections Added

- **New Discussion 3rd paragraph:** Time savings of AQuA compared to traditional HS workflow, on page 12
- **New Discussion 5th paragraph:** Comparison between AQuA and competitive methods, on page 12-13
- **New Discussion 6th paragraph:** Generalization of AQuA on unseen sample types, on page 13
- **New Discussion 8th paragraph:** Potential applications of AQuA framework on protecting against adversarial attacks and improving network robustness, on page 14
- **New Methods sub-section:** Implementation of TransMIL
- **New Methods sub-section:** Comparison between HistoQC and AQuA
- **New Methods sub-section:** Majority voting
- **New Methods sub-section:** Grad-CAM visualization
- **New Methods sub-section:** Quantitative evaluation of the virtual staining model used in VS-AF iterations

Reviewer #1

1. Many of the quality assessments and detection of hallucinations in this work are based on the images that are generated with poor-staining models. However, those poor-staining models are being created with not enough number of epochs or with sub-optimal validation loss. That means the poor-staining models do not exist naturally, unlike the artifacts in histochemical staining. The authors may have to justify more on when will we have the poor-staining models in practice as we always choose the lowest validation loss values.

We thank the reviewer for raising this important concern. In addition to detecting low-quality VS images generated by poor-staining models, **we demonstrated that the same AQuA model can detect the generalization failures of a well-trained, overfitted VS model.** This generalization failure is common in practice as the differences in individual subjects, sample preparation protocols and imaging parameters may result in distribution shifts at the test time. This part of the results is illustrated in the **Extended Data Fig. 2**, analyzed in the Results section, quoted below:

“Furthermore, AQuA is effective in detecting various failure patterns of VS models, including suboptimal convergence and generalization failure. In addition to testing our AQuA model on hallucinated images from early stopped checkpoints, we conducted an external generalization test to an overfitted VS model using the same classifier described above, which has never seen such overfitted VS models. As shown in Extended Data Fig. 2a, the overfitted VS model, well trained on a small subset of the training data, fails to generalize on testing data and exhibits a distinct staining failure pattern compared with the early stopped VS models, such as missing a large portion of nuclei and generating blurry features. Despite the difficulty of detecting these staining failure patterns never seen before during the classifier training, AQuA successfully discriminated all VS images from the overfitted model and the early stopped model as illustrated in Extended Data Fig. 2b. This generalization test further demonstrates the robustness and generalizability of AQuA to various unseen, unknown staining failure patterns of VS models.” (Results section, **Image-level autonomous quality and hallucination assessment of kidney tissue virtual staining**, 4th paragraph)

In the revised manuscript, we have further extended our testing of AQuA’s generalization to unseen types of organs. In these new experiments, both the VS model and AQuA were trained on one specific type of organ and then blindly tested on another type of organ never seen before. Due to the distribution shifts, the VS model could fail to generalize and as a result produce low-quality VS images, while AQuA generalizes and rejects poor VS images, scoring a high sensitivity. We have summarized these new experiments in the new **Extended Data Figure 10** and revised the Discussion section, quoted below:

“Furthermore, AQuA exhibits robust external generalization to various unseen distributions during the blind testing stage. Extended Data Figure 2 demonstrates AQuA’s strong external generalization to unseen VS model failure patterns, where overfitted VS models are successfully

*detected by AQuA, which was trained only on failure modes of early stopped models. Additionally, AQuA's robustness can be further extended to generalization on unseen tissue types. We blindly tested two AQuA models trained with human kidney (and lung) samples on VS images of human lung (and kidney) samples, respectively. Despite the distinct tissue structures of these two organs and different imaging parameters, including varying signal-to-noise ratio, AQuA successfully generalized and detected poor VS images with a sensitivity of 98.9%, as reported in **Extended Data Fig. 10(a, b)**. This strong generalization of AQuA effectively prevents low-quality VS images from a shifted data distribution passing through a pre-trained AQuA model. In addition to these, we also used transfer learning on each new tissue type to show that AQuA could achieve further improved inference performance, comparable to an AQuA model trained from scratch solely on the testing tissue type; see **Extended Data Fig. 10(c, d)**. This robustness of AQuA to new tissue types exemplifies its strong generalization, helping to avoid false negatives due to a shifted data distribution.” (Discussion section, 6th paragraph)*

Please also refer to **Extended Data Figure 10** for these detailed generalization test results.

In addition to these, our manuscript also demonstrated that AQuA is very effective at detecting poorly stained HS (histochemical) slides, better than other state-of-the-art approaches.

To better showcase the advantages of spatial-temporal information that AQuA harnesses, we compared AQuA against a well-known transformer-based model **TransMIL**, which was designed to classify HS tissue slides. This new comparison is summarized in the newly added **Extended Data Fig. 8** and analyzed in the revised Discussion section, quoted:

*“...When comparing AQuA with TransMIL, both of the models shared the same pre-trained ResNet-50 backbone⁴⁹ for feature extraction, and were trained on the same dataset of human lung HS images employing the same majority voting mechanism ($C = 5$). **Extended Data Figure 8** summarizes the comparison of these two models on the same testing set of human lung HS images and reports the corresponding confusion matrices; **the results show that TransMIL partially failed to separate the positive and negative populations with a low accuracy of 68.2%, whereas AQuA provides clear separation between the two populations and scored 100% accuracy. Additionally, the total number of trainable parameters of AQuA (~2.1 M) is less than TransMIL (~3.2 M), showcasing AQuA's efficiency...**” (Discussion section, 5th paragraph)*

In the revised manuscript, we have also compared AQuA against a state-of-the-art quality control tool named **HistoQC** designed for digital pathology slides. To showcase this comparative analysis, we added **Extended Data Fig. 9** to our revised manuscript and compared the quality control detection results from the HistoQC model and the AQuA model. **The related analysis confirmed the superior performance of AQuA for detecting poorly stained HS images**, and it is reported in the revised Discussion section, quoted:

*“...In addition to these, when comparing AQuA with HistoQC, AQuA demonstrated a more robust concordance with pathologists' judgments in assessing the quality of the slide staining. **Extended Data Figure 9** illustrates the quality control detection results from both the AQuA and HistoQC models for a poorly stained HS slide and a well-stained HS slide, as labeled by pathologists. As depicted in **Extended Data Fig. 9(a-d)**, AQuA consistently rejected all the regions from the poorly stained HS slide, while HistoQC only partially rejected some of these poorly stained regions. Although HistoQC successfully identified and rejected regions where RBCs obscured lung architecture (**Extended Data Fig. 9(e)**) as well as blurred, out-of-focus areas (**Extended Data Fig. 9(f)**), it failed to accurately detect regions lacking hematoxylin staining, as shown in **Extended Data Fig. 9(g-h)**. As for the well-stained HS slide shown in **Extended Data Fig. 9(i-l)**, both AQuA and HistoQC agreed on accepting most of the tissue regions, as desired. However, HistoQC erroneously rejected some regions with high-quality tissue staining (**Extended Data Fig. 9(o-p)**), while AQuA accurately accepted and preserved these stained areas. (Discussion section, 5th paragraph)*

Please also refer to **Extended Data Figure 9** for detailed detection results of HistoQC and AQuA.

2. Apart from being an automated framework, the authors should also mention how much time is saved for their proposed framework. For example, in Fig. 1(a), the authors can also compare the time for the two workflows so as to further show the strength of their AQuA.

We thank the reviewer for this valuable feedback. We have modified Fig. 1(a) captions to reflect the inference speed of AQuA framework per image, i.e., **~1-2 sec per image, where the average inference time of VS and VAF network models is ~2.6 ms for a single 1024×1024 pixel image**. Following the referee's suggestion, to better highlight the advantages of the AQuA framework, we have revised the Discussion section to shed more light on the time savings of AQuA, quoted:

“In addition to providing better detection accuracy, AQuA also presents considerable time savings compared to laborious manual assessment in the traditional HS workflow. H&E staining requires approximately 30-60 minutes using an automated slide stainer, which can take longer using manual staining protocols – especially for special stains. After the staining, human experts conduct quality control by examining the stained tissue slide under a microscope. A thorough examination, which involves assessing nuclear, cytoplasmic, and extracellular morphology under high magnification, can take an additional hour, depending on the training level of the examiner. For virtually stained tissue images, the chemical staining process is eliminated; however, the manual examination of the image by experts still needs to be performed to ensure a high stain quality. As we have shown in this work, realistic hallucinations of VS models can deceive human experts and remain undetected in their manual examination process, potentially misleading them to diagnose features that do not appear in the actual specimen, although exhibiting a good stain quality. In contrast, AQuA achieves a throughput of ~1-2 sec per image field-of-view (FOV) and

can accurately detect staining quality failures and hallucinations without the need for ground truth HS images of the samples. Parallelization of AQuA inference can further reduce this automated evaluation time as needed.” (Discussion section, 3rd paragraph)

3. The authors should provide a more in-depth analysis of why AQuA is superior to pathologists, especially in distinguishing realistic hallucinations. I would suggest the authors look into data similar to Extended Data Fig. 1(a) for more details. For example, what are the features in the images for the lung/kidney samples that make pathologists believe that the images are properly stained or what are the features that make AQuA able to catch those cases are realistic hallucinations? A follow-up interesting investigation is if the pathologists feel that it is ready for diagnosis, whether those features are important or not for diagnosis.

We appreciate the constructive and valuable advice. **Firstly, we would like to emphasize that the advantages acquired by AQuA were established upon blind tests on virtually stained images where our framework did not use any HS reference images (ground truth).** Our Results section reported that:

*“...In Fig. 3c, when detecting hallucinated VS images coming from poor-staining models, AQuA scores 100% rejection rate without any human supervision or access to HS ground truth images. In contrast, when presented with these VS images featuring realistic hallucinations (yellow zones), pathologists failed to flag these hallucinated images and could not identify the mismatch between VS images and their HS counterparts when the access to HS ground truth was unavailable. Examples of these realistic hallucinations that mislead the pathologists are shown in Extended Data Fig. 1a, and further analyses of pathologists’ failure on these images are detailed in the Discussion section. This comparison clearly demonstrated the super-human level performance of AQuA as an autonomous VS quality monitor and hallucination detector, rendering its potential in substituting human supervision in a VS workflow where HS is eliminated and not available...” (Results section, **Image-level autonomous quality and hallucination assessment of kidney tissue virtual staining**, 3rd paragraph)*

To better understand the superiority of AQuA over human pathologists, a group of board-certified pathologists further checked VS images containing realistic hallucinations while they were also given access to the corresponding HS reference images for each field of view (FOV). We have summarized three examples of tissue FOVs and the pathologists’ comments in the new **Extended Data Figure 3**, quoted below:

*“...An additional side-by-side comparison between these realistic hallucinations and the corresponding HS images is illustrated in **Extended Data Fig. 3**, showing a notable mismatch of nuclei in renal tubules and interstitium in the VS images that are inconspicuous without HS reference images...” (Results section, **Image-level autonomous quality and hallucination assessment of lung tissue virtual staining**, 2nd paragraph)*

Please also refer to **Extended Data Figure 3** to see these realistic hallucination examples and pathologists' comments.

Additionally, **the superiority of AQuA in detecting VS images with staining artifacts and hallucinations stems from the VS-AF iterations.** VS-AF iterations elevate AQuA's perception from 2D spatial-only information exploited by human perception and common image classifiers, into **3D spatial-temporal information evolution.** **To support this point, in our revised manuscript we utilized Grad-CAM to provide further interpretability on AQuA classification.** To summarize this analysis, we have added **Extended Data Figure 6, Supplementary Note 1** and revised the Discussion section, as quoted below:

*“...The effectiveness of VS-AF iterations can be further confirmed by the spatial-temporal Grad-CAM^{44,45} activation maps, as illustrated in **Extended Data Fig. 6** and analyzed in **Supplementary Note 1.** Leveraging the spatial-temporal features in the VS image sequence, the time-averaged activation map of AQuA effectively captures and locates modes of abnormality and inconsistency in the original VS images ($t=0$), including missing tissue regions, areas with high background noise, and residual tissue areas. As detailed in **Supplementary Note 1**, AQuA's activation maps successfully emphasize relatively large inconsistent regions that are also evidenced by the difference map between the VS and HS images (**Extended Data Fig. 6(d)**), as well as some of the smaller abnormal features that are undetectable by traditional pixel-wise difference maps (see **Extended Data Fig. 6(e, f)**). This comparison between AQuA's activation map and the HS reference image-based difference map further reveals the difficulty of VS image quality and hallucination assessment task even with the use of an HS reference image; this highlights the merits of AQuA as an autonomous hallucination and artifact assessment tool without the need for ground truth HS images...”* (Discussion section, 4th paragraph)

“Supplementary Note 1 Grad-CAM visualization analysis

*In this analysis, we visualize the spatial-temporal distribution of attributing features in AQuA inference on poor VS images of human kidney samples. Leveraging the activation maps within the residual blocks of the ResNet-50 backbone in AQuA-Net and the gradient from the positive class label, heatmaps are generated, overlapped with the VS image and displayed for VS images at each time step $VS(t)$; see the Methods section for implementation details. In addition, heatmaps are averaged over time, denoted as $\langle \text{Grad-CAM} \rangle$ to show the spatial locations of attributing features. For reference, the squared difference map between the VS image and HS reference image is also calculated and smoothed by a Gaussian kernel k to match the resolution of Grad-CAM. **Extended Data Figure 6** illustrates VS-AF iterations of a poor VS image and the corresponding Grad-CAM heatmaps. AQuA correctly classifies this VS image as positive. As shown by **Extended Data Fig. 6(c)**, heatmaps generated by AQuA without the HS reference image correlate well with the difference map calculated with respect to the HS reference image, confirming its autonomous quality assessment ability to serve as a watchdog in the VS workflow. Specifically, as pointed out by the arrows of **Extended Data Fig. 6(d)**, a relatively large tissue*

area is erased by the poor-staining VS network model and replaced by the hallucinatory boundaries of the tissue. Without the HS reference image, AQuA successfully detects this abnormal and inconsistent hallucination and highlights this area in the heatmaps. AQuA's decision is also attributed to small areas that are rare or abnormal in common good-quality VS images, including areas with high AF background noise and residual tissue, as pointed out by Extended Data Fig. 6(e) and (f), respectively. In contrast, the difference map based on the HS reference image fails to detect such minor inconsistencies or abnormalities. These observations confirm the difficulty of autonomous VS quality assessment task using traditional structural metrics-based terms and further emphasize the merits of the AQuA framework."

Please also refer to **Extended Data Figure 6** to see detailed Grad-CAM visualization results.

4. For Fig. 6(d), why the red curve can exceed 100% when M is 10?

The red curve represents KL divergence, whose axis label (also in red) was placed on the right of Fig. 6(d). The blue curve stands for accuracy and is capped at 100%.

5. I am afraid that equation (17) is incorrect. The denominators of sensitivity and specificity are not all negatives and positives, respectively. According to the equation (17), it should be PPV and NPV instead. Furthermore, the Specificity is also spelled wrongly in the equation. The authors should double check.

We appreciate the reviewer's important feedback. In our previous annotations in Eq. (17), P and N represented the number of actual positives and negatives, respectively. To avoid further confusion, we have clarified the definitions of these annotations, quoted below:

"For classification tasks involved in this work, we denote the number of true positives, true negatives, actual positives and actual negatives as TP, TN, P and N, respectively. The accuracy, sensitivity and specificity are respectively defined as

$$Acc. = \frac{TP+TN}{P+N}, Sensitivity = \frac{TP}{P}, Specificity = \frac{TN}{N}. \quad (17)$$

„

6. Following what the authors mentioned in the second last paragraph of the discussion, the lung sample is composed of red blood cells (RBCs) and scattered lymphocytes, which are under-representation in the training data, which causes AQuA accepting that image in Extended Data Fig. 1b. The authors should further address the data imbalance issues'

methodology as this is one of the common problems in deep learning, which already shows as a problem in the presented case.

We appreciate the reviewer for raising this important point. First of all, we would like to point out that the underrepresentation of RBCs in the training data of the VS model causes AQuA to accept the poorly stained RBC image. This mainly results from the following reasons: 1. the underrepresentation of RBCs in the training data. Most of the tissue slides available for training originate from neoplastic, inflammatory, and benign conditions. Tissues with a macroscopic engorged appearance are not sampled on a regular basis. 2. Tissue slices with severe bleeding typically undergo suboptimal fixation, leaving these areas with numerous artifacts. 3. These fixation related artifacts are a major culprit to the difficulty of registering label-free and HS FOVs with dense RBCs. **These RBC-rich areas are vulnerable to deformation, missing tissue and redistribution during the histological staining process (ground truth).**

Furthermore, in common pathologist workflows, standard pathological WSIs that are chemically stained do not contain areas with large, dense RBCs. When encountering small FOVs with high density and large areas of RBCs, which rarely appear in daily workflows, pathologists usually regard these images as poorly stained and assign very low scores.

Importantly, from a pathological diagnosis perspective, sample FOVs with dense RBCs carry almost no diagnostic information and are, therefore, automatically rejected by pathologists. As such, the false negatives produced by AQuA will not have any practical negative impact on pathological workflow or downstream tasks. We have further clarified this point in the Discussion section, as cited below:

*“...The only instance of a false negative evaluation (compared to pathologist consensus) that we observed is reported in **Extended Data Fig. 1b**; it is taken from another lung sample, where the entire patch is composed of RBCs and scattered lymphocytes. The under-representation of RBCs in the training data of normal lung tissue slides led to AQuA accepting this image FOV, whereas human pathologists, who are familiar with similar morphologies in lung specimens, disapproved its quality. Considering iatrogenic changes, including displaced epithelium and procedure-related bleeding, human diagnosticians typically disregard these areas when evaluating a tissue slide.” (Discussion section, 7th paragraph)*

7. Given the number of patients with limited amounts, how confident are the authors in supporting their claims on strong external generalization? This is also related to question 6.

We thank the reviewer for raising this important point. First, we would like to clarify that the effective area and the number of pixels of the tissue samples directly determine the size of the training and testing dataset for stain quality assessment and hallucination detection since different hallucinations or staining artifacts can independently appear at different pixels of an image. In fact, the research community on VS of label-free tissue has emphasized this distinction

in several articles that we have cited in our manuscript: *for VS-related learning tasks, every diffraction-limited pixel constitutes an independent label and, therefore, with billions of pixels of well registered tissue image FOVs, VS learning becomes rigorous with strong external generalization.* Therefore, the training dataset size, in the number of non-overlapping image FOVs and the number of pixels, affects the performance and generalizability of AQuA models. Regarding this, we have clarified the training, validation and test dataset sizes in the Methods section, quoted below:

“...For human kidney samples, a set of 1054 non-overlapping AF FOVs (each 1424×1424 pixels) were collected from 7 individuals for training and validation, and a set of 76 non-overlapping AF FOVs collected from another 3 subjects were used for testing... The testing dataset was generated by 7 good-staining and 6 poor-staining VS models excluded from training, consisting of ~1,000 VS images...”

For human lung samples, we collected 572 non-overlapping AF FOVs (each 1024×1024 pixels) from 7 individuals for training and validation (a subset of the training and validation data of VS models), and 268 non-overlapping AF FOVs from another subject for testing... The testing dataset was generated by 5 good-staining and 5 poor-staining VS models excluded from training and contains ~2,400 VS images...

The histochemical-stained H&E image dataset of human lung tissue sections consists of a subset of 100 non-overlapping FOVs (each 2048×2048 pixels) for training, 16 non-overlapping FOVs for validation, and 220 non-overlapping FOVs for testing.” (Methods section, Dataset preparation and splitting of AQuA)”

In brief, the total size of the training/validation and testing dataset that we used is ~150GB composed of ~4.6 billion unique pixels/labels.

In addition to detecting low-quality VS images generated by poor VS models, **we demonstrated that the same AQuA model can detect the generalization failures of a well-trained, overfitted VS model.** This generalization failure is common in practice as the differences in individual subjects, sample preparation protocols and imaging parameters may result in distribution shifts at the test time. This part of the results is illustrated in the **Extended Data Fig. 2**, analyzed in the Results section, quoted below:

“Furthermore, AQuA is effective in detecting various failure patterns of VS models, including suboptimal convergence and generalization failure. In addition to testing our AQuA model on hallucinated images from early stopped checkpoints, we conducted an external generalization test to an overfitted VS model using the same classifier described above, which has never seen such overfitted VS models. As shown in Extended Data Fig. 2a, the overfitted VS model, well trained on a small subset of the training data, fails to generalize on testing data and exhibits a distinct staining failure pattern compared with the early stopped VS models, such as missing a large portion of nuclei and generating blurry features. Despite the difficulty of detecting these

*staining failure patterns never seen before during the classifier training, AQuA successfully discriminated all VS images from the overfitted model and the early stopped model as illustrated in Extended Data Fig. 2b. This generalization test further demonstrates the robustness and generalizability of AQuA to various unseen, unknown staining failure patterns of VS models.” (Results section, **Image-level autonomous quality and hallucination assessment of kidney tissue virtual staining**, 4th paragraph)*

In the revised manuscript, we have further extended our testing of AQuA’s generalization to unseen types of organs. In these new experiments, both the VS model and AQuA were trained on one specific type of organ and then blindly tested on another type of organ never seen before. Due to the distribution shifts, the VS model could fail to generalize and as a result produce low-quality VS images, while AQuA generalizes and rejects poor VS images, scoring a high sensitivity. We have summarized these new experiments in the new **Extended Data Figure 10** and revised the Discussion section, quoted below:

*“Furthermore, AQuA exhibits robust external generalization to various unseen distributions during the blind testing stage. Extended Data Figure 2 demonstrates AQuA’s strong external generalization to unseen VS model failure patterns, where overfitted VS models are successfully detected by AQuA, which was trained only on failure modes of early stopped models. Additionally, AQuA’s robustness can be further extended to generalization on unseen tissue types. We blindly tested two AQuA models trained with human kidney (and lung) samples on VS images of human lung (and kidney) samples, respectively. Despite the distinct tissue structures of these two organs and different imaging parameters, including varying signal-to-noise ratio, AQuA successfully generalized and detected poor VS images with a sensitivity of 98.9%, as reported in **Extended Data Fig. 10(a, b)**. This strong generalization of AQuA effectively prevents low-quality VS images from a shifted data distribution passing through a pre-trained AQuA model. In addition to these, we also used transfer learning on each new tissue type to show that AQuA could achieve further improved inference performance, comparable to an AQuA model trained from scratch solely on the testing tissue type; see **Extended Data Fig. 10(c, d)**. This robustness of AQuA to new tissue types exemplifies its strong generalization, helping to avoid false negatives due to a shifted data distribution.” (Discussion section, 6th paragraph)*

Please also refer to **Extended Data Figure 10** for these detailed generalization test results.

In addition to these, our manuscript also demonstrated that AQuA is very effective at detecting poorly stained HS (histochemical) slides, better than other state-of-the-art approaches.

To better showcase the advantages of spatial-temporal information that AQuA harnesses, we compared AQuA against a well-known transformer-based model **TransMIL**, which was designed to classify HS tissue slides. This new comparison is summarized in the newly added **Extended Data Fig. 8** and analyzed in the revised Discussion section, quoted:

*“...When comparing AQuA with TransMIL, both of the models shared the same pre-trained ResNet-50 backbone⁴⁹ for feature extraction, and were trained on the same dataset of human lung HS images employing the same majority voting mechanism ($C = 5$). **Extended Data Figure 8** summarizes the comparison of these two models on the same testing set of human lung HS images and reports the corresponding confusion matrices; **the results show that TransMIL partially failed to separate the positive and negative populations with a low accuracy of 68.2%, whereas AQuA provides clear separation between the two populations and scored 100% accuracy. Additionally, the total number of trainable parameters of AQuA (~2.1 M) is less than TransMIL (~3.2 M), showcasing AQuA’s efficiency...**”(Discussion section, 5th paragraph)*

In the revised manuscript, we have also compared AQuA against a state-of-the-art quality control tool named **HistoQC** designed for digital pathology slides. To showcase this comparative analysis, we added **Extended Data Fig. 9** to our revised manuscript and compared the quality control detection results from the HistoQC model and the AQuA model. **The related analysis confirmed the superior performance of AQuA for detecting poorly stained HS images** and it is reported in the revised Discussion section, quoted:

*“...In addition to these, when comparing AQuA with HistoQC, AQuA demonstrated a more robust concordance with pathologists' judgments in assessing the quality of the slide staining. **Extended Data Figure 9** illustrates the quality control detection results from both the AQuA and HistoQC models for a poorly stained HS slide and a well-stained HS slide, as labeled by pathologists. As depicted in **Extended Data Fig. 9(a-d)**, AQuA consistently rejected all the regions from the poorly stained HS slide, while HistoQC only partially rejected some of these poorly stained regions. Although HistoQC successfully identified and rejected regions where RBCs obscured lung architecture (**Extended Data Fig. 9(e)**) as well as blurred, out-of-focus areas (**Extended Data Fig. 9(f)**), it failed to accurately detect regions lacking hematoxylin staining, as shown in **Extended Data Fig. 9(g-h)**. As for the well-stained HS slide shown in **Extended Data Fig. 9(i-l)**, both AQuA and HistoQC agreed on accepting most of the tissue regions, as desired. However, HistoQC erroneously rejected some regions with high-quality tissue staining (**Extended Data Fig. 9(o-p)**), while AQuA accurately accepted and preserved these stained areas. (Discussion section, 5th paragraph)*

Please also refer to **Extended Data Figure 9** for detailed detection results of HistoQC and AQuA.

Writing issues

1. Once the acronym is defined, the authors should use it throughout. For example, histochemical stain/staining (HS) is defined in the introduction. However, I do see many other places using histochemical stain/staining and HS interchangeably. It is also the case for autofluorescence vs AF. Please double check.

We have accordingly revised the manuscript and made consistent use of the acronyms.

Reviewer #2

1. Clinical significance: It's crucial to acknowledge that while virtual staining is advancing rapidly, its validation and acceptance in clinical settings are still controversial. Consequently, the assessment of virtual stains remains of limited clinical significance.

We agree that the virtual tissue staining field based on AI is rapidly booming both in academia, university medical centers, and research labs, as well as in commercial space and industry. Therefore, this field of research and development has been experiencing enormous growth in terms of papers, patents and commercialization efforts.

There are some regulatory steps that need to be carefully executed for virtual tissue staining to enter clinical decision-making as part of the standard digital pathology workflow for primary diagnosis. Among these, arguably, the most critical one is the reliability of the virtual staining outputs, which is the exact focus of our paper.

We believe AQuA offers a robust, scalable, and efficient quality assurance framework for detecting hallucinations in virtual staining images, representing a significant step towards more reliable and trustworthy AI in digital pathology applications related to virtual staining. Therefore, our rigorous analyses and methods reported in this manuscript would be transformative for academia, university medical centers, research labs and commercial efforts in translating their AI-based virtual staining methods into clinical workflow with autonomous quality assurance.

Toward this end, the key advantages acquired by AQuA were established upon blind tests on virtually stained images where our framework did not use any HS reference images (ground truth). Our Results section reported that:

*“...In Fig. 3c, when detecting hallucinated VS images coming from poor-staining models, AQuA scores 100% rejection rate without any human supervision or access to HS ground truth images. In contrast, when presented with these VS images featuring realistic hallucinations (yellow zones), pathologists failed to flag these hallucinated images and could not identify the mismatch between VS images and their HS counterparts when the access to HS ground truth was unavailable. Examples of these realistic hallucinations that mislead the pathologists are shown in Extended Data Fig. 1a, and further analyses of pathologists' failure on these images are detailed in the Discussion section. This comparison clearly demonstrated the super-human level performance of AQuA as an autonomous VS quality monitor and hallucination detector, rendering its potential in substituting human supervision in a VS workflow where HS is eliminated and not available.” (Results section, **Image-level autonomous quality and hallucination assessment of kidney tissue virtual staining**, 3rd paragraph)*

In addition to these, our manuscript also demonstrated that AQuA is very effective at detecting poorly stained HS (histochemical) slides (part of the standard workflow), better than other state-of-the-art approaches. To better showcase the advantages of spatial-temporal information that AQuA harnesses, we compared AQuA against a well-known transformer-based

model **TransMIL**, which was designed to classify HS tissue slides. This new comparison is summarized in the newly added **Extended Data Fig. 8** and analyzed in the revised Discussion section, quoted:

*“...When comparing AQuA with TransMIL, both of the models shared the same pre-trained ResNet-50 backbone⁴⁹ for feature extraction, and were trained on the same dataset of human lung HS images employing the same majority voting mechanism ($C = 5$). **Extended Data Figure 8** summarizes the comparison of these two models on the same testing set of human lung HS images and reports the corresponding confusion matrices; **the results show that TransMIL partially failed to separate the positive and negative populations with a low accuracy of 68.2%, whereas AQuA provides clear separation between the two populations and scored 100% accuracy. Additionally, the total number of trainable parameters of AQuA (~2.1 M) is less than TransMIL (~3.2 M), showcasing AQuA’s efficiency...**” (Discussion section, 5th paragraph)*

In the revised manuscript, we have also compared AQuA against a state-of-the-art quality control tool named **HistoQC** designed for digital pathology slides. To showcase this comparative analysis, we added **Extended Data Fig. 9** to our revised manuscript and compared the quality control detection results from the HistoQC model and the AQuA model. **The related analysis confirmed the superior performance of AQuA for detecting poorly stained HS images**, and it is reported in the revised Discussion section, quoted:

*“...In addition to these, when comparing AQuA with HistoQC, AQuA demonstrated a more robust concordance with pathologists' judgments in assessing the quality of the slide staining. **Extended Data Figure 9** illustrates the quality control detection results from both the AQuA and HistoQC models for a poorly stained HS slide and a well-stained HS slide, as labeled by pathologists. As depicted in **Extended Data Fig. 9(a-d)**, AQuA consistently rejected all the regions from the poorly stained HS slide, while HistoQC only partially rejected some of these poorly stained regions. Although HistoQC successfully identified and rejected regions where RBCs obscured lung architecture (**Extended Data Fig. 9(e)**) as well as blurred, out-of-focus areas (**Extended Data Fig. 9(f)**), it failed to accurately detect regions lacking hematoxylin staining, as shown in **Extended Data Fig. 9(g-h)**. As for the well-stained HS slide shown in **Extended Data Fig. 9(i-l)**, both AQuA and HistoQC agreed on accepting most of the tissue regions, as desired. However, HistoQC erroneously rejected some regions with high-quality tissue staining (**Extended Data Fig. 9(o-p)**), while AQuA accurately accepted and preserved these stained areas.(Discussion section, 5th paragraph)*

Please also refer to **Extended Data Figure 9** for detailed detection results of HistoQC and AQuA.

2. Insufficient interpretability: Highlighting areas that contribute to the prediction of classification models is crucial to understanding the decision-making process. However, the proposed framework only provides binary classification results instead of heatmap within the images, hence the potential clinical application is constrained.

We appreciate this valuable comment from the reviewer.

For better interpretability and to better understand the superiority of AQuA over human pathologists, a group of board-certified pathologists further checked VS images containing realistic hallucinations while they were also given access to the corresponding HS reference images for each field of view (FOV). We have summarized three examples of tissue FOVs and the pathologists' comments in **Extended Data Figure 3**, quoted below:

*"...An additional side-by-side comparison between these realistic hallucinations and the corresponding HS images is illustrated in **Extended Data Fig. 3**, showing a notable mismatch of nuclei in renal tubules and interstitium in the VS images that are inconspicuous without HS reference images..." (Results section, **Image-level autonomous quality and hallucination assessment of lung tissue virtual staining**, 2nd paragraph)*

Please also refer to **Extended Data Figure 3** to see these realistic hallucination examples and pathologists' comments.

Additionally, **the superiority of AQuA in detecting VS images with staining artifacts and hallucinations stems from the VS-AF iterations**. VS-AF iterations elevate AQuA's perception from 2D spatial-only information exploited by human perception and common image classifiers, into **3D spatial-temporal information evolution**. **To support this point, in our revised manuscript we utilized Grad-CAM to provide further interpretability on AQuA classification**. To summarize this analysis, we have added **Extended Data Figure 6, Supplementary Note 1** and revised the Discussion section, as quoted below:

*"...The effectiveness of VS-AF iterations can be further confirmed by the spatial-temporal Grad-CAM^{44,45} activation maps, as illustrated in **Extended Data Fig. 6** and analyzed in **Supplementary Note 1**. Leveraging the spatial-temporal features in the VS image sequence, the time-averaged activation map of AQuA effectively captures and locates modes of abnormality and inconsistency in the original VS images ($t=0$), including missing tissue regions, areas with high background noise, and residual tissue areas. As detailed in **Supplementary Note 1**, AQuA's activation maps successfully emphasize relatively large inconsistent regions that are also evidenced by the difference map between the VS and HS images (**Extended Data Fig. 6(d)**), as well as some of the smaller abnormal features that are undetectable by traditional pixel-wise difference maps (see **Extended Data Fig. 6(e, f)**). This comparison between AQuA's activation map and the HS reference image-based difference map further reveals the difficulty of VS image quality and hallucination assessment task even with the use of an HS reference image; this*

highlights the merits of AQuA as an autonomous hallucination and artifact assessment tool without the need for ground truth HS images...” (Discussion section, 4th paragraph)

“Supplementary Note 1 Grad-CAM visualization analysis

*In this analysis, we visualize the spatial-temporal distribution of attributing features in AQuA inference on poor VS images of human kidney samples. Leveraging the activation maps within the residual blocks of the ResNet-50 backbone in AQuA-Net and the gradient from the positive class label, heatmaps are generated, overlapped with the VS image and displayed for VS images at each time step $VS(t)$; see the Methods section for implementation details. In addition, heatmaps are averaged over time, denoted as $\langle \text{Grad-CAM} \rangle$ to show the spatial locations of attributing features. For reference, the squared difference map between the VS image and HS reference image is also calculated and smoothed by a Gaussian kernel k to match the resolution of Grad-CAM. **Extended Data Figure 6** illustrates VS-AF iterations of a poor VS image and the corresponding Grad-CAM heatmaps. AQuA correctly classifies this VS image as positive. As shown by **Extended Data Fig. 6(c)**, heatmaps generated by AQuA without the HS reference image correlate well with the difference map calculated with respect to the HS reference image, confirming its autonomous quality assessment ability to serve as a watchdog in the VS workflow. Specifically, as pointed out by the arrows of **Extended Data Fig. 6(d)**, a relatively large tissue area is erased by the poor-staining VS network model and replaced by the hallucinatory boundaries of the tissue. **Without the HS reference image, AQuA successfully detects this abnormal and inconsistent hallucination and highlights this area in the heatmaps. AQuA’s decision is also attributed to small areas that are rare or abnormal in common good-quality VS images, including areas with high AF background noise and residual tissue, as pointed out by Extended Data Fig. 6(e) and (f), respectively. In contrast, the difference map based on the HS reference image fails to detect such minor inconsistencies or abnormalities. These observations confirm the difficulty of autonomous VS quality assessment task using traditional structural metrics-based terms and further emphasize the merits of the AQuA framework.**”*

Please also refer to **Extended Data Figure 6** to see detailed Grad-CAM visualization results.

3. The construction of a virtual stain dataset: AQuA-Net is trained with a virtual stain dataset generated using a set of poor-staining and good-staining models. The label of a virtual stain is determined by the model instead of the hue within the stain. Note that even a good-staining model may produce some unacceptable stains, and vice versa, a poor-staining model may generate acceptable stains. The authors assume that all stains from a good-staining model are acceptable, yet theoretically, it may lead to degraded quality of the dataset.

Firstly, as pointed out in Figs. 2 and 4, common structural metrics such as MSE, PCC and PSNR failed to distinguish VS image outputs generated by well-trained and early-stopped VS models. Therefore, we applied the validation loss, which consists of both structural and perceptual metrics to comprehensively evaluate the quality of the VS images. Secondly, evaluating each

checkpoint on a set of validation images enables us to reduce the variance of the assigned labels. We believe this process precisely and consistently assigns labels for VS images, reduces the variance of training labels and stabilizes the training of AQuA-Net.

Thirdly, proper gray zones around the thresholds were set to avoid ambiguity in checkpoint labelling so that the checkpoints used for training, validation and testing were well separated in both validation loss and the number of training epochs. We have added a new **Extended Data Figure 12** and revised the Methods section to detail this checkpoint thresholding and selection process, quoted below:

*“...Extended Data Fig. 12 illustrates the thresholding and checkpoint selections for both human kidney and lung tissue VS models.” (Methods section, **Definition of good-staining and poor-staining models for virtual staining**)*

Please also refer to the new **Extended Data Figure 12** to see the validation loss curve and distribution of the selected checkpoints.

Lastly, AQuA performance is not only determined by the training dataset but also by its architecture, training strategy, etc. Through comparisons with consensus from board-certified pathologists, we validated the performance of AQuA. By the introduction of ensemble learning (majority voting) and VS-AF iterations, AQuA-Net scored very good agreement in comparison to pathologist consensus at the blind testing phase, and did not exhibit any sign of degraded performance caused by a low-quality training dataset, as shown in Figs. 3 and 5.

4. The dataset comprises images from merely 10 patients for kidney tissue and images from 29 patients for lung tissue, which is notably tiny dataset considering the complexity and heterogeneity of human pathology. This small-scale sample may hinder the generalizability and robustness of the developed framework, not to mention the wide application.

We thank the reviewer for raising this important point. First, we would like to clarify that the effective area and the number of pixels of the tissue samples directly determine the size of the training and testing dataset for stain quality assessment and hallucination detection since different hallucinations or staining artifacts can independently appear at different pixels of an image. In fact, the research community on VS of label-free tissue has emphasized this distinction in several articles that we have cited in our manuscript: *for VS-related learning tasks, every diffraction-limited pixel constitutes an independent label and, therefore, with billions of pixels of well registered tissue image FOVs, VS learning becomes rigorous with strong external generalization*. Therefore, the training dataset size, in the number of non-overlapping image FOVs and the number of pixels, affects the performance and generalizability of AQuA models. Therefore, we clarified the training, validation and test dataset sizes in the **Methods**, quoted: *“...For human kidney samples, a set of 1054 non-overlapping AF FOVs (each 1424×1424 pixels) were collected from 7 individuals for training and validation, and a set of 76 non-*

overlapping AF FOVs collected from another 3 subjects were used for testing... The testing dataset was generated by 7 good-staining and 6 poor-staining VS models excluded from training, consisting of ~1,000 VS images...

*For **human lung samples**, we collected **572 non-overlapping AF FOVs (each 1024×1024 pixels)** from 7 individuals for training and validation (a subset of the training and validation data of VS models), and **268 non-overlapping AF FOVs** from another subject for testing... The testing dataset was generated by 5 good-staining and 5 poor-staining VS models excluded from training and contains ~2,400 VS images...*

*The **histochemical-stained H&E image dataset** of human lung tissue sections consists of a subset of **100 non-overlapping FOVs (each 2048×2048 pixels)** for training, 16 non-overlapping FOVs for validation, and **220 non-overlapping FOVs for testing.**” (Methods section, **Dataset preparation and splitting of AQuA**)”*

→ In brief, the total size of the training/validation and testing dataset that we used is ~150GB composed of ~4.6 billion unique pixels/labels.

In addition to detecting low-quality VS images generated by poor VS models, **we demonstrated the same AQuA model can detect the generalization failures of a well-trained, overfitted VS model.** This generalization failure is common in practice as the differences in individual subjects, sample preparation protocols and imaging parameters may result in distribution shifts at the test time. This part of the results is illustrated in the **Extended Data Fig. 2**, analyzed in the Results section, quoted below:

*“Furthermore, AQuA is effective in detecting various failure patterns of VS models, including suboptimal convergence and generalization failure. In addition to testing our AQuA model on hallucinated images from early stopped checkpoints, we conducted an external generalization test to an overfitted VS model using the same classifier described above, which has never seen such overfitted VS models. As shown in **Extended Data Fig. 2a**, the overfitted VS model, well trained on a small subset of the training data, fails to generalize on testing data and exhibits a distinct staining failure pattern compared with the early stopped VS models, such as missing a large portion of nuclei and generating blurry features. Despite the difficulty of detecting these staining failure patterns never seen before during the classifier training, AQuA successfully discriminated all VS images from the overfitted model and the early stopped model as illustrated in **Extended Data Fig. 2b**. This generalization test further demonstrates the robustness and generalizability of AQuA to various unseen, unknown staining failure patterns of VS models.” (Results section, **Image-level autonomous quality and hallucination assessment of kidney tissue virtual staining**, 4th paragraph)*

In the revised manuscript, we have further extended our testing of AQuA’s generalization to unseen types of organs. In these new experiments, both the VS model and AQuA were trained on one specific type of organ and then blindly tested on another type of organ never seen

before. Due to the distribution shifts, the VS model could fail to generalize and as a result produce low-quality VS images, while AQuA generalizes and rejects poor VS images, scoring a high sensitivity. We have summarized these new experiments in the new **Extended Data Figure 10** and revised the Discussion section, quoted below:

*“Furthermore, AQuA exhibits robust external generalization to various unseen distributions during the blind testing stage. **Extended Data Figure 2** demonstrates AQuA’s strong external generalization to unseen VS model failure patterns, where overfitted VS models are successfully detected by AQuA, which was trained only on failure modes of early stopped models. Additionally, AQuA’s robustness can be further extended to generalization on unseen tissue types. We blindly tested two AQuA models trained with human kidney (and lung) samples on VS images of human lung (and kidney) samples, respectively. Despite the distinct tissue structures of these two organs and different imaging parameters, including varying signal-to-noise ratio, AQuA successfully generalized and detected poor VS images with a sensitivity of 98.9%, as reported in **Extended Data Fig. 10(a, b)**. This strong generalization of AQuA effectively prevents low-quality VS images from a shifted data distribution passing through a pre-trained AQuA model. In addition to these, we also used transfer learning on each new tissue type to show that AQuA could achieve further improved inference performance, comparable to an AQuA model trained from scratch solely on the testing tissue type; see **Extended Data Fig. 10(c, d)**. This robustness of AQuA to new tissue types exemplifies its strong generalization, helping to avoid false negatives due to a shifted data distribution.”* (Discussion section, 6th paragraph)

Please also refer to **Extended Data Figure 10** for these detailed generalization test results.

In addition to these, our manuscript also demonstrated that AQuA is very effective at detecting poorly stained HS (histochemically stained) slides, better than other state-of-the-art approaches. To better showcase the advantages of spatial-temporal information that AQuA harnesses, we compared AQuA against a well-known transformer-based model **TransMIL**, which was designed to classify HS tissue slides. This new comparison is summarized in the newly added **Extended Data Fig. 8** and analyzed in the revised Discussion section, quoted:

*“...When comparing AQuA with TransMIL, both of the models shared the same pre-trained ResNet-50 backbone⁴⁹ for feature extraction, and were trained on the same dataset of human lung HS images employing the same majority voting mechanism ($C = 5$). **Extended Data Figure 8** summarizes the comparison of these two models on the same testing set of human lung HS images and reports the corresponding confusion matrices; **the results show that TransMIL partially failed to separate the positive and negative populations with a low accuracy of 68.2%, whereas AQuA provides clear separation between the two populations and scored 100% accuracy. Additionally, the total number of trainable parameters of AQuA (~2.1 M) is less than TransMIL (~3.2 M), showcasing AQuA’s efficiency...**”* (Discussion section, 5th paragraph)

In the revised manuscript, we have also compared AQuA against a state-of-the-art quality control tool named **HistoQC** designed for digital pathology slides. To showcase this comparative analysis, we added **Extended Data Fig. 9** to our revised manuscript and compared the quality control detection results from the HistoQC model and the AQuA model. **The related analysis confirmed the superior performance of AQuA for detecting poorly stained HS images**, and it is reported in the revised Discussion section, quoted:

*“...In addition to these, when comparing AQuA with HistoQC, AQuA demonstrated a more robust concordance with pathologists' judgments in assessing the quality of the slide staining. **Extended Data Figure 9** illustrates the quality control detection results from both the AQuA and HistoQC models for a poorly stained HS slide and a well-stained HS slide, as labeled by pathologists. As depicted in **Extended Data Fig. 9(a-d)**, AQuA consistently rejected all the regions from the poorly stained HS slide, while HistoQC only partially rejected some of these poorly stained regions. Although HistoQC successfully identified and rejected regions where RBCs obscured lung architecture (**Extended Data Fig. 9(e)**) as well as blurred, out-of-focus areas (**Extended Data Fig. 9(f)**), it failed to accurately detect regions lacking hematoxylin staining, as shown in **Extended Data Fig. 9(g-h)**. As for the well-stained HS slide shown in **Extended Data Fig. 9(i-l)**, both AQuA and HistoQC agreed on accepting most of the tissue regions, as desired. However, HistoQC erroneously rejected some regions with high-quality tissue staining (**Extended Data Fig. 9(o-p)**), while AQuA accurately accepted and preserved these stained areas.”* (Discussion section, 5th paragraph)

Please also refer to **Extended Data Figure 9** for detailed detection results of HistoQC and AQuA.

Finally, considering that AQuA-Net is a lightweight network with only ~2.1 million trainable parameters, we believe our dataset size is sufficiently large to show the generalization of AQuA.

5. The performance of the virtual stain model: In Fig. 4(a), examples of negative and positive virtual stains are compared with the corresponding histochemically stained H&E image. It's evident that both virtual stains fail to present nuclei well, as indicated by the region to the left of the black arrow.

Following the reviewer's comment, **we demonstrated that the VS images generated by our optimal VS model (used in the VS-AF iterations) showed no statistically significant difference when compared with the corresponding HS images using multiple quantitative aspects**. This analysis is summarized in the newly added **Supplementary Note 2, Supplementary Figures 1 and 2**, quoted below:

*“**Supplementary Note 2 Comparison between VS and HS quality***

To validate the inference performance of the VS models used in the iterative inference, we

*conducted a quantitative study on kidney tissue samples to compare the similarity between VS H&E images and their HS counterparts. **Supplementary Figure 1** presents comparisons of the color distributions of 76 paired VS and HS FOVs (1024×1024 pixels) in the YCbCr color space and separated eosin and hematoxylin staining channels, demonstrating a high level of overlap between the virtual and histochemical staining. Additionally, as summarized in **Supplementary Figure 2**, the VS H&E images and their HS equivalents were further evaluated and compared using standard metrics such as PSNR and SSIM, along with different features, including the number of nuclei per FOV and the average area of nuclei. The high average PSNR (23.8241) and SSIM (0.8817) values also confirmed the success of the virtual staining performance. Furthermore, the closely matching distributions for the number of nuclei per FOV and average area of nuclei, together with the calculated Hellinger distances and related statistical analyses, substantiate that there is no statistically significant difference between the VS images and their HS counterparts, further validating the reliability of the G_{VS} model used in the VS-AF iterations.”*

Please also refer to **Supplementary Figures 1 and 2** to see quantitative comparisons between VS and HS images.

Besides, HS images intrinsically have variations, which means two adjacent H&E slides could have different appearances, including different nuclei numbers and locations. While subtle slide-to-slide variations may be detected within adjacent sections, their clinical importance is usually insignificant.

Minor

1. The model selection: The framework relies solely on Generative Adversarial Networks (GANs), and no experiments are conducted with other generative models, e.g. diffusion models. This lack of experimentation with alternative models raises questions about the comprehensiveness of the approach. Also, the approach to evaluate the quality of virtual stains from a different generative model is absent.

We thank the reviewer for raising this point. With the fast development of generative models recently, it can potentially be promising to apply advanced models such as diffusion models for virtual staining. However, to the best of our knowledge, such an application has not been rigorously demonstrated and examined/validated by board-certified pathologists yet. Therefore, we believe that the demonstration of AQUA on alternative and much less explored/validated generative models should be left for future research **only after** their applications on VS are demonstrated and rigorously validated through clinical studies. The VS model architecture reported in our work has been rigorously tested for various types of organs and stain types, including IHC (immunohistochemical) staining of tissue (e.g., HER2 IHC), as listed in our references.

2. Iterations between different domains: Several iterations between the H&E and AF domains are performed to generate image pairs as the input for AQuA-Net. It is of high risk that one model is not properly trained, then the whole pipeline fails.

We agree that the correct convergence of the VS and VAF models is important for AQuA. The VS and VAF models employed for the iterations between the two image domains were the checkpoints with the lowest validation losses. In the Methods section, we detailed our training strategies and dataset preparation to ensure that the two models converged correctly, quoted:

*“...The optimal VS generator network models for kidney and lung samples, utilized to facilitate the cycle of AF-H&E transformations within the AQuA model, were chosen based on the lowest validation loss values and evaluations conducted through human visual inspection. The staining quality of the optimal VS models was assessed and validated against HS reference images in **Supplementary Figs. 1, 2** and **Supplementary Note 2**, where multiple staining quality evaluation metrics indicated no statistically significant difference between the VS and the corresponding HS images.” (Methods section, **Network architecture and training schedule of VS networks**)*

Furthermore, we have added a new **Extended Data Figure 12** on the thresholding and selection of VS model checkpoints, visually confirming the convergence of our VS models, and revised the Methods section, quoted:

*“...**Extended Data Fig. 12** illustrates the thresholding and checkpoint selections for both human kidney and lung tissue VS models.” (Methods section, **Definition of good-staining and poor-staining models for virtual staining**)*

Please also refer to **Extended Data Figure 12** to see the validation loss curve and distribution of selected checkpoints.

3. The definition of good-staining models and poor-staining models: Good-staining and poor-staining models are defined based on a simple threshold applied to the number of epochs and loss value. However, this threshold-based approach lacks explanatory details and context.

Following the reviewer’s comment, we have added a new **Extended Data Figure 12** and revised the Methods section to provide more details and visualization of the selection of good- and poor-staining checkpoints, quoted below:

*“...**Extended Data Fig. 12** illustrates the thresholding and checkpoint selections for both human kidney and lung tissue VS models.” (Methods section, **Definition of good-staining and poor-staining models for virtual staining**)*

Please also refer to **Extended Data Figure 12** to see the validation loss curve and distribution of selected checkpoints.

Missing details

1. Details of dataset. The article lacks precise descriptions of the dataset used in the study. The specific types of diseases and the number of WSIs were missing.

Following the reviewer's comment, we have added more information about the pathological slides to the Methods section, quoted below:

*"...The kidney samples originated from 10 unique patients who experienced a myriad of nonneoplastic kidney diseases (7 used for training and validating the VS/VAE and AQuA models, and 3 for testing), while the lung samples came from 29 unique lung transplant recipients who underwent biopsy for transplant rejection evaluation... Note that each unique patient corresponds to a unique WSI in our study." (Methods section, **Sample preparation and standard histochemical H&E staining**)*

2. How to perform majority voting. The article should elaborate how majority voting is conducted within the framework. For example, how were the different heads trained?

Following the referee's comments, we have revised the Methods section to provide detailed clarification on the training and selection of multiple ensembles, quoted:

*"Each ensemble was trained with the same training and validation datasets and hyperparameters except for the random seeds. We initially trained 10 ensembles for each sample type and selected the best combination of C ensembles out of 10 based on the validation set. During transfer learning, only the C ensembles of the best combination on the pre-trained tissue type were transferred and then tested on the target tissue type." (Methods section, **Majority Voting**)*

Suggestions for improvement

1. Addressing artifacts in chemical staining: In addition to tackling the issue of virtual staining artifacts, it is important to focus on addressing natural artifacts in chemical staining, such as tissue folding or uneven staining. Excluding both virtual and chemical staining artifacts will improve the overall reliability and accuracy of the staining process.

We agree with the reviewer that detecting artifacts in HS images is as important as the quality control on VS images. In the Results section "***Autonomous staining quality assessment on histochemically stained images***", we demonstrated the adaptability of AQuA on artifact detection in HS (histochemically stained tissue) images. Our results and analyses demonstrated that AQuA is very effective at detecting poorly stained HS slides, better than other state-of-the-art approaches. To better showcase the advantages of spatial-temporal information that AQuA harnesses, we compared AQuA against a well-known transformer-based

model **TransMIL**, which was designed to classify HS tissue slides. This new comparison is summarized in the newly added **Extended Data Fig. 8** and analyzed in the revised Discussion section, quoted:

*“...When comparing AQuA with TransMIL, both of the models shared the same pre-trained ResNet-50 backbone⁴⁹ for feature extraction, and were trained on the same dataset of human lung HS images employing the same majority voting mechanism ($C = 5$). **Extended Data Figure 8** summarizes the comparison of these two models on the same testing set of human lung HS images and reports the corresponding confusion matrices; **the results show that TransMIL partially failed to separate the positive and negative populations with a low accuracy of 68.2%, whereas AQuA provides clear separation between the two populations and scored 100% accuracy. Additionally, the total number of trainable parameters of AQuA (~2.1 M) is less than TransMIL (~3.2 M), showcasing AQuA’s efficiency...**” (Discussion section, 5th paragraph)*

In the revised manuscript, we have also compared AQuA against a state-of-the-art quality control tool named **HistoQC** designed for digital pathology slides. To showcase this comparative analysis, we added **Extended Data Fig. 9** to our revised manuscript and compared the quality control detection results from the HistoQC model and the AQuA model. **The related analysis confirmed the superior performance of AQuA for detecting poorly stained HS images**, and it is reported in the revised Discussion section, quoted:

*“...In addition to these, when comparing AQuA with HistoQC, AQuA demonstrated a more robust concordance with pathologists' judgments in assessing the quality of the slide staining. **Extended Data Figure 9** illustrates the quality control detection results from both the AQuA and HistoQC models for a poorly stained HS slide and a well-stained HS slide, as labeled by pathologists. As depicted in **Extended Data Fig. 9(a-d)**, AQuA consistently rejected all the regions from the poorly stained HS slide, while HistoQC only partially rejected some of these poorly stained regions. Although HistoQC successfully identified and rejected regions where RBCs obscured lung architecture (**Extended Data Fig. 9(e)**) as well as blurred, out-of-focus areas (**Extended Data Fig. 9(f)**), it failed to accurately detect regions lacking hematoxylin staining, as shown in **Extended Data Fig. 9(g-h)**. As for the well-stained HS slide shown in **Extended Data Fig. 9(i-l)**, both AQuA and HistoQC agreed on accepting most of the tissue regions, as desired. However, HistoQC erroneously rejected some regions with high-quality tissue staining (**Extended Data Fig. 9(o-p)**), while AQuA accurately accepted and preserved these stained areas. (Discussion section, 5th paragraph)*

Please also refer to **Extended Data Figure 9** for detailed detection results of HistoQC and AQuA.

Besides, we have added a new **Extended Data Fig. 4** and revised the Results section to further illustrate examples of poorly stained HS images, quoted: “...**Extended Data Figure 4** further illustrates additional examples of poorly stained HS images and highlights several regions with

*unacceptable staining quality.” (Results section, **Autonomous staining quality assessment on HS images**)*

2. Explainability: To prove the effectiveness of rejecting a certain virtual stain, it is essential to visualize the regions of interest that the network focuses on.

We appreciate this valuable comment from the reviewer.

For better interpretability and to better understand the superiority of AQuA over human pathologists, a group of board-certified pathologists further checked VS images containing realistic hallucinations while they were also given access to the corresponding HS reference images for each field of view (FOV). **We have summarized three examples of tissue FOVs and the pathologists’ comments in Extended Data Figure 3**, quoted below:

*“...An additional side-by-side comparison between these realistic hallucinations and the corresponding HS images is illustrated in **Extended Data Fig. 3**, showing a notable mismatch of nuclei in renal tubules and interstitium in the VS images that are inconspicuous without HS reference images...” (Results section, **Image-level autonomous quality and hallucination assessment of lung tissue virtual staining**, 2nd paragraph)*

Please also refer to **Extended Data Figure 3** to see these realistic hallucination examples and pathologists’ comments.

Additionally, **the superiority of AQuA in detecting VS images with staining artifacts and hallucinations stems from the VS-AF iterations**. VS-AF iterations elevate AQuA’s perception from 2D spatial-only information exploited by human perception and common image classifiers, into **3D spatial-temporal information evolution**.

To support this point, in our revised manuscript we utilized Grad-CAM to provide further interpretability on AQuA classification. To summarize this analysis, we have added **Extended Data Figure 6, Supplementary Note 1** and revised the Discussion section, as quoted below:

*“...The effectiveness of VS-AF iterations can be further confirmed by the spatial-temporal Grad-CAM^{44,45} activation maps, as illustrated in **Extended Data Fig. 6** and analyzed in **Supplementary Note 1**. Leveraging the spatial-temporal features in the VS image sequence, the time-averaged activation map of AQuA effectively captures and locates modes of abnormality and inconsistency in the original VS images ($t=0$), including missing tissue regions, areas with high background noise, and residual tissue areas. As detailed in **Supplementary Note 1**, AQuA’s activation maps successfully emphasize relatively large inconsistent regions that are also evidenced by the difference map between the VS and HS images (**Extended Data Fig. 6(d)**), as well as some of the smaller abnormal features that are undetectable by traditional pixel-wise difference maps (see **Extended Data Fig. 6(e, f)**). This comparison between AQuA’s activation map and the HS reference image-based difference map further reveals the difficulty of VS image*

quality and hallucination assessment task even with the use of an HS reference image; this highlights the merits of AQuA as an autonomous hallucination and artifact assessment tool without the need for ground truth HS images...”

“Supplementary Note 1 Grad-CAM visualization analysis

*In this analysis, we visualize the spatial-temporal distribution of attributing features in AQuA inference on poor VS images of human kidney samples. Leveraging the activation maps within the residual blocks of the ResNet-50 backbone in AQuA-Net and the gradient from the positive class label, heatmaps are generated, overlapped with the VS image and displayed for VS images at each time step VS(t); see the Methods section for implementation details. In addition, heatmaps are averaged over time, denoted as <Grad-CAM> to show the spatial locations of attributing features. For reference, the squared difference map between the VS image and HS reference image is also calculated and smoothed by a Gaussian kernel k to match the resolution of Grad-CAM. **Extended Data Figure 6** illustrates VS-AF iterations of a poor VS image and the corresponding Grad-CAM heatmaps. AQuA correctly classifies this VS image as positive. As shown by **Extended Data Fig. 6(c)**, heatmaps generated by AQuA without the HS reference image correlate well with the difference map calculated with respect to the HS reference image, confirming its autonomous quality assessment ability to serve as a watchdog in the VS workflow. Specifically, as pointed out by the arrows of **Extended Data Fig. 6(d)**, a relatively large tissue area is erased by the poor-staining VS network model and replaced by the hallucinatory boundaries of the tissue. **Without the HS reference image, AQuA successfully detects this abnormal and inconsistent hallucination and highlights this area in the heatmaps. AQuA’s decision is also attributed to small areas that are rare or abnormal in common good-quality VS images, including areas with high AF background noise and residual tissue, as pointed out by Extended Data Fig. 6(e) and (f), respectively. In contrast, the difference map based on the HS reference image fails to detect such minor inconsistencies or abnormalities. These observations confirm the difficulty of autonomous VS quality assessment task using traditional structural metrics-based terms and further emphasize the merits of the AQuA framework.**”*

Please also refer to **Extended Data Figure 6** to see detailed Grad-CAM visualization results.

Reviewer #3

1. In general, the clinical use of the proposed method is unclear. Once a section is prepared, the physical staining can be performed very fast and highly cost-efficiently. Thereby, the exact rationale for virtually staining the sections of unstained slides, particularly since these need to be scanned using fluorescent microscopes, is not completely clear. In which setting this should be used? What would be the added computational requirements and time needed to run the model compared to the physical staining?

We sincerely thank the referee for helping us better clarify some of the important points of our manuscript.

First, the computational requirement for implementing AQuA is modest, and our results used a standard GPU-based PC (Intel Core i9-12900F, 64GB RAM, and a Nvidia RTX 3090 graphic card). We have modified Fig. 1(a) captions to reflect the inference speed of AQuA framework per image, i.e., **~1-2 sec per image, where the average inference time of VS and VAF models is ~2.6 ms for a single 1024×1024 pixel image. Therefore, our approach is very fast and does not require specialized, high-cost computers.**

To better highlight the advantages of the AQuA framework, we have revised the Discussion section to shed more light on the advantages of AQuA, quoted below:

“In addition to providing better detection accuracy, AQuA also presents considerable time savings compared to laborious manual assessment in the traditional HS workflow. H&E staining requires approximately 30-60 minutes using an automated slide stainer, which can take longer using manual staining protocols – especially for special stains. After the staining, human experts conduct quality control by examining the stained tissue slide under a microscope. A thorough examination, which involves assessing nuclear, cytoplasmic, and extracellular morphology under high magnification, can take an additional hour, depending on the training level of the examiner. For virtually stained tissue images, the chemical staining process is eliminated; however, the manual examination of the image by experts still needs to be performed to ensure a high stain quality. As we have shown in this work, realistic hallucinations of VS models can deceive human experts and remain undetected in their manual examination process, potentially misleading them to diagnose features that do not appear in the actual specimen, although exhibiting a good stain quality. In contrast, AQuA achieves a throughput of ~1-2 sec per image field-of-view (FOV) and can accurately detect staining quality failures and hallucinations without the need for ground truth HS images of the samples. Parallelization of AQuA inference can further reduce this automated evaluation time as needed.” (Discussion section, 3rd paragraph)

To summarize: Deep learning-based virtual tissue staining techniques enable rapid, cost-effective, and chemical-free histopathology, providing a powerful alternative to the traditional histological staining methods developed over a century. The VS model architecture reported in our work has been rigorously tested for various types of organs and stain types, **including IHC**

(immunohistochemical) staining of tissue (e.g., HER2 IHC), as listed in our references.

Virtual staining techniques eliminate the staining steps of the histology procedures, while leaving the sample pre-processing and preparation steps unchanged, making them compatible with the existing clinical workflow. The sample turnaround time (TAT) in pathology is defined by the Collage of American Pathologists as “the day the specimen is accessioned in the lab to the day the final report is signed out”. Even a modest shortening in the specimen preparation time by virtual staining (before the microscopic examination by pathologists) can make the difference between its examination before the end of the working day and the following day (resulting in at least a full-day difference in the TAT). In addition, since virtual staining exhibits reduced variability in the staining quality from slide to slide, it can result in a reduction of the number of technically failed stained slides, which will also benefit reducing the TAT. Furthermore, it is important to emphasize that the chemical staining process often makes up the major burden for histology labs, which requires the use, storage, and waste-processing of multiple types of reagents and antibodies, some of which are highly costly or toxic. Another major complication with various chemical stains is the quality assurance of the underlying chemicals, which are prone to supply-chain issues, as most histology labs experienced during the recent COVID pandemic. The staining procedures that involve trained histotechnicians to perform multiple staining protocols also form one of the most labor-intensive and time-consuming steps in histology. Therefore, eliminating the chemical staining process will greatly release the demanding requirements for lab infrastructure and personnel training, save valuable lab resources, and allow more samples to be processed under the same lab capacity.

Furthermore, as a general framework, virtual staining methods can be widely adapted to various sample preparation procedures, such as frozen sections, IHC (immunohistochemical stains), or freshly cut tissue blocks. Along with the advancement of label-free imaging/microscopy techniques, the traditional sample preparation process can potentially be replaced so that the whole histology workflow can be further accelerated. Besides time, cost and labor savings, virtual staining also inherently carries the capability of **stain multiplexing**. Different types of stains can be simultaneously generated at the same tissue cross-section using a single (or multiple) virtual staining model(s) to provide additional histological information that aids the diagnostic evaluation. By allowing different stains to be performed on the same tissue section, more tissue will be preserved and be available in diagnostically challenging cases for ancillary tests (e.g., DNA/RNA sequencing) that may be required to reach a diagnosis.

Motivated by these transformative advantages, the virtual tissue staining field based on AI is rapidly booming in academia, university medical centers, and research labs, as well as in commercial space and industry. Therefore, this field of research and development has been experiencing enormous growth in terms of papers, patents and commercialization efforts.

There are some regulatory steps that need to be carefully executed for virtual tissue staining to enter clinical decision-making as part of the standard digital pathology

workflow for primary diagnosis. Among these, arguably, the most critical one is the reliability of the virtual staining outputs, which is the exact focus of our paper. We believe AQuA offers a robust, scalable, and efficient quality assurance framework for detecting hallucinations in virtual staining images, representing a significant step towards more reliable and trustworthy AI in digital pathology applications related to virtual staining. Therefore, our rigorous analyses and methods reported in this manuscript would be transformative for academia, university medical centers, research labs and commercial efforts in translating their AI-based virtual staining methods into clinical workflow with autonomous quality assurance.

Toward this end, the key advantages acquired by AQuA were established upon blind tests on virtually stained images where our framework did not use any HS reference images (ground truth). Our Results section reported that:

*“...In Fig. 3c, when detecting hallucinated VS images coming from poor-staining models, AQuA scores 100% rejection rate without any human supervision or access to HS ground truth images. In contrast, when presented with these VS images featuring realistic hallucinations (yellow zones), pathologists failed to flag these hallucinated images and could not identify the mismatch between VS images and their HS counterparts when the access to HS ground truth was unavailable. Examples of these realistic hallucinations that mislead the pathologists are shown in Extended Data Fig. 1a, and further analyses of pathologists’ failure on these images are detailed in the Discussion section. This comparison clearly demonstrated the super-human level performance of AQuA as an autonomous VS quality monitor and hallucination detector, rendering its potential in substituting human supervision in a VS workflow where HS is eliminated and not available...” (Results section, **Image-level autonomous quality and hallucination assessment of kidney tissue virtual staining**, 3rd paragraph)*

In addition to these, our manuscript also demonstrated that AQuA is very effective at detecting poorly stained HS (histochemical) slides (part of the standard workflow), better than other state-of-the-art approaches. To better showcase the advantages of spatial-temporal information that AQuA harnesses, we compared AQuA against a well-known transformer-based model **TransMIL**, which was designed to classify HS tissue slides. This new comparison is summarized in the newly added **Extended Data Fig. 8** and analyzed in the revised Discussion section, quoted:

*“...When comparing AQuA with TransMIL, both of the models shared the same pre-trained ResNet-50 backbone⁴⁹ for feature extraction, and were trained on the same dataset of human lung HS images employing the same majority voting mechanism ($C = 5$). **Extended Data Figure 8** summarizes the comparison of these two models on the same testing set of human lung HS images and reports the corresponding confusion matrices; **the results show that TransMIL partially failed to separate the positive and negative populations with a low accuracy of 68.2%, whereas AQuA provides clear separation between the two populations and scored 100% accuracy. Additionally, the total number of trainable parameters of AQuA (~2.1 M) is less***

than TransMIL (~3.2 M), showcasing AQuA's efficiency..." (Discussion section, 5th paragraph)

In the revised manuscript, we have also compared AQuA against a state-of-the-art quality control tool named **HistoQC** designed for digital pathology slides. To showcase this comparative analysis, we added **Extended Data Fig. 9** to our revised manuscript and compared the quality control detection results from the HistoQC model and the AQuA model. **The related analysis confirmed the superior performance of AQuA for detecting poorly stained HS images**, and it is reported in the revised Discussion section, quoted:

*"...In addition to these, when comparing AQuA with HistoQC, AQuA demonstrated a more robust concordance with pathologists' judgments in assessing the quality of the slide staining. **Extended Data Figure 9** illustrates the quality control detection results from both the AQuA and HistoQC models for a poorly stained HS slide and a well-stained HS slide, as labeled by pathologists. As depicted in **Extended Data Fig. 9(a-d)**, AQuA consistently rejected all the regions from the poorly stained HS slide, while HistoQC only partially rejected some of these poorly stained regions. Although HistoQC successfully identified and rejected regions where RBCs obscured lung architecture (**Extended Data Fig. 9(e)**) as well as blurred, out-of-focus areas (**Extended Data Fig. 9(f)**), it failed to accurately detect regions lacking hematoxylin staining, as shown in **Extended Data Fig. 9(g-h)**. As for the well-stained HS slide shown in **Extended Data Fig. 9(i-l)**, both AQuA and HistoQC agreed on accepting most of the tissue regions, as desired. However, HistoQC erroneously rejected some regions with high-quality tissue staining (**Extended Data Fig. 9(o-p)**), while AQuA accurately accepted and preserved these stained areas. (Discussion section, 5th paragraph)*

Please also refer to **Extended Data Figure 9** for detailed detection results of HistoQC and AQuA.

Finally, we would like to emphasize that all unstained/label-free slides involved in our experiments were never specifically labelled and were imaged under autofluorescence channels using a standard digital pathology scanner—i.e., no external labeling by fluorophores was needed.

2. It is not possible for humans to accurately find inconsistencies and differences between the few shown exemplary virtually and physically stained slides. How could it be excluded that the “hallucinations” that are not detected by the proposed model wouldn't change other diagnoses that were not tested?

Firstly, we would like to clarify that no HS reference images (chemically stained H&E slides) were accessible when the board-certified pathologists participated in this study to analyze the virtually stained slides and their quality. In this study, we categorize hallucinations in VS images as two main types: (1) ‘technical and unrealistic’ hallucinations, resembling artifacts in HS

workflow, and (2) ‘realistic’ hallucinations. For the first type of hallucination, including blurred areas, missing and aberrantly stained areas, the pathologists are expected to flag them based on their experience (no access to HS images or the ground truth info). Therefore, by evaluating the performance of AQuA with respect to the consensus of board-certified pathologists, we showed the accuracy and effectiveness of AQuA in detecting these hallucinations (without access to HS reference images).

The second type of hallucination mainly results from the poor performance and/or poor generalization of the VS network model, such that the generated VS image does not reflect the ground truth image but passes the examination of the discriminator network. However, even experienced pathologists may fail to detect such hallucination (without access to HS images or the ground truth info) because it realistically resembles a potentially existing tissue slide in the real world.

In summary, we would like to emphasize that the advantages acquired by AQuA were established upon blind tests on virtually stained images where our framework did not use any HS reference images (ground truth). Our Results section reported that:

“...In Fig. 3c, when detecting hallucinated VS images coming from poor-staining models, AQuA scores 100% rejection rate without any human supervision or access to HS ground truth images. In contrast, when presented with these VS images featuring realistic hallucinations (yellow zones), pathologists failed to flag these hallucinated images and could not identify the mismatch between VS images and their HS counterparts when the access to HS ground truth was unavailable. Examples of these realistic hallucinations that mislead the pathologists are shown in Extended Data Fig. 1a, and further analyses of pathologists’ failure on these images are detailed in the Discussion section. This comparison clearly demonstrated the super-human level performance of AQuA as an autonomous VS quality monitor and hallucination detector, rendering its potential in substituting human supervision in a VS workflow where HS is eliminated and not available...”

To better understand this superiority of AQuA over human pathologists, a group of board-certified pathologists further checked VS images containing realistic hallucinations while they were also given access to the corresponding HS reference images for each field of view (FOV). Therefore, we further presented these images (located in the yellow zones in Fig. 3 and Fig. 5) again to the board-certified pathologists, along with the corresponding HS reference images. The pathologists confirmed the existence of realistic hallucinations and explained how they resemble the features from normal HS slides. We have added a new **Extended Data Fig. 3** and revised the Results section to show side-by-side comparisons between exemplar VS images with realistic hallucinations and the corresponding HS reference images, along with comments from pathologists, quoted below:

*“...An additional side-by-side comparison between these realistic hallucinations and the corresponding HS images is illustrated in **Extended Data Fig. 3**, showing a notable mismatch of*

*nuclei in renal tubules and interstitium in the VS images that are inconspicuous without HS reference images.” (Results section, **Image-level autonomous quality and hallucination assessment of lung tissue virtual staining**, 2nd paragraph)*

Please also refer to **Extended Data Figure 3** to see these realistic hallucination examples and pathologists’ comments.

This superiority of AQuA in detecting VS images with staining artifacts and hallucinations stems from the VS-AF iterations. VS-AF iterations elevate AQuA’s perception from 2D spatial-only information exploited by human perception and common image classifiers, into **3D spatial-temporal information evolution.** **To support this point, in our revised manuscript we utilized Grad-CAM to provide further interpretability on AQuA classification.** To summarize this analysis, we have added **Extended Data Figure 6, Supplementary Note 1** and revised the Discussion section, as quoted below:

*“...The effectiveness of VS-AF iterations can be further confirmed by the spatial-temporal Grad-CAM^{44,45} activation maps, as illustrated in **Extended Data Fig. 6** and analyzed in **Supplementary Note 1.** Leveraging the spatial-temporal features in the VS image sequence, the time-averaged activation map of AQuA effectively captures and locates modes of abnormality and inconsistency in the original VS images ($t=0$), including missing tissue regions, areas with high background noise, and residual tissue areas. As detailed in **Supplementary Note 1**, AQuA’s activation maps successfully emphasize relatively large inconsistent regions that are also evidenced by the difference map between the VS and HS images (**Extended Data Fig. 6(d)**), as well as some of the smaller abnormal features that are undetectable by traditional pixel-wise difference maps (see **Extended Data Fig. 6(e, f)**). This comparison between AQuA’s activation map and the HS reference image-based difference map further reveals the difficulty of VS image quality and hallucination assessment task even with the use of an HS reference image; this highlights the merits of AQuA as an autonomous hallucination and artifact assessment tool without the need for ground truth HS images...”*

“Supplementary Note 1 Grad-CAM visualization analysis

*In this analysis, we visualize the spatial-temporal distribution of attributing features in AQuA inference on poor VS images of human kidney samples. Leveraging the activation maps within the residual blocks of the ResNet-50 backbone in AQuA-Net and the gradient from the positive class label, heatmaps are generated, overlapped with the VS image and displayed for VS images at each time step $VS(t)$; see the Methods section for implementation details. In addition, heatmaps are averaged over time, denoted as $\langle \text{Grad-CAM} \rangle$ to show the spatial locations of attributing features. For reference, the squared difference map between the VS image and HS reference image is also calculated and smoothed by a Gaussian kernel k to match the resolution of Grad-CAM. **Extended Data Figure 6** illustrates VS-AF iterations of a poor VS image and the corresponding Grad-CAM heatmaps. AQuA correctly classifies this VS image as positive. As shown by **Extended Data Fig. 6(c)**, heatmaps generated by AQuA without the HS reference*

*image correlate well with the difference map calculated with respect to the HS reference image, confirming its autonomous quality assessment ability to serve as a watchdog in the VS workflow. Specifically, as pointed out by the arrows of **Extended Data Fig. 6(d)**, a relatively large tissue area is erased by the poor-staining VS network model and replaced by the hallucinatory boundaries of the tissue. **Without the HS reference image, AQuA successfully detects this abnormal and inconsistent hallucination and highlights this area in the heatmaps. AQuA's decision is also attributed to small areas that are rare or abnormal in common good-quality VS images, including areas with high AF background noise and residual tissue, as pointed out by Extended Data Fig. 6(e) and (f), respectively. In contrast, the difference map based on the HS reference image fails to detect such minor inconsistencies or abnormalities. These observations confirm the difficulty of autonomous VS quality assessment task using traditional structural metrics-based terms and further emphasize the merits of the AQuA framework.***

Please also refer to **Extended Data Figure 6** to see detailed Grad-CAM visualization results.

Lastly, to better showcase the advantages of spatial-temporal information, we compared AQuA against a well-known transformer-based model **TransMIL**, which was designed to classify HS slides. This new comparison is summarized in the newly added **Extended Data Fig. 8** and analyzed in the revised Discussion section, quoted:

*“...When comparing AQuA with TransMIL, both of the models shared the same pre-trained ResNet-50 backbone⁴⁹ for feature extraction, and were trained on the same dataset of human lung HS images employing the same majority voting mechanism ($C = 5$). **Extended Data Figure 8** summarizes the comparison of these two models on the same testing set of human lung HS images and reports the corresponding confusion matrices; **the results show that TransMIL partially failed to separate the positive and negative populations with a low accuracy of 68.2%, whereas AQuA provides clear separation between the two populations and scored 100% accuracy. Additionally, the total number of trainable parameters of AQuA (~2.1 M) is less than TransMIL (~3.2 M), showcasing AQuA's efficiency...**” (Discussion section, 5th paragraph)*

3. To validate the model, it is important to look at quantifiable metrics between the virtually and physically stained sections. Unfortunately, the authors didn't use such hand-crafted features to validate their model. As a simple example, this could allow checking if the number and shape of nuclei inside different structures are consistent. E.g., does Glomerulus number 1 in virtual stain have the same number of nuclei as the Glomerulus number 1 in the physically stained section? This would be particularly interesting since the authors showed some of the artifacts as being fake or missing nuclei. This would be diagnostically relevant. Numerous such features would be relevant.

We appreciate the reviewer's insights on this matter. To quantitatively validate the reliability and

accuracy of the fixed VS model used in AQuA, we conducted a comprehensive study. This study involved a thorough comparison of VS and HS kidney tissue images from multiple aspects, including color distributions, standard structural metrics (PSNR and SSIM), and hand-crafted features such as the number of nuclei per FOV and average nuclei area. Additionally, we utilized Hellinger distance calculations and statistical analyses to further substantiate our findings. The results were summarized in **Supplementary Figures 1 - 2** and added in the new **Supplementary Note 2**, quoted below:

“Supplementary Note 2 Comparison between VS and HS quality

*To validate the inference performance of the VS models used in the iterative inference, we conducted a quantitative study on kidney tissue samples to compare the similarity between VS H&E images and their HS counterparts. **Supplementary Figure 1** presents comparisons of the color distributions of 76 paired VS and HS FOVs (1024×1024 pixels) in the YCbCr color space and separated eosin and hematoxylin staining channels, demonstrating a high level of overlap between the virtual and histochemical staining. Additionally, as summarized in **Supplementary Figure 2**, the VS H&E images and their HS equivalents were further evaluated and compared using standard metrics such as PSNR and SSIM, along with different features, including the number of nuclei per FOV and the average area of nuclei. **The high average PSNR (23.8241) and SSIM (0.8817) values also confirmed the success of the virtual staining performance. Furthermore, the closely matching distributions for the number of nuclei per FOV and average area of nuclei, together with the calculated Hellinger distances and related statistical analyses, substantiate that there is no statistically significant difference between the VS images and their HS counterparts, further validating the reliability of the G_{VS} model used in the VS-AF iterations.**”*

Please also refer to **Supplementary Figures 1 and 2** to see quantitative comparisons between VS and HS images.

The detailed methods to perform this quantitative evaluation of the VS model used in the iterative inference were detailed in the **revised Methods section**, quoted below:

“Quantitative evaluation of the virtual staining model used in VS-AF iterations

To quantitatively assess the match between the VS images and their HS counterparts, we analyzed 76 pairs of VS and HS kidney images, each with 1424×1424 pixels. First, to compare the color distributions of the VS and HS images, they were converted from the RGB color space into the YCbCr color space. The intensity histograms for all 76 test FOVs were plotted for the Y, Cb, and Cr channels, as shown in Supplementary Figure 1(b). Additionally, using the stain vector decomposition, we separated the RGB images into channels corresponding to eosin and hematoxylin staining, and their respective intensity histograms were presented in Supplementary Figure 1(c). To further evaluate the similarity between VS and HS images, PSNR and SSIM values were calculated for the same test FOVs based on Eqs. 13 and 3, respectively, and reported in Supplementary Figure 2(b-c). Moreover, the nuclei segmentation and the quantification of

different features—the number of nuclei per FOV and average nuclei area—were performed using the same methodologies outlined in the previous section. The box plots used to illustrate the distribution of the number of nuclei per FOV and average nuclei area of VS and HS images are shown in Supplementary Figure 2(d-e). Then, the Hellinger distances⁶⁹ for each feature were calculated as follows:

$$D_{\text{Hellinger}} = \sqrt{1 - \sum_{i=1}^k \sqrt{p_i q_i}} \quad (20)$$

where $P = (p_1, p_2, \dots, p_k)$ represents the discrete probability distribution of the histograms for each feature in VS images, $Q = (q_1, q_2, \dots, q_k)$ in HS images, and k is the number of bins of histograms, which is 7 in measuring the number of nuclei per FOV and 6 for the average nuclei area. Lastly, the G -tests⁷⁰ were employed to statistically compare the distribution differences in the number of nuclei per FOV and average nuclei area between the VS and HS images, with a statistical significance level of 0.05. $k = 8$ was adopted for both G -tests.”

4. The authors calculated metrics such as MSE, PCC, or PSNR for quality assessment between the statistical approaches and the virtually stained slides. The results are, unsurprisingly, in favor of the proposed model. This might be because these metrics are too simple to capture the hallucinations. The authors should compare their model to a supervised model, e.g., such as TransMIL.

We thank the reviewer for suggesting this valuable comparison. We trained the TransMIL on the same training dataset of human lung HS images with the same majority voting mechanism as AQuA. **Though utilizing fewer training parameters, AQuA scored significantly higher classification accuracy and provided better separation of the good and bad HS image populations.** This new comparison is summarized in the new **Extended Data Fig. 8** and analyzed in the revised Discussion section, quoted below:

*“..When comparing AQuA with TransMIL, both of the models shared the same pre-trained ResNet-50 backbone⁴⁹ for feature extraction, and were trained on the same dataset of human lung HS images employing the same majority voting mechanism ($C = 5$). **Extended Data Figure 8** summarizes the comparison of these two models on the same testing set of human lung HS images and reports the corresponding confusion matrices; **the results show that TransMIL partially failed to separate the positive and negative populations with a low accuracy of 68.2%, whereas AQuA provides clear separation between the two populations and scored 100% accuracy. Additionally, the total number of trainable parameters of AQuA (~2.1 M) is less than TransMIL (~3.2 M), showcasing AQuA’s efficiency...**” (Discussion section, 5th paragraph)*

Please refer to the **Extended Data Figure 8** for a detailed comparison between the two models. Moreover, the detailed implementation of TransMIL was provided in the Methods, quoted below:

“Implementation of TransMIL

Following the work of Shao et al.⁴⁶, TransMIL was trained, validated and tested on the HS image dataset of human lung tissue samples. Each full-resolution HS image (2048×2048 pixel) was first cropped into a 9×9 grid consisting of 224×224 patches. The same pre-trained ResNet-50 backbone was utilized to encode the sequence of patches, producing an encoding matrix of 81×2048 for further processing by the transformer. An AdamW optimizer with an initial learning rate of 10^{-4} and a BCE loss function were employed for optimization. All TransMIL models were trained for 100 epochs from scratch, and the checkpoints with the best validation accuracy were picked for testing.”

5. There are many methods available to detect tissue artifacts, the most prominent probably being HistoQC. A benchmark against such available methods/tools is missing.

We thank the reviewer for this insightful suggestion. In the revised manuscript, we added **Extended Data Fig. 9** to compare the quality control detection results from the HistoQC model and the AQuA model. The related analysis was provided in the revised Discussion section, quoted below:

*“...In addition to these, when comparing AQuA with HistoQC, AQuA demonstrated a more robust concordance with pathologists' judgments in assessing the quality of the slide staining. **Extended Data Figure 9** illustrates the quality control detection results from both the AQuA and HistoQC models for a poorly stained HS slide and a well-stained HS slide, as labeled by pathologists. As depicted in **Extended Data Fig. 9(a-d)**, AQuA consistently rejected all the regions from the poorly stained HS slide, while HistoQC only partially rejected some of these poorly stained regions. Although HistoQC successfully identified and rejected regions where RBCs obscured lung architecture (**Extended Data Fig. 9(e)**) as well as blurred, out-of-focus areas (**Extended Data Fig. 9(f)**), it failed to accurately detect regions lacking hematoxylin staining, as shown in **Extended Data Fig. 9(g-h)**. As for the well-stained HS slide shown in **Extended Data Fig. 9(i-l)**, both AQuA and HistoQC agreed on accepting most of the tissue regions, as desired. However, HistoQC erroneously rejected some regions with high-quality tissue staining (**Extended Data Fig. 9(o-p)**), while AQuA accurately accepted and preserved these stained areas. (Discussion section, 5th paragraph)*

Please also refer to the **Extended Data Figure 9** for the detailed detection results of HistoQC and AQuA.

Moreover, the detailed methods to perform the comparison between the HistoQC model and the AQuA-Net model were provided in the Methods section, quoted below:

“Comparison between HistoQC and AQuA

The quality control detection results for the HistoQC model were generated using the open-source HistoQC model with its default configuration (version 2.1)^{47,48}. We optimized the

parameter “blur_radius” to 5 to improve the detection accuracy. In our comparative analysis, the acceptance probability maps predicted by the AQuA model were obtained by cropping the test HS WSIs into 2048×2048 pixel patches without overlapping, which were then independently fed through the AQuA model to obtain the predicted acceptance scores.”

6. The authors mention attack strategies. Please discuss or evaluate how integrating AQuA into the virtual staining model, e.g., as part of the discriminator that needs to be fooled, would change the study results. Please also evaluate in a classification setting whether AQuA can be used to detect malicious changes in images (adversarial attacks).

We thank the reviewer for this insightful suggestion. In the revised manuscript, this discussion has been added to the Discussion section, quoted below:

“Besides detecting hallucinations and generalization failures of a trained VS neural network, AQuA can be potentially applied to provide another level of protection to the VS pipeline against adversarial attacks. Common adversarial attack techniques utilize the gradient of a trained neural network or its approximation to create imperceptible perturbations that mislead the network^{51–53}. With the iterative inference of two separate neural networks (VS and VAF) between the two image domains, AQuA is intrinsically robust to gradient-based attacks and potentially applicable to detecting noise-corrupted inputs⁵⁴. Moreover, AQuA framework can be prospectively beneficial when serving as an additional supervision metric to enhance the robustness of generative models^{55–58}.” (Discussion section, 8th paragraph)

Minor points

1. Several parts of the manuscript are repetitive and could be shortened. Most of the methods section is rather common knowledge and could be moved to the appendix section.

Thanks for the suggestion. We have revised the Methods section to make it more concise.

2. Figures 3 and 5 - the second rows of both figures are redundant and add no extra information.

Thanks for this suggestion. However, we think the second row (part (b)) provides the complete decomposition of part (a), so we kept it in the revised figures.

3. Some text sections are printed bold, please omit such modifications.

We have re-organized the manuscript according to the formatting guidance of Nature BME.

Response Letter:

We briefly summarize below the main additions made in the revised manuscript, following the reviewers' feedback. We then address, point by point, the specific comments that were raised by the reviewers. Throughout this letter, the original referee comments are shown in black color, whereas for ease of communication, our answers are provided in blue. Our revisions have also been marked in the main text and supplementary information files using yellow highlighting.

Summary of the revisions

- Detailed analysis on the influence of varying thresholds on AQuA classification performance. The following new analysis and figure related to this topic are added to our revised manuscript:
 - Revision to the Results section
 - **The new Supplementary Figure 1:** Receiver operation characteristic (ROC) curve and precision-recall curve (PRC) of (a) kidney and (b) lung VS AQuA models.
- Extensive new experiments and analyses on the external and publicly available TCGA dataset. For these analyses, we used whole slide images (WSIs) of human lung tissue collected by 49 different tissue sites within the TCGA dataset. After transferring pre-trained AQuA on only a small subset of 10 well-stained WSIs in this dataset, we tested it on an extensive testing set of 989 HS images, including 519 good and 470 bad images, sampled from 451 TCGA WSIs. Our experimental results and analyses strongly show, once again, the good generalization and robustness of AQuA to various unseen data variations, including various sample sources and patient conditions, varied sample preparation protocols and imaging hardware from tens of different sites. These new experiments and analyses are summarized in the following new figure and the revised Results and Methods sections:
 - Revised “Autonomous staining quality assessment on HS images” subsection of the Results section
 - **The new Extended Data Figure 5:** Autonomous quality assessment on TCGA image dataset.
 - **The new subsection “TCGA Dataset” in the Methods section**
- Additional results on Grad-CAM visualization. We provide additional results on different fields of view (FOVs) to better confirm the conclusions observed from Grad-CAM visualizations. The following new figures and texts related to this topic are added to the revised manuscript:
 - **The new Supplementary Figures 2 and 3:** Spatial-temporal Grad-CAM visualization of the activation maps during the AQuA inference on poor VS images of human kidney sample.
 - **New paragraph in the revised Supplementary Note 1:** More analysis on Grad-CAM visualization results
- Extensive comparisons between TransMIL and our method. In addition to the image-level comparisons between the two models provided in the previous round of revisions, we further implemented a comparison at the WSI level. This comparison confirmed the high accuracy of AQuA in evaluating the quality of histochemically stained images. This new result has been summarized in the following new figures and revisions in our manuscript:
 - Revised Discussion section: WSI-level quality control detection comparisons between

- the HistoQC model, the TransMIL model and the AQuA model.
 - **The revised Extended Data Figure 10:** Quality control detection results from the HistoQC model, the TransMIL model and the AQuA model.
 - **The new Supplementary Figure 4:** Additional examples of the quality control detection results from the HistoQC model, the TransMIL model and the AQuA model.
- New visualization results and analysis on the poor-staining VS models identified by our AQuA workflow on diagnosis. In this round of the revision, we implemented a case study on a patient with lung transplant rejection by comparing the VS outputs from two poor-staining VS models and a good-staining VS model against their HS ground truth. The new results and analyses are summarized in the following new figure and the revised sections:
 - **The new Extended Data Figure 12:** Comparisons of images from two poor-staining and one good-staining VS models, identified by the AQuA workflow, against their HS references in a lung transplant rejection case.
 - **New paragraph in the Discussion section:** Diagnostic analysis using VS outputs from two poor-staining VS models and a good-staining VS model against their HS ground truth.
- In addition to these, we have added the following content to our revised manuscript:
 - Revised paragraph in the Discussion section about potential applications of AQuA on stain-to-stain transformations and stain normalization methods
 - **Newly added subsection in the Methods section:** Division of good-staining and poor-staining histochemical WSIs
 - Added details in the Methods section to clarify the appropriateness of the model selection method based on our loss functions.

In summary, the following new figures, new supplementary notes and new sub-sections are added to our revised manuscript:

New Figures Added

- **New Extended Data Figure 5:** Autonomous quality assessment on TCGA image dataset. (a) 519 good and 470 bad HS images were randomly cropped from the selected WSIs for testing. (b) Boxplot of confidence scores given by AQuA. AQuA model used transfer learning on a small subset (10 slides) of only well-stained TCGA WSIs. This blind testing reveals the external generalization capability of AQuA on bad TCGA WSIs from various tissue sites, staining protocols and imaging hardware since no bad HS images from the TCGA dataset were used in the training phase. (c) Confusion matrix of AQuA's testing results on the TCGA dataset. See the Methods for details.
- **New Extended Data Figure 12:** Comparisons of images from two poor-staining and one good-staining VS models, identified by the AQuA workflow, against their HS references in a lung transplant rejection case. (a-d) Large region images from a poor-staining VS model with non-realistic hallucinations; a poor-staining VS model with realistic hallucinations; a good-staining VS model; and the corresponding HS reference. (e-h) Zoomed-in images from the same large regions - from the poor-staining VS models, the good-staining model and the HS reference. The circled region corresponds to cell infiltrate, indicative of acute transplant rejection. (i-l) Zoomed-

in images from another diagnostically relevant region, maintaining the same comparison format as in (e-h). The circled region corresponds to cell infiltrate, indicative of acute transplant rejection.

- **New Supplementary Figure 1:** Receiver operation characteristic (ROC) curve and precision-recall curve (PRC) of (a) kidney and (b) lung VS AQuA models. (c) ROC and PRC curves of *M*-AQuA model on lung VS images. AUC: area under the curve; N: the number of samples in the testing set. Metrics are calculated on testing sets of the corresponding organ type.
- **New Supplementary Figure 2:** Spatial-temporal Grad-CAM visualization of the activation maps during the AQuA inference on poor VS images of human kidney sample. (a) AF and VAF images in the VS-AF iterations, and the HS reference image (ground truth). (b) VS images in the VS-AF iterations, and the difference map between the VS($t=0$) and the HS reference image is smoothed by a Gaussian kernel k with $\sigma = 15.5\text{px}$ to match the resolution of Grad-CAM. \otimes stands for 2D convolution. (c) Grad-CAM heatmaps for VS(t) and the averaged heatmap. $\langle \cdot \rangle$ stands for averaging over time. PCC values were calculated between the difference map and the averaged heatmap. (d) Nuclei artifacts generated by virtual staining were successfully identified by AQuA without relying on an HS reference image. (e) A tiny residual tissue not cleared in sample preparation was successfully detected by AQuA. (f) Significant artifacts and hallucinatory tissues generated by virtual staining were successfully detected by AQuA without relying on an HS reference image. (g) Aggregate of large nuclei identified by AQuA without relying on an HS reference image. (h) Hallucinatory, irregular nuclei detected by AQuA. (d, e, f, g, h) are pointed by arrows in (a).
- **New Supplementary Figure 3:** Spatial-temporal Grad-CAM visualization of the activation maps during the AQuA inference on poor VS images of human kidney sample. (a) AF and VAF images in the VS-AF iterations, and the HS reference image (ground truth). (b) VS images in the VS-AF iterations, and the difference map between the VS($t=0$) and the HS reference image is smoothed by a Gaussian kernel k with $\sigma = 15.5\text{px}$ to match the resolution of Grad-CAM. \otimes stands for 2D convolution. (c) Grad-CAM heatmaps for VS(t) and the averaged heatmap. $\langle \cdot \rangle$ stands for averaging over time. PCC values were calculated between the difference map and the averaged heatmap. (d, e) Small fragmented areas detected by AQuA without relying on an HS reference image. Note that parts of the tissue in the corresponding regions were not present in the HS reference. (f, g) Areas with high AF intensity not derived from tissue morphology were successfully detected by AQuA. The HS reference confirmed AQuA predictions.
- **New Supplementary Figure 4:** Additional examples of the quality control detection results from the HistoQC model, the TransMIL model and the AQuA model. (a) The WSI of a poorly stained HS slide labelled by pathologists. (b) The detection results of (a) using the HistoQC model, where pink regions indicate high-quality stained areas accepted by the model, and green regions correspond to poorly stained areas rejected by the quality control model. (c) The acceptance probability map of (a) predicted by the TransMIL model, where values close to 1 indicate acceptance, while values near 0 signify rejection. (d) The detection results of (a) using the TransMIL model, which was obtained by thresholding (c) by 0.5. (e) The acceptance probability map of (a) predicted by the AQuA model. (f) The detection results of (a) using the AQuA model, which was obtained by thresholding (e) by 0.5. (g-h) Two examples of zoomed-in tissue regions rejected by the HistoQC model, the TransMIL model and AQuA model, which contain out-of-

focus areas. (i-j) Two examples of zoomed-in regions with out-of-focus areas (arrows 1 and 2 in (i-j)), purple staining artifacts (arrow 2 in (i)) and brown exogenous pigments masking the tissue (arrow 2 in (j)), which were accepted by the HistoQC model but rejected by the AQuA model. (k-l) Two examples of zoomed-in tissue regions corresponding to air bubbles, which are correctly rejected by the AQuA model, but incorrectly accepted by the TransMIL model. (m) The WSI of a well-stained HS slide labelled by pathologists. (n) The detection results of (m) using the HistoQC model. (o) The acceptance probability map of (m) predicted by the TransMIL model. (p) The detection results of (m) using the TransMIL model, which was obtained by thresholding (o) by 0.5. (q) The acceptance probability map of (m) predicted by the AQuA model. (r) The detection results of (m) using the AQuA model, which was obtained by thresholding (q) by 0.5. (s) One example of a zoomed-in tissue region accepted by the HistoQC model, the TransMIL model and the AQuA model. (t) Another example of a zoomed-in tissue region accepted by the HistoQC model, the TransMIL model and the AQuA model. (u-v) Two examples of zoomed-in regions with good staining quality, which were rejected by the HistoQC model but accepted by the AQuA model. (w-x) Another two examples of zoomed-in regions with good staining quality, which were rejected by the TransMIL model but accepted by the AQuA model.

New Sub-sections Added

- **New Results 2nd paragraph under the section “Autonomous staining quality assessment on HS images”:** Generalization of the AQuA model to the TCGA image dataset.
- **Extensively revised Discussion 5th paragraph:** WSI-level quality control detection comparisons between the HistoQC model, the TransMIL model and the AQuA model.
- **New Discussion (8th) paragraph:** Diagnostic analysis using VS outputs from two poor-staining VS models and a good-staining VS model against their HS ground truth.
- **New Methods sub-section:** TCGA Dataset
- **New Methods sub-section:** Division of good-staining and poor-staining histochemical WSIs
- **New paragraph in Supplementary Note 1:** Additional analysis on Grad-CAM visualization results

Reviewer #1:

I am glad to see the improvement of the manuscript based on the comments from the last round. The revision is of high quality. Most of the comments have been appropriately addressed by the authors. We thank the reviewer for helping us improve the quality of our manuscript.

The only outstanding question I still have is related to comment 3 from the last round, which was not fully addressed. Although I understand the focus of the work is on the quality check of the VS or HS images, it is still meaningful to investigate whether the hallucinated images cause any wrong diagnoses. I suggest the authors perform one additional quality assessment with the board-certified pathologists, confirming whether they will make the correct/incorrect diagnostic outcome based on the hallucinated images (with respect to the HS ground truth). If the hallucinated images cause wrong diagnoses by board-certified pathologists, which can be detected by the AQUA, the value of the work will be further strengthened.

Thanks for the reviewer's constructive suggestion. Following the referee's suggestions, we have added **Extended Data Fig. 12** and supplementary analysis corresponding to the images from two poor-staining VS models identified by the AQUA workflow and one good-staining VS model. The results of this analysis along with the new **Extended Data Fig. 12** were also discussed in the Discussion section, quoted: *"...To further elaborate on how the poor-quality VS models identified by our AQUA workflow can affect the diagnostic process, we conducted an additional analysis using a case study of a patient with lung transplant rejection. The results are demonstrated in **Extended Data Fig. 12**, where we compared the outputs from two poor-staining VS models and one good-staining VS model against their HS references. From the HS images (**Extended Data Fig. 12(d, h, l)**), a prominent mononuclear cell infiltrate was seen, suggesting acute cellular rejection. However, the first poor-staining VS model, as automatically identified by our AQUA model, generated images with artifacts, such as aberrant color distributions and blurred nuclear boundaries (see **Extended Data Figure 12(a, e, i)**), making the images unsuitable for pathological review. Despite appearing to have good staining quality perceptually, the second poor-staining VS model introduced realistic hallucinations by missing many lymphocyte nuclei. This deficiency, shown in **Extended Data Figure 12(b, f, j)**, could mislead pathologists to mistakenly assess the histological picture of this patient as "non-rejection" rather than "rejection." In contrast, images produced by the good-staining VS model, correctly identified by the AQUA workflow (illustrated in **Extended Data Figure 12(c, g, k)**), aligned closely with the HS ground truths, leading to consistent diagnostic outcomes between the VS and HS images. These comparisons underscore the critical importance of AQUA in automatically identifying hallucinations in the VS process, which can potentially mislead the diagnostic process by presenting pathologists with incorrect tissue characteristics."*

Reviewer #2:

Here are comments for the revised manuscript:

1. Generally, the evaluation dataset is quite limited. The inclusion of an external validation set, such as images from TCGA, would be beneficial to validate the generalizability of the proposed framework. We appreciate the reviewer's suggestion regarding the TCGA dataset. To further demonstrate the adaptability of our method and its generalization to large datasets with high variations, we transferred the pre-trained AQUA previously demonstrated on HS lung images using only 10 good-quality HS images from the TCGA dataset to adapt to various H&E styles from 49 different tissue source sites, and then

blindly tested its generalization on an extensive test set of both good- and bad-quality TCGA HS images. This additional experiment is summarized in **Extended Data Fig. 5** and analyzed in the Results section, quoted below:

*“...To further demonstrate the adaptability of AQuA to handle an even larger variation of HS image distributions, we extended this pre-trained AQuA model to The Cancer Genome Atlas Program (TCGA) image dataset³⁶. A total of 461 whole slide images (WSIs) of human lung tissue collected by 49 distinct tissue processing sites within the TCGA dataset were labelled by a board-certified pathologist to identify the presence of artifacts. Using transfer learning, we finetuned the pre-trained AQuA model using only 10 well-stained WSIs (8 for training and 2 for validation) and blindly tested it on a separate set consisting of 989 HS images (519 good and 470 bad HS images) sampled from the remaining 451 TCGA WSIs, which were excluded from the transfer learning; see the Methods section for details. Despite encountering various forms of new artifact modes in the TCGA dataset never seen before (**Extended Data Fig. 5(a)**), the AQuA model successfully discriminated bad-quality HS images, scoring an overall accuracy of 87.6% on the test set, as shown in **Extended Data Fig. 5(b, c)**. **These results highlight the external generalization ability of AQuA to various unseen patterns of artifacts and its robustness on new testing data distributions from various tissue processing sites that exhibit several modes of HS artifacts never seen before.**”*

This experiment strongly demonstrates the generalization and robustness of AQuA to various unseen artifacts in HS images resulting from multiple sources, including different tissue source sites, sample preparation protocols and imaging hardware. This point is further emphasized in the Discussion section, quoted below:

*“...As another example, **Extended Data Fig. 5** demonstrates AQuA’s external generalization on various unseen types of artifacts that appeared in HS images with varying tissue sources, different sample preparation protocols and different imaging hardware from tens of different sites.”*

Furthermore, the details of these TCGA-related results and analyses can be found in the newly added subsection “**TCGA dataset**” in the Methods section, quoted:

“TCGA dataset

*677 human lung tissue WSIs from 57 tissue source sites collected from the TCGA dataset³⁶ were labelled by a board-certified pathologist, and 395 good and 66 bad WSIs (i.e., 461 WSIs in total) from 49 distinct source sites were determined suitable for this experiment, while the other WSIs contained artifacts not overlapping with large tissue areas and were therefore labelled “ambiguous” and not used. Good WSIs had consistently good quality and no apparent artifacts. Bad WSIs contained artifacts overlapping with the tissue areas resulting from e.g., defects of prepared slides, deblurring and artificial markers. Each WSI was randomly cropped into multiple 2048×2048 pixel patches, and each patch was then checked by a pathologist to have a consistent label (good/bad) with its source WSI. During the transfer learning, two small subsets including 36 patches from 8 good WSIs, and 8 patches from 2 good WSIs were used as the negative samples for training and validation, respectively – i.e., no bad WSIs were used in the transfer learning step. The rest of the 989 patches (519 good and 490 bad patches) generated from 385 good WSIs and 66 bad WSIs were used for blind testing. Since no bad WSIs from the TCGA dataset were used during transfer learning, we were able to test the external generalization ability of AQuA to various unseen patterns of HS artifacts from multiple tissue sites, as reported in **Extended Data Fig. 5.**”*

2. More experiments are needed on the selection of thresholds used to differentiate between good-stain and bad-stain models. It's worth investigating the impact of different thresholds on classification accuracy.

We thank the reviewer for this constructive comment. We have added the **receiver operating characteristic (ROC) curve** and **precision-recall curve (PRC)** for AQuA models on human kidney and lung samples and M-AQuA model on human lung samples to comprehensively show the impact of threshold to classification performance. **These results demonstrate excellent performance with the area under the curve approaching 1.0, and are summarized in the new Supplementary Figure 1** and analyzed in the Results section, quoted:

“...AQuA’s performance at varying threshold values is demonstrated using the receiver operating characteristic (ROC) curve and the precision-recall curve (PRC) reported in Supplementary Fig. 1(a). The area under curve (AUC) scores ~1.0 for each curve, indicating the strong classification performance of AQuA...

...For lung tissue samples, AQuA’s good classification performance at varying threshold values is also reported in Supplementary Fig. 1(b)...

...The PRC and ROC curves of M-AQuA are illustrated in Supplementary Fig. 1(c), where the high AUC values for both curves confirm M-AQuA’s successful performance at varying levels of threshold...”

3. The manuscript lacks detailed explanation on how histochemically stained images are classified as 'positive' or 'bad'.

In our study, whether a histochemically stained WSI is of high quality or poor quality was determined based on pathologists’ evaluations. The detailed procedure was described in a newly added subsection **“Division of good-staining and poor-staining histochemical WSIs”** in the Methods section, quoted:

Division of good-staining and poor-staining histochemical WSIs

To determine whether a histochemically stained WSI is of good quality or bad quality, three board-certified pathologists reviewed and assessed each WSI from three aspects: nuclear detail, cytoplasmic detail, and extracellular detail. For each aspect, the pathologists were requested to assign a score to every WSI on a scale from 1 and 4, where 4 indicates “perfect” quality, 3 denotes “very good”, 2 corresponds to “acceptable”, and 1 represents “unacceptable” quality. The final quality score for each WSI was calculated by averaging the scores across all three aspects from all three pathologists. WSIs with an average score above 3 are classified as high-quality (good-staining) histochemical stains, whereas those with an average score below 2 are classified as low-quality (bad-staining).

”

4. In the task of histochemical stain (HS) evaluation, additional comparisons between the use of ResNet and Transformer would be beneficial.

In the manuscript, we have already compared AQuA with TransMIL and performed an ablation study on AQuA’s performance with respect to various hyperparameters (ensemble number C and the number of iterations T). **We would like to kindly emphasize that TransMIL is a transformer-based pathology image classification model; and AQuA ($T = 1$) is already equivalent to a ResNet baseline – therefore, we had already addressed this comment in our previous revisions.**

The comparison between AQuA and TransMIL is summarized in the Discussion section and **Extended Data Figure 9**, quoted:

*“...When comparing AQuA with TransMIL, both of the models shared the same pre-trained ResNet-50 backbone⁵⁰ for feature extraction, and were trained on the same dataset of human lung HS images employing the same majority voting mechanism ($C = 5$). **Extended Data Figure 9** summarizes the comparison of these two models on the same testing set of human lung HS images and reports the corresponding confusion matrices; **the results show that TransMIL partially failed to separate the positive and negative populations with a low accuracy of 68.2%, whereas AQuA provides clear separation between the two populations and scored 100% accuracy. Additionally, the total number of trainable parameters of AQuA (~2.1 M) is less than TransMIL (~3.2 M), showcasing AQuA’s efficiency.**”*

Additionally, in the revised **Extended Data Figure 10** and the newly added **Supplementary Fig. 4**, we also included TransMIL and compared AQuA, TransMIL and HistoQC at the whole slide level. These results are analyzed in the Discussion section, quoted below:

*“...In addition to these, when comparing AQuA with HistoQC and TransMIL at the WSI-level, AQuA demonstrated a more robust concordance with pathologists' judgments in assessing the quality of the slide staining. **Extended Data Figure 10** illustrates the quality control detection results from the AQuA, HistoQC and TransMIL models for a poorly stained HS slide and a well-stained HS slide, as labeled by pathologists. **As depicted in Extended Data Fig. 10(a-f), AQuA consistently rejected all the regions from the poorly stained HS slide, while HistoQC and TransMIL only partially rejected some of these poorly stained regions. Although HistoQC and TransMIL successfully identified and rejected severely out-of-focus areas (Extended Data Fig. 10(g-h)), they both failed to accurately detect regions lacking hematoxylin staining, exhibiting small blurred areas, containing detached tissue fragments, and obscured lung architecture due to the presence of excessive RBCs, as shown in Extended Data Fig. 10(i-l).** As for the well-stained HS slide shown in Extended Data Fig. 10(m-r), all three models, AQuA, HistoQC, and TransMIL, agreed on accepting most of the tissue regions, as desired. **However, HistoQC erroneously rejected some regions with high-quality tissue staining (Extended Data Fig. 10(u-v)), while AQuA accurately accepted and preserved these stained areas. TransMIL also erroneously rejected several regions with high-quality staining, as depicted in Extended Data Fig. 10(w-x), whereas AQuA accurately accepted these regions. Additional examples from another poorly stained HS slide as well as another well-stained HS slide, labeled by pathologists, are illustrated in Supplementary Fig. 4. Here, AQuA was able to accurately reject poor-staining regions aligning with the pathologists' assessments as shown in Supplementary Fig. 4(a-f). However, HistoQC incorrectly accepted regions that contained out-of-focus areas, staining artifacts and exogenous pigments, as shown in Supplementary Fig. 4(i-j), and TransMIL inaccurately accepted regions with air bubbles in Supplementary Fig. 4(k-l). Furthermore, both HistoQC and TransMIL erroneously rejected several regions that exhibited normal tissue structures and high staining quality, as depicted in Supplementary Fig. 4(m-r), which AQuA correctly identified and accepted.**”*

Furthermore, AQuA ($T = 1$) is already equivalent to a **ResNet** baseline whose input is only 2D VS and AF images in comparison to the 3D spatial-temporal image sequence of AQuA ($T > 1$).

The ablation study on hyperparameter T is analyzed and summarized in the Discussion section, **Extended Data Fig. 6**, and **Extended Data Table 1**. We further modified the related paragraphs to emphasize the equivalence between AQuA ($T = 1$) and a ResNet baseline, quoted:

*“...The success of AQuA is closely linked to the novel design of AQuA-Net. First, by leveraging successive iterations between the VS and VAF models, inference uncertainty within the VS image gradually accumulates and therefore becomes easier to accurately detect. The advantages of these VS-AF iterations are studied in additional ablation studies performed on the hyperparameter T , as summarized in **Extended Data Fig. 6**. Compared to the AQuA-Net performance without such VS-AF iterations ($T = 1$), which is equivalent to a **ResNet** baseline, the introduction of these iterations significantly improves both the classification accuracy and the KL divergence between the positive and negative VS image distributions.”*

Additionally, the effectiveness of spatial-temporal input of AQuA ($T > 1$) was also validated through the Grad-CAM visualization summarized in the Discussion section, **Extended Data Fig. 7** and **Supplementary Note 1**, quoted:

*“...The effectiveness of VS-AF iterations can be further confirmed by the spatial-temporal Grad-CAM^{45,46} activation maps, as illustrated in **Extended Data Fig. 7**, **Supplementary Figs. 2-3**, and analyzed in **Supplementary Note 1**. Leveraging the spatial-temporal features in the VS image sequence, the time-averaged activation map of AQuA effectively captures and locates modes of abnormality and inconsistency in the original VS images ($t=0$), including missing tissue regions, areas with high background noise, and residual tissue areas. As detailed in **Supplementary Note 1**, AQuA’s activation maps successfully emphasize relatively large inconsistent regions that are also evidenced by the difference map between the VS and HS images (**Extended Data Fig. 7(d)**), as well as some of the smaller abnormal features that are undetectable by traditional pixel-wise difference maps (see **Extended Data Fig. 7(e, f)**). This comparison between AQuA’s activation map and the HS reference image-based difference map further reveals the difficulty of VS image quality and hallucination assessment task even with the use of an HS reference image; this highlights the merits of AQuA as an autonomous hallucination and artifact assessment tool without the need for ground truth HS images.”*

5. Given the varying significance of different iterations, a straightforward average of all heatmaps (from various iterations) could potentially generate inaccurate attention maps.

First, we would like to kindly point out that Grad-CAM activation map at each iteration (time step), denoted as Grad-CAM($t=0,1,2,3$) is also shown in **Extended Data Fig. 7**, supporting the observations and conclusions in **Supplementary Note 1**. For example, the significant abnormalities pointed by **Extended Data Figure 7(d, e, f)** are also highlighted in Grad-CAM($t=1$), ($t=0$) and ($t=2$), respectively.

Besides, we would like to clarify that Grad-CAM assigns the weight of each activation map based on their gradients and then averages all weighted activation maps from a set of layers (Ref. 44, 45). **The visualization of both individual heatmaps at each time step and the averaged heatmap in Extended Data Figure 7 already reflect the weighting.** To avoid potential ambiguity, we have revised the “**Grad-CAM visualization**” subsection in the Methods section, quoted:

“Gradient-based weights were applied to the heatmap at each time step t for the visualization and the

generation of the averaged heatmap.”

6. The manuscript could provide more detailed information about the number of positive and negative stains in the test sets for the evaluation of VS and HS.

The numbers of positive and negative samples were already detailed in our previous revised manuscript in **Figs. 2-5, Fig. 7, and Extended Data Figs. 8, 9, and 11**, wherever a box plot or confusion matrix was shown.

To further emphasize these numbers, we revised the “**Dataset preparation and splitting of AQuA**” subsection in the Methods section, quoted:

“...The testing dataset was generated by 7 good-staining and 6 poor-staining VS models excluded from training, consisting of 988 VS images (456 positives, 532 negatives)...

...The testing dataset was generated by 5 good-staining and 5 poor-staining VS models excluded from training and contains 2,412 VS images (1,072 positives, 1,340 negatives)...

...The histochemically stained H&E image dataset of human lung tissue sections consists of a subset of 100 non-overlapping FOVs (each 2048×2048 pixels) for training, 16 non-overlapping FOVs for validation, and 220 non-overlapping FOVs for testing (110 positives, 110 negatives)...”

7. Generalization and Robustness with Limited Data. The issue of generalization and robustness due dataset has not been adequately addressed. Experiments conducted with such a small sample size (not pixel count) make the test results for kidneys and lungs less meaningful for assessing robustness and generalization of the method. While transfer-based validation suggests some level of generalization, overfitting is also a potential concern due to the limited amount of data.

The generalization and robustness of AQuA have been extensively studied in our manuscript. We implemented multiple internal and external generalization tests for AQuA, including generalization to unseen VS failure modes and unseen organ types, and confirmed that AQuA does not present an overfitting issue. AQuA exhibits good generalization and robustness in all test cases.

To further support these claims, we summarize below and list all the facts from our manuscript:

(1) AQuA is an extremely light-weight image classification model that takes in VS and AF images on a single image patch, rather than a WSI. And each diffraction-limited pixel contains unique information and serves for training. Therefore, the size of training and testing datasets must be measured in terms of the number of non-overlapping FOVs and the number of diffraction-limited pixels. In the “**Dataset preparation and splitting of AQuA**” subsection of the Methods section, we elucidate the details of the training, validation and testing datasets for each task performed in this work, quoted:

“...For human kidney samples, a set of 1054 non-overlapping AF FOVs (each 1424×1424 pixels) were collected from 7 individuals for training and validation, and a set of 76 non-overlapping AF FOVs collected from another 3 subjects were used for testing... The training and validation dataset was generated by 5 good-staining and 5 poor-staining VS models, and each model randomly picked and stained 20% of the full set of AF images, producing ~2,100 VS images. Then the dataset was randomly divided into training and validation sets at a ratio of 9:1. The testing dataset was generated by 7 good-staining and 6 poor-staining VS models excluded from training, consisting of 988 VS images (456 positives, 532 negatives)...

For human lung samples, we collected 572 non-overlapping AF FOVs (each 1024×1024 pixels) from 7 individuals for training and validation (a subset of the training and validation data of VS models), and 268 non-overlapping AF FOVs from another subject for testing. The training and validation dataset of VS images was generated by 8 good-staining and 7 poor-staining VS models, and each model randomly picked and stained 15% of the full set of AF images, generating ~2,300 VS images. Then the dataset was randomly divided into training and validation sets at a ratio of 9:1. The testing dataset was generated by 5 good-staining and 5 poor-staining VS models excluded from training and contains 2,412 VS images (1,072 positives, 1,340 negatives)... The histochemically stained H&E image dataset of human lung tissue sections consists of a subset of 100 non-overlapping FOVs (each 2048×2048 pixels) for training, 16 non-overlapping FOVs for validation, and 220 non-overlapping FOVs for testing (110 positives, 110 negatives)...

In summary, the total number of trainable parameters (~2.1 million) is far smaller than the dataset size (~4.6 billion unique pixels). **On average, the dataset for each task contains ~0.77 billion unique pixels per class. As a reference, the gold standard in machine learning, ImageNet-1K dataset contains ~0.26 billion unique pixels per class (1,431,167 images, average dimension 469*387 pixels).**

Therefore, we believe there is no fact supporting or potentially indicating that the dataset size in this work causes overfitting or limits the generalization of AQuA.

- (2) As emphasized in the “**Sample preparation and standard histochemical H&E staining**” subsection in the Methods section, **the training and testing images were strictly collected from different patients**, quoted below:

“...The kidney samples originated from 10 unique patients who experienced a myriad of nonneoplastic kidney diseases (7 used for training and validating the VS/VAF and AQuA models, and 3 for testing), while the lung samples came from 29 unique lung transplant recipients who underwent biopsy for transplant rejection evaluation. Among these 29 unique patients, 18 cases were allocated for training and validating the VS/VAF models, 7 out of these 18 cases were chosen for training and validating the AQuA model. The rest of the 11 cases were left for testing the AQuA model (1 case for evaluating VS slides and 10 cases for assessing HS slides). Note that each unique patient corresponds to a unique WSI in our study.”

The exclusive sources of training/validation and testing sets guarantee that the testing set are sampled from a similar, but not identical distribution as the training set, and consequently eliminates the possibility of concealment of model overfitting when evaluated on the test set. In other words, if AQuA overfits to the specific WSIs or patients in training dataset, it could not generalize on the test set. In fact, AQuA scored very high accuracy on all test sets, as evidenced by the results in **Figures 2, 4 and 7**, quoted:

*“...**Figure 2c** showcases the confidence score distribution of the two populations and the confusion matrix with respect to the ground truth labels. Based on a validation set, we selected the threshold $\alpha = 0.5253$ scoring 100% sensitivity, and then blindly applied it to the test set (containing ~2,100 VS images) as shown by the green dashed line in **Fig. 2c**. Both the scatter plot and the confusion matrix*

confirm a very good classification performance provided by our method on the two populations, with an accuracy of 99.8% and a sensitivity of 99.8%.

...On the contrary, AQuA successfully discriminates the two groups of VS images as shown in **Fig. 4c**, and achieves a high accuracy of 97.8% and a sensitivity of 99.5% based on a threshold of $\alpha = 0.6263$.

... However, the application of autonomous stain quality evaluation using AQuA framework leads to successful discrimination between the well-stained histology images and those laden with artifacts. Here AQuA was trained on a set of HS images of human lung tissue samples excluded from the testing stage (see the Methods section). This demonstration once again confirms the effectiveness of AQuA framework and its advantages over existing quality assessment methods, clearly highlighting the versatility of AQuA for autonomous assessment and detection of various artifacts, appearing in either VS or HS tissue images.”

- (3) Multiple external generalization tests of AQuA using VS images were demonstrated, including generalization to unseen VS failure modes and types of tissues. We showcased these two experiments in **Extended Data Figure 2** and **Extended Data Figure 11**, respectively. The external generalizability and robustness of AQuA is extensively analyzed in the Discussion section, quoted: “...Furthermore, AQuA exhibits robust external generalization to various unseen distributions during the blind testing stage. **Extended Data Figure 2** demonstrates AQuA’s strong external generalization to unseen VS model failure patterns, where overfitted VS models are successfully detected by AQuA, which was trained only on failure modes of early stopped models. Additionally, AQuA’s robustness can be further extended to generalization on unseen tissue types. We blindly tested two AQuA models trained with human kidney (and lung) samples on VS images of human lung (and kidney) samples, respectively. Despite the distinct tissue structures of these two organs and different imaging parameters, including varying signal-to-noise ratio, AQuA successfully generalized and detected poor VS images with a sensitivity of 98.9%, as reported in **Extended Data Fig. 11(a, b)**. This strong generalization of AQuA effectively prevents low-quality VS images from a shifted data distribution passing through a pre-trained AQuA model. In addition to these, we also used transfer learning on each new tissue type to show that AQuA could achieve further improved inference performance, comparable to an AQuA model trained from scratch solely on the testing tissue type; see **Extended Data Fig. 11(c, d)**. This robustness of AQuA to new tissue types exemplifies its strong generalization, helping to avoid false negatives due to a shifted data distribution...”

These external generalizations demonstrate AQuA is highly robust and generalizable to various unseen VS image distributions, not overfitted to a specific type of tissue, a group of patients, or a training dataset.

- (4) **An additional extensive generalization experiment of AQuA using HS images of the TCGA dataset was implemented.** We transferred the AQuA model pre-trained on our HS image dataset of human lung tissue samples using only 10 well-stained HS images from the TCGA dataset. Then, the transferred model was blindly tested on a large test set consisting of ~1,000 images randomly sampled from 385 good and 66 bad WSIs from the TCGA dataset. This experiment further showcased the generalization of AQuA on a large-scale HS image dataset with high variations resulting from varying tissue sources (involving 49 different sites), sample preparation protocols and

imaging hardware, and we summarized the results in **Extended Data Fig. 5** and the Results section, quoted below:

*“...To further demonstrate the adaptability of AQuA to handle an even larger variation of HS image distributions, we extended this pre-trained AQuA model to The Cancer Genome Atlas Program (TCGA) image dataset³⁶. A total of 461 whole slide images (WSIs) of human lung tissue collected by 49 distinct tissue processing sites within the TCGA dataset were labelled by a board-certified pathologist to identify the presence of artifacts. Using transfer learning, we finetuned the pre-trained AQuA model using only 10 well-stained WSIs (8 for training and 2 for validation) and blindly tested it on a separate set consisting of 989 HS images (519 good and 470 bad HS images) sampled from the remaining 451 TCGA WSIs, which were excluded from the transfer learning; see the Methods section for details. Despite encountering various forms of new artifact modes in the TCGA dataset never seen before (**Extended Data Fig. 5(a)**), the AQuA model successfully discriminated bad-quality HS images, scoring an overall accuracy of 87.6% on the test set, as shown in **Extended Data Fig. 5(b, c)**. **These results highlight the external generalization ability of AQuA to various unseen patterns of artifacts and its robustness on new testing data distributions from various tissue processing sites that exhibit several modes of HS artifacts never seen before.**”*

These additional experiments strongly demonstrate AQuA’s generalization and robustness on a large-scale dataset with high variations and, therefore, help us further address the concern of potential overfitting.

8. Human Visual Inspection in H&E and AF Domain Iterations. The requirement for human visual inspection of models in each iteration between the H&E and AF domains raises questions about the stability of training. Does this imply that training stability is prone to issues, or does it serve as a form of supervisory information in some regard?

We believe that this comment does not belong to our manuscript, as we have never claimed this in the manuscript.

To better clarify:

As illustrated in Fig. 1, traditional HS workflow requires manual quality assessment for each stained image, which is laborious and inefficient. To address this issue, **AQuA is designed as a fully autonomous quality and hallucination evaluation tool, without any need for human intervention or supervision**. Once a VS model and a VAF model are established, **the training, validation and inference of AQuA are fully autonomous and no manual inspection or intervention is needed**.

The data pipeline and dataset preparation are detailed in the Introduction, Results and Methods section, quoted:

“...Leveraging a novel architecture design (Fig. 1b), AQuA-Net automatically detects these morphological hallucinations and low-quality VS images without the need for any ground truth information. By introducing a label-free autofluorescence (AF) based virtual tissue staining model (i.e., VS: AF \rightarrow H&E) and its corresponding reverse image transformation, i.e., virtual autofluorescence model (VAF: H&E \rightarrow AF), we employed virtual iterations between the H&E and AF domains, which were utilized as the input of AQuA-Net to autonomously detect artifacts and hallucinations in VS H&E images, without

access to the ground truth histochemical H&E images. Blindly tested on human kidney and lung tissue samples from new patients, AQuA successfully classified each VS image as having an acceptable or unacceptable stain quality with high specificity and sensitivity; it also showcased an exceptional external generalization to new types and styles of hallucinations and artifacts generated by poorly trained VS models that were never used before...

...Starting with the establishment of good- and poor-staining models, we trained one VS model using paired AF-HS images (see the Methods section) until it converged. With the convergence thresholds determined on both the validation loss and training epochs, early stopped checkpoints with fewer epochs and higher validation losses were defined as poor VS models, and the other checkpoints with more epochs and lower validation losses were regarded as good VS models..."

Architecture and training schedule of AQuA-Net

AQuA-Net starts with the T VS-AF cycles between histological stain and AF domains. Let us denote the initial AF measurement as x_0 and the given VS image as y_0 , the input sequence $(x_0, y_0, \dots, x_{T-1}, y_{T-1})$ are formulated by iteratively passing x_t and y_t through VS and VAF networks G_{VS} and G_{VAF} , respectively:

$$\begin{aligned} x_{t+1} &= G_{VAF}(y_t), t = 0, \dots, T - 2 \\ y_t &= G_{VS}(x_t), t = 1, \dots, T - 1 \end{aligned} \quad (8)$$

Here G_{VS} and G_{VAF} always use the pre-determined, fixed best checkpoints with the lowest validation loss, which are independent of (x_0, y_0) . It is worth noting that specially for the HS image datasets of human lung tissue sections, actual AF measurement is not accessible. Therefore, we shift the VAF sequence one time step backward and substitute the AF measurement with $G_{VAF}(y_0)$. In other words, AQuA-Net for HS images takes in the input sequence $(y_0, x_0, \dots, y_{T-1}, x_{T-1})$, defined as:

$$\begin{aligned} x_t &= G_{VAF}(y_t), t = 0, \dots, T - 1 \\ y_{t+1} &= G_{VS}(x_t), t = 0, \dots, T - 2 \end{aligned} \quad (9)$$

... All classifiers were trained for 75 epochs and the checkpoints with the best validation accuracy were picked for testing. During the transfer learning, the pretrained AQuA models were finetuned on the full training dataset of the target tissue with a learning rate of 10^{-4} for 75 epochs. Later, the checkpoints with the best validation accuracy were picked for testing."

In fact, even hyper-parameters, including the number of ensembles C and the number of iterations T were determined in a fully automated scanning and selection process in AQuA, as summarized in Extended Data Figure 6, Extended Data Table 1 and reported in the Discussion section: "...The success of AQuA is closely linked to the novel design of AQuA-Net. First, by leveraging successive iterations between the VS and VAF models, inference uncertainty within the VS image gradually accumulates and therefore becomes easier to accurately detect. The advantages of these VS-AF iterations are studied in additional ablation studies performed on the hyperparameter T , as summarized in Extended Data Fig. 6... Second, the use of majority voting mechanism considerably enhances the classification performance in terms of accuracy and sensitivity. As reported in Extended Data Table 1, compared to a single classifier ($C = 1$), majority voting mechanism with $C > 1$ effectively boosts both the accuracy and the sensitivity. The optimal hyperparameters T and C were determined through a grid scanning-based selection process and highlighted in Extended Data Table 1 for our experiments using human kidney and lung tissue samples."

9. Explanation of Good vs. Poor Staining Models. The explanation of good and poor staining models only describes how specific models are selected, without addressing why these choices are made or why they are appropriate. More detailed justification for the selection criteria and their correctness is needed.

Thanks for highlighting this point. We have added more details to substantiate the correctness and appropriateness of our model division method in the subsection titled "**Definition of good-staining and poor-staining models for virtual staining**," located in the Methods section, quoted:

"...Our model selection methodology relies on the validation loss metrics, including MAE, SSIM and TV, and is frequently utilized in the field of image generation and translation^{68,69}. In addition, a board-certified pathologist further verified the correctness of the model definitions."

Furthermore, for comparison purposes, whether a histochemically stained WSI is of high quality or poor quality was determined based on pathologists' evaluations. The detailed procedure was described in a newly added subsection "**Division of good-staining and poor-staining histochemical WSIs**" in the Methods section, quoted:

"

Division of good-staining and poor-staining histochemical WSIs

To determine whether a histochemically stained WSI is of good quality or bad quality, three board-certified pathologists reviewed and assessed each WSI from three aspects: nuclear detail, cytoplasmic detail, and extracellular detail. For each aspect, the pathologists were requested to assign a score to every WSI on a scale from 1 and 4, where 4 indicates "perfect" quality, 3 denotes "very good", 2 corresponds to "acceptable", and 1 represents "unacceptable" quality. The final quality score for each WSI was calculated by averaging the scores across all three aspects from all three pathologists. WSIs with an average score above 3 are classified as high-quality (good-staining) histochemical stains, whereas those with an average score below 2 are classified as low-quality (bad-staining).

"

10. Transfer of Staining Techniques Using AQuA-Net. AQuA-Net iterates images between the H&E and AF domains. Is it possible to apply staining transfers using this method? Images with different staining styles should map consistently to the AF domain. Since there is substantial validation data available for staining normalization issues, this could be a way to enhance the clinical application of the method.

We appreciate this constructive comment from the reviewer. Following the referee's suggestion, we have revised the Discussion section to explain this potential future study, quoted:

"...Additionally, the AQuA framework can be extended to some of the recent advances in stain-to-stain transformations and stain normalization methods^{25,59-61}. Through iterations between image domains of different stains, or between image domains of AF and multiple types of stains, AQuA can potentially provide autonomous quality and hallucination assessment for virtually transformed and normalized pathology images, which is especially important for cases where HS references are inaccessible."

11. Virtual stain may miss or mis-stain some diagnostic important information, i.e., nuclei. The authors should quantify the results on this essential metric.

To ensure that the virtual staining model used in our cycle inference operates at a high level of performance and does not lead to diagnostic errors, we conducted a comprehensive quantitative comparison between the virtually stained outputs and their histochemically stained ground truths. **This**

comprehensive comparison encompassed several dimensions, including color distributions, Peak Signal-to-Noise Ratio (PSNR), Structural Similarity Index (SSIM), as well as comparisons of the number and average size of nuclei. These quantitative assessments demonstrated that there are no statistically significant differences between the virtual and histochemical staining results.

Detailed findings were presented in:

- (1) **Supplementary Note 2:** Comparison between VS and HS quality;
- (2) **Supplementary Figure 5:** Color distribution comparisons of paired VS and HS kidney tissue images;
- (3) **Supplementary Figure 6:** PSNR, SSIM and structural feature comparisons of paired VS and HS kidney tissue images.

The detailed methodologies we used to perform this quantitative comparison were clarified in the subsection **“Quantitative evaluation of the virtual staining model used in VS-AF iterations”** in the Methods section.

Once again: all of these quantitative assessments demonstrated that there are no statistically significant differences between the virtual and histochemical staining results.

12. There is still issue with the evaluation method, “each pathologist was asked to give a binary flag on the overall stain quality of each VS image, indicating whether the image has acceptable or unacceptable stain quality.” The pathologists cannot judge the overall stain quality. They can hardly tell if only 8 out of 10 nuclei are properly stained, or real stain is unnecessary. Careful comparison between virtual stained and real HE stained on large-scale histology images by pathologists is almost impossible unless the dataset is small.

First, the quotation cited by the reviewer reflects a specific part of our evaluation approach, which might be misunderstood from its original meaning.

To better clarify:

In the “Pathologist evaluations” section, the complete evaluation process includes multiple evaluation metrics and discrete scores in addition to the overall binary flag (“pass” or “fail”) to provide a meticulous assessment from various perspectives and confirm the correctness of pathologist evaluation. **“pass/fail” label (binary flag) was only used to establish consensus from pathologists on the evaluation dataset and not directly used in comparison with our method.** We have revised the **“Pathologist evaluations”** subsection of the Methods section to further clarify this process, quoted below:

“...Besides, each pathologist was asked to give a pass/fail label, indicating whether the image has acceptable or unacceptable stain quality. Consensus among pathologists were established based on the pass/fail labels such that on the FOVs with consensus, the diagnosticians unanimously agreed on the overall pass/fail label assessment...”

Secondly, one of the main contributions of our work is the development of AQuA as a fully autonomous quality and hallucination assessment tool without the need for HS references. In the pathologist evaluation stage of this work, we mimic this scenario and ask pathologists to evaluate stain quality without HS ground truths, as this is an important objective of our work. **This reference-free**

evaluation is the same for both pathologists and AQuA, rendering a fair comparison. The advantages of AQuA over pathologists on various VS images without HS references are reported in the Results section, quoted below:

“...In addition to these comparisons against traditional structural HS-based metrics, we further assessed the performance of our method against the decisions of three board-certified pathologists, covering the condition that no histochemical reference images are accessible. Each pathologist independently and blindly evaluated the quality of a set of VS images of human kidney samples. A consensus was reached on $N = 127$ images, among which $N_G = 66$ images came from good-staining VS models and $N_P = 61$ images from poor-staining VS models. Specifically, the pathologists were asked to score each image on three metrics: the stain quality of nuclei, cytoplasm and extracellular space from 1 to 4, with higher scores indicating better quality (refer to the evaluation details in the Methods section). Figure 3a reports the overall agreements between our method’s confidence scores and the pathologists’ scores on all $N = 127$ test images. Here, the range for the average score over three pathologists for an acceptable stain quality was set as (2,4]. Figure 3a reveals that our method agrees well with the pathologists’ opinion (average stain quality score), corresponding to the blue and green areas in the confusion matrices shown in Fig. 3a. We further break down Fig. 3a into Figs. 3b and 3c, comparing our method against the pathologist scores on the good-staining and poor-staining VS models, respectively. As illustrated in Fig. 3b, AQuA achieves 100% acceptance rate without any human intervention or supervision; in other words, all VS images from good-staining VS models are accepted. As a reference, the pathologists agreed on 98.5% of the classification of our method (green zones) over the three separate stain quality scores and the average stain quality score. In Fig. 3c, when detecting hallucinated VS images coming from poor-staining models, AQuA scores 100% rejection rate without any human supervision or access to HS ground truth images. In contrast, when presented with these VS images featuring realistic hallucinations (yellow zones), pathologists failed to flag these hallucinated images and could not identify the mismatch between VS images and their HS counterparts when the access to HS ground truth was unavailable...”

A blinded comparison with pathologists’ scores was also implemented on human lung tissue samples, following the previously outlined methodology. On a set of $N = 99$ testing VS images, where pathologists reached consensus, our method shows a good agreement with their scoring as shown in Fig. 5a. Figure 5b and 5c illustrate the breakdown of Fig. 5a into the two cases on good-staining and poor-staining VS models, respectively. As shown in Fig. 5b, AQuA correctly identifies 100% of the negative VS images generated by good-staining models ($N_G = 59$), on which board-certified pathologists also accepted 98.3% of the same VS images and rejected only one disagreement case consisting almost entirely of red blood cells (RBCs). The disagreement case is further shown in Extended Data Fig. 1b and analyzed in the Discussion section. On the poor-staining VS models with $N_P = 40$ images, AQuA successfully rejected all hallucinatory images generated by these poor-staining models, including unrealistic (blue zone) and realistic (yellow zone) hallucinations. In contrast, board-certified pathologists failed to identify the VS images generated by poor-staining models harboring realistic hallucinations in the yellow zone. An example of these realistic hallucinations is shown in Extended Data Fig. 1a. An additional side-by-side comparison between these realistic hallucinations and the corresponding HS images is illustrated in Extended Data Fig. 3, showing a notable mismatch of nuclei in renal tubules and interstitium in the VS images that are inconspicuous without HS reference images. These comparisons further confirm the super-human performance of AQuA’s autonomous assessment of staining quality

and hallucinations of VS models without access to ground truth HS images.”

Reviewer #3

Thank you for addressing several of my previous issues raised. Still, some ambiguity remains that would need further clarification:

1) Even if the study is designed to only target kidney and lung histology, it is essential to demonstrate the diagnosis of the cases. The authors need to provide data about the WSIs that were used to train and test the model and their respective diagnoses to see the generalizability of the AQUA results in different disease geometries. It is also important to describe how the patches were selected, were these healthy, diseased, etc.

We thank the reviewer for raising this important point. WSI information is provided in the “**Sample preparation and standard histochemical H&E staining**” subsection in the Methods section. The training, validation and testing image patches were randomly cropped from the tissue regions in these WSIs, and the dataset preparation process is elucidated in the “**Dataset preparation for training VS and VAF networks**” subsection of the Methods section. However, we need to point out that the diagnosis labels cannot be established for each individual patch because of the limited area of each patch (~100*100um²) and the inherent tissue heterogeneity.

2) Thinking about the pathology workflow – if the diagnosis for a slide is “disease X” on the chemically stained tissue, does this align with the pathologist’s diagnosis made on the virtual stain? This would be more relevant on the slide level. It is also interesting if the artificial stains from the so-called “low-quality artificial stain generator” output would also have the same diagnosis.

3) If according to AQUA a slide includes patches with hallucinations, how do these hallucinations affect the diagnosis of a pathologist from the artificial stain?

We will respond to these comments (2) and (3) together.

To demonstrate how the poor-staining VS models might affect the diagnostic process, we have added a new **Extended Data Fig. 12** to show the diagnostic comparisons between the images from two poor-staining VS models identified by the AQUA workflow and one good-staining VS model. The results of this analysis and **Extended Data Fig. 12** were discussed in the Discussion section of our revised manuscript, quoted:

*“...To further elaborate on how the poor-quality VS models identified by our AQUA workflow can affect the diagnostic process, we conducted an additional analysis using a case study of a patient with lung transplant rejection. The results are demonstrated in **Extended Data Fig. 12**, where we compared the outputs from two poor-staining VS models and one good-staining VS model against their HS references. From the HS images (**Extended Data Fig. 12(d, h, l)**), a prominent mononuclear cell infiltrate was seen, suggesting acute cellular rejection. However, the first poor-staining VS model, as automatically identified by our AQUA model, generated images with artifacts, such as aberrant color distributions and blurred nuclear boundaries (see **Extended Data Figure 12(a, e, i)**), making the images unsuitable for pathological review. Despite appearing to have good staining quality perceptually, the second poor-staining VS model introduced realistic hallucinations by missing many lymphocyte nuclei. This deficiency, shown in **Extended Data Figure 12(b, f, j)**, could mislead pathologists to mistakenly assess the histological picture of this patient as “non-rejection” rather than “rejection.” In contrast, images produced by the good-staining VS model, correctly identified by the AQUA workflow (illustrated in **Extended Data Figure 12(c, g, k)**), aligned closely with the HS ground truths, leading to consistent*

*diagnostic outcomes between the VS and HS images. **These comparisons underscore the critical importance of AQUA in automatically identifying hallucinations in the VS process, which can potentially mislead the diagnostic process by presenting pathologists with incorrect tissue characteristics.***

4) How can the information from image patch labels received from AQUA be transferred to the slide label? If the recommended approach is to exclude the patches that involve hallucinations, it could make the diagnosis impossible or inaccurate (i.e., what if the part that is essential for the diagnosis of the WSI lies in the patches that AQUA suggests excluding?) How would such limitation affect AQUA and virtual stains in routine pathology?

We thank the reviewer for this constructive point.

In this work, we focused on the following 2 important tasks:

- (1) **Image level autonomous quality assessment of virtual staining where the AQUA framework automatically assesses the VS quality at the image patch level.** This scenario broadly applies to cases where already approved and rigorously validated VS models routinely and continuously generate VS images on new patient specimens, eliminating the standard HS workflow. Therefore, in these cases, AQUA serves as an autonomous and unsupervised “watchdog” to detect when the VS images should not be used, even if the VS model has already been validated and approved for a given workflow. If there are patches of a WSI that AQUA framework detects as hallucinations, the label-free sample needs to be re-scanned using an AF scanning microscope - ideally with slower scanning speed and higher signal to noise ratio to ensure improved image quality. **Since VS technique is non-destructive to label-free tissue, repeating the AF image scanning process does not cause an issue or any significant delays, and it might eliminate potential sources of hallucinations that might randomly appear due to scanning artifacts, out of focus regions, microscopist errors, misalignments, etc.**
- (2) **Model-level autonomous quality assessment of virtual staining.** Another important scenario that we considered in our manuscript is the VS model pre-approval process; here we addressed the question of should we accept a given VS model into our workflow or not. To answer this question, we created an autonomous method for VS Model quality assessment, termed M-AQUA – which is also based on the AQUA framework. In practice, the performance of VS models, even if they are pre-approved, should be periodically evaluated, as part of a quality assurance routine, for generalization failures, which may result from a variety of causes including e.g., changes in imaging and sample preparation protocols, replacement or aberrations of imaging hardware as well as variations in biological content and storage conditions of tissue, among other factors. Hence, the periodic assessment of each VS model is necessary to assure the quality of the whole VS workflow on top of autonomous quality monitoring of each VS image. Of course, the use of M-AQUA is supposed to be much less frequent compared to image-level VS quality assessments performed through AQUA as they serve two different functions; the former (M-AQUA) decides on the quality of the VS model, whether e.g., it should be approved for use or trained with more data; the latter (AQUA), however, decides if the generated VS image should be used for expert examination or not.

In general, from the perspective of pathologists, hallucinatory patches rejected by AQUA should be excluded from the diagnosis process. However, since the importance of each specific patch to WSI diagnosis heavily depends on the semantic features within that patch and the other surrounding

patches, its influence on the WSI diagnosis result cannot be easily quantified or objectively measured. There are a few studies exploiting neural networks to automatically evaluate the importance of an image patch with respect to the WSI, such as [Jaume et al. Multistain Pretraining for Slide Representation Learning in Pathology, 2024]; however, these studies fall outside the scope of our work.

5) The patch that is used for grad cam is heavily background dominant, please add more images, especially high-resolution images with full tissue.

Following the referee's comments, we have added **new Supplementary Figures 2 and 3**, and another two examples showing the Grad-CAM visualization results. Note that since the image resolution of AQUA is fixed to 1424*1424 (kidney) and 1024*1024 (lung) throughout the training and testing and the Grad-CAM activation value is normalized on a patch basis, simply stitching Grad-CAM heatmaps from multiple FOVs would be incorrect and misleading.

6) Regarding the comparison of TransMIL vs AQUA, please provide more confusion matrices to show in which types of hallucinations AQUA outperforms TransMIL.

We would like to clarify that both TransMIL and AQUA were tested on HS images, which means there are no "unrealistic" or "realistic" hallucinations as these are HS stained slides.

Besides, it is impossible to exclusively classify a single HS image into a single type of poorly stained HS slides as many low-quality patterns are mixed and concurrent, as demonstrated in **Extended Data Figure 4**. For example, **Extended Data Figure 4(b)** presents a FOV where black asterisks mark out-of-focus, blurry regions while such regions also lack nuclei staining (hematoxylin).

7) Regarding the labeling of the slide, TransMIL can have the WSI as one bag and therefore identify if the WSI is acceptable or not. If AQUA would be able to provide a method for a decision on slide level, would it still outperform TransMIL in giving a label (good quality or bad quality stain) to the WSI?

Since many histochemical image quality control tools, including AQUA currently process images on a patch-by-patch basis, we conducted a fair comparison by processing WSIs in a scanning manner, feeding patches into AQUA, HistoQC and TransMIL models. Each method generates a confidence score map over the WSI.

The results, which demonstrate AQUA's superior performance compared with HistoQC and TransMIL, are shown in the revised **Extended Data Fig. 9, the revised Extended Data Fig. 10 and the new Supplementary Fig. 4**. A detailed analysis of these findings is also included in the Discussion section, quoted:

*"...To further demonstrate the effectiveness of AQUA, we also compared it against other HS slide quality control algorithms, including (1) a transformer-based architecture, TransMIL⁴⁷; and (2) an open-source quality control tool named HistoQC^{48,49} designed for digital pathology slides. When comparing AQUA with TransMIL, both of the models shared the same pre-trained ResNet-50 backbone⁵⁰ for feature extraction, and were trained on the same dataset of human lung HS images employing the same majority voting mechanism ($C = 5$). Extended Data Figure 9 summarizes the comparison of these two models on the same testing set of human lung HS images and reports the corresponding confusion matrices; **the results show that TransMIL partially failed to separate the positive and negative populations with a low accuracy of 68.2%, whereas AQUA provides clear separation between the two populations and scored 100% accuracy. Additionally, the total number of trainable parameters of AQUA (~2.1 M) is less than TransMIL (~3.2 M), showcasing AQUA's efficiency. In addition to these,***

when comparing AQuA with HistoQC and TransMIL at the WSI-level, AQuA demonstrated a more robust concordance with pathologists' judgments in assessing the quality of the slide staining. Extended Data Figure 10 illustrates the quality control detection results from the AQuA, HistoQC and TransMIL models for a poorly stained HS slide and a well-stained HS slide, as labeled by pathologists. As depicted in Extended Data Fig. 10(a-f), AQuA consistently rejected all the regions from the poorly stained HS slide, while HistoQC and TransMIL only partially rejected some of these poorly stained regions. Although HistoQC and TransMIL successfully identified and rejected severely out-of-focus areas (Extended Data Fig. 10(g-h)), they both failed to accurately detect regions lacking hematoxylin staining, exhibiting small blurred areas, containing detached tissue fragments, and obscured lung architecture due to the presence of excessive RBCs, as shown in Extended Data Fig. 10(i-l). As for the well-stained HS slide shown in Extended Data Fig. 10(m-r), all three models, AQuA, HistoQC, and TransMIL, agreed on accepting most of the tissue regions, as desired. However, HistoQC erroneously rejected some regions with high-quality tissue staining (Extended Data Fig. 10(u-v)), while AQuA accurately accepted and preserved these stained areas. TransMIL also erroneously rejected several regions with high-quality staining, as depicted in Extended Data Fig. 10(w-x), whereas AQuA accurately accepted these regions. Additional examples from another poorly stained HS slide as well as another well-stained HS slide, labeled by pathologists, are illustrated in Supplementary Fig. 4. Here, AQuA was able to accurately reject poor-staining regions aligning with the pathologists' assessments as shown in Supplementary Fig. 4(a-f). However, HistoQC incorrectly accepted regions that contained out-of-focus areas, staining artifacts and exogenous pigments, as shown in Supplementary Fig. 4(i-j), and TransMIL inaccurately accepted regions with air bubbles in Supplementary Fig. 4(k-l). Furthermore, both HistoQC and TransMIL erroneously rejected several regions that exhibited normal tissue structures and high staining quality, as depicted in Supplementary Fig. 4(m-r), which AQuA correctly identified and accepted.”

Minor:

1) Names of figures are missing from Extended Data Figure 6 where are parts e, f, etc?

(e, f) are arrows pointed in 6(a). We have modified the captions of Extended Data Figure 7 (previous Extended Data Fig. 6) to avoid confusion, quoted:

“...(d, e, f) are pointed by arrows in (a)...”

2) The workflow depends on fluorescent scanning of the unstained tissue, this needs to be considered when judging the overall suitability and efficacy.

This is an important point that the referee is bringing up. Please note that virtual staining does not involve the labor-intensive process of fluorescent labeling. Instead, we utilize the **auto-fluorescence properties** inherent in **label-free** human tissues as the input information, leveraging their endogenous emission mechanisms. Therefore, no labeling was applied to the unstained tissues involved in this work.

Given that (auto)fluorescence imaging modules are already integrated into clinical, FDA-cleared tissue scanners from leading manufacturers such as Leica, Zeiss, and Olympus, etc. adapting our virtual staining and AQuA workflow to the current histological assessment processes can be readily achieved, benefiting from the existing digital pathology infrastructure. **From this perspective, the digitization of the tissue sample using the AF channel as opposed to the brightfield channel of a digital pathology scanner does not add extra complexity or delays to the workflow.**

Response Letter:

We briefly summarize below the main additions made in the revised manuscript, following the reviewers' feedback. We then address, point by point, the specific comments that our reviewers raised. Throughout this letter, the original referee comments are shown in black color, whereas for ease of communication, our answers are provided in blue. Our revisions have also been marked in the main text and supplementary information files using yellow highlighting.

Summary of the revisions:

- A detailed analysis regarding nuclei counts of good VS and HS images has been added. The level of change in nuclei count of VS images compared to their HS counterparts is smaller than the variability observed between two sequential cuts of the same tissue block, which arises from inherent tissue heterogeneity. This analysis is reported in:
 - Revised Discussion section
 - **The new Supplementary Figure 5:** Quantification of nuclei count differences between VS and HS images, as well as between serially sectioned adjacent HS tissue cuts stained with HE, MT, and EVG.
 - **The new Supplementary Note 2:** Analysis of nuclei count differences between VS and HS images, as well as between serially sectioned adjacent HS tissue cuts.
- For the two FOVs included in Extended Data Figure 12, the confidence scores predicted by the AQuA model (added in the existing Extended Data Figure 12) and the TransMIL model, along with the corresponding Grad-CAM results, are presented in our revised manuscript:
 - Revision to the current **Extended Data Figure 12**
- The results of testing HistoQC on the TCGA dataset are compared with those achieved by our AQuA model.
 - Revision to the Discussion section
- AQuA's performance on the TCGA dataset before transfer learning was added and compared with its performance after transfer learning (demonstrated in existing Extended Data Figure 5), emphasizing the necessity and impact of transfer learning:
 - Revision to the Methods section
- **The new Extended Data Figure 13:** Autonomous quality assessment of AQuA model before and after transfer learning on a subset of good HS images from the TCGA dataset.
- Details of optimal VS model selection were added in the revised Methods section.

In summary, the following new figures, new supplementary notes and new sub-sections have been added to our revised manuscript:

New Figures Added

- **New Extended Data Figure 13:** Autonomous quality assessment of AQuA model before and after transfer learning on a subset of good HS images from the TCGA dataset. (a) Boxplot of confidence scores given by AQuA after transfer learning. (b) Confusion matrix of AQuA's testing result on TCGA dataset after transfer learning. (c) Same as (a), but using the AQuA model before transfer learning. (d) Same as (b), but using the AQuA model before transfer learning.
- **New Supplementary Figure 5:** Quantification of nuclei count differences between VS and HS images, as well as between serially sectioned adjacent HS tissue cuts stained with HE, MT, and EVG. (a) Reference HE-stained HS image. (b-d) VS images generated by a bad VS model with non-realistic hallucinations, a bad VS model with realistic hallucinations, and a good VS model, respectively. (e) HS image from an adjacent tissue section stained with MT. (f) HS image from an adjacent tissue section stained with EVG. (g) Plot of nuclei counts for images (a-f), demonstrating that the differences between good VS and HS images are comparatively smaller than the differences observed between HS and its adjacent tissue cuts. (h-n) Same as in (a-g) but for a second FOV.

New Sub-section Added

- **New Supplementary Note 2:** Analysis of nuclei count differences between VS and HS images, as well as between serially sectioned adjacent HS tissue cuts.

In addition to the above revisions, we would like to express our gratitude to all reviewers for their constructive feedback, which has been invaluable in improving our manuscript.

Reviewer #1:

I am glad to see the authors have improved their manuscript further by taking my suggestions and showing the comparison of images from the poor-staining VS model and the good-staining VS model with the ground truth (newly added Extended Data Figure 12). The results show that the AQuA workflow can differentiate the quality of the virtual staining models, further showing the value of the workflow. However, this detailed comparison shows that even the good VS model is not as good by comparing Fig. 12(g) and (h). The good VS model still misses lots of cancer cells/lymphocytes (cannot be confirmed with the current magnification). Therefore, additional effort has to be made in the virtual staining algorithm to minimize this digital staining accuracy. This additional study shows the weakness of the current work (even the good ones identified by AQuA are not good enough). The authors should comment more on the difference between Fig. 12(g) and (h), and how to improve the virtual staining performance.

We appreciate the reviewer for pointing this out. To better address this point of the referee, we have newly added **Supplementary Note 2** and **Supplementary Figure 5**, and revised the corresponding parts in the Discussion section, quote:

*"...We want to clarify that although some morphological differences can be observed between good VS images and their HS counterparts, as shown in Extended Data Figs. 12 (g) and (h), these differences are comparable to the natural variations seen within HS workflow and should not impact or alter downstream diagnostic results. For instance, the level of change in nuclei count of VS images compared to their HS counterparts is smaller than the variability observed between two sequential cuts of the same tissue block, which arises from inherent tissue heterogeneity. Additional analysis and supporting information are provided in **Supplementary Note 2** and **Supplementary Fig. 5**..."*

*"**Supplementary Figure 5** Quantification of nuclei count differences between VS and HS images, as well as between serially sectioned adjacent HS tissue cuts stained with HE, MT, and EVG. (a) Reference HE-stained HS image. (b-d) VS images generated by a bad VS model with non-realistic hallucinations, a bad VS model with realistic hallucinations, and a good VS model, respectively. (e) HS image from an adjacent tissue section stained with MT. (f) HS image from an adjacent tissue section stained with EVG. (g) Plot of nuclei counts for images (a-f), demonstrating that the differences between good VS and HS images are comparatively smaller than the differences observed between HS and its adjacent tissue cuts. (h-n) Same as in (a-g) but for a second FOV."*

*"**Supplementary Note 2** Analysis of nuclei count differences between VS and HS images, as well as between serially sectioned adjacent HS tissue cuts."*

In traditional histopathological workflows, even adjacent tissue slides from the same sample of the same patient inevitably have intrinsic variance due to tissue heterogeneity. In diagnosing the potential lung transplant rejection, nuclei aggregation within a local region provides critical diagnostic information for the decision. However, the degree of change in terms of nuclei counts and aggregation observed between VS and the HS reference is even smaller than the variations observed between adjacent tissue sections of the same tissue block. For the two FOVs included in Extended Data Figure 12, a detailed comparison of nuclei count differences is provided in Supplementary Figure 5. This includes comparisons between HS and VS for each FOV, as well as between HS and its adjacent section cuts histochemically stained with different stain types (Masson's Trichrome (MT) stain and Elastic Verhoeff-Van Gieson (EVG) stain) for each FOV. These comparisons clearly demonstrate that the nuclei count discrepancies between HS and VS are smaller than those observed between HS and its adjacent section cuts."

Reviewer #2:

Most comments have been appropriately addressed. The remaining issue is just clarity of certain sections, for example, the following ones.

Human visual inspection in Methods section:

'The optimal VS generator network models for kidney and lung samples, utilized to facilitate the cycle of AF-H&E transformations within the AQuA model, were chosen based on the lowest validation loss values and evaluations conducted through human visual inspection.'

Thank the reviewer for pointing this out. To clarify the process of human visual inspection, we have added the following descriptions in the revised Methods section, quote:

"...More specifically, the VS images from the validation set generated by the models exhibiting the minimal validation loss were manually inspected and compared with their HS counterparts under the supervision of pathologists. The optimal VS model was then selected based on its best perceptual similarity to the HS ground truth in terms of nuclear, cytoplasmic and extracellular details and the accurate representation of relevant diagnostic characteristics..."

'2. Iterations between different domains: Several iterations between the H&E and AF domains are performed to generate image pairs as the input for AQuA-Net. It is of high risk that one model is not properly trained, then the whole pipeline fails.'

Thanks for the reviewer's feedback. We agree that the accuracy of the models used in cycle

inference is important to the overall effectiveness of our pipeline, which is the reason why we included **the original Supplementary Figures 5 and 6 (currently Supplementary Figures 6 and 7), along with the original Supplementary Note 2 (currently Supplementary Note 3) to demonstrate that there is no statistically significant difference between the images produced by the optimal VS model used in the cycle and their corresponding HS images, in the existing Methods section, quote:**

*"...The staining quality of the optimal VS models was assessed and validated against HS reference images in **Supplementary Figs. 6, 7 and Supplementary Note 3**, where multiple staining quality evaluation metrics indicated no statistically significant difference between the VS and the corresponding HS images."*

Furthermore, we should emphasize our AQuA framework also shows robust resilience to potential model imperfections in the cycle inference. A detailed analysis of this was added to the revised Results section, quote:

*"...It is also worth noting that the VS and VAF models used in the cycle inference were not specifically fine-tuned to accommodate the unseen data distribution of the TCGA dataset. Despite this, the AQuA pipeline still achieved a descent accuracy of 87.6%, **demonstrating that even when the forward and backward models used in the cycle inference are not ideal, the AQuA pipeline maintains strong resilience to such imperfections.**"*

Reviewer #3:

Thanks for addressing most of the issues. There are only few points remaining. One aspect concerns the newly added Extended Data Figure 12 and paragraph, that addressed my previous points 2 and 3, regarding the effect of hallucinations on diagnosis.

1. Would it be possible for the authors to add the grad cam visualization of the ROIs for the samples that are illustrated in the Extended Data Fig. 12? The authors stated: "the second poor-staining VS model introduced realistic hallucinations by missing many lymphocyte nuclei". It would be beneficial to show that AQuA is rejecting patches due to the hallucination of lymphocyte nuclei.

2. Could the authors provide the output of TransMil on these patches to show whether TransMil also identifies such hallucinations?

We will address points 1-2 together. Regarding the ROIs included in Extended Data Figure 12, the GradCAM visualization results are provided below (see the figure below), alongside the confidence scores predicted by AQuA and TransMIL. The GradCAM visualizations pinpoint the locations (indicated by arrows) where AQuA accurately identified realistic and unrealistic

hallucinations in VS images from bad models when compared to the HS ground truths. To be specific, the location highlighted in FOV1 indicates aggregation of nuclei due to transplant rejection. The bad VS models cannot reveal nuclei aggregation, while the good VS model generates the diagnostically equivalent aggregation area in comparison to the HS reference. In FOV2, the arrows highlight a local region corresponding to a blood vessel. Both of the bad VS models fail to stain this structure into the correct red color, while the good VS model successfully stains it and reaches an equivalent diagnosis conclusion as the HS reference.

Furthermore, the confidence scores predicted by AQuA for the VS images from two bad models were below 0.5, leading to the successful rejection of these images. Conversely, for VS images from the good model, the confidence scores exceeded 0.8, indicating successful acceptance. In contrast to our performance with AQuA, TransMIL consistently assigned similar confidence scores to VS images from both bad and good models (all above 0.5), failing to reject those VS images with realistic and unrealistic hallucinations from bad VS models. **These results further underscore the reliability and superiority of the AQuA system over TransMIL.**

Figure: Zoom-in regions of VS and HS images, and the Grad-CAM visualization of AQuA in detecting artifacts and hallucinations in bad VS models. Arrows highlight regions of diagnostic importance and VS artifacts and hallucinations affect the diagnosis. The bar plots compare the performance of AQuA against TransMIL in distinguishing VS artifacts and hallucinations.

We have also revised our Extended Data Figure 12 to include the confidence scores predicted by the AQuA model for each VS FOV, quote:

"Extended Data Figure 12 Comparisons of images from two poor-staining and one good-staining VS models, identified by the AQuA workflow, against their HS references in a lung transplant rejection case. (a-d) Large region images from a poor-staining VS model with non-realistic hallucinations; a poor-staining VS model with realistic hallucinations; a good-staining VS model; and the corresponding HS reference. (e-h) Zoomed-in images from the same large regions - from the poor-staining VS models, the good-staining model and the HS reference. The circled region corresponds to cell infiltrate, indicative of acute transplant rejection. The confidence scores \bar{s} predicted by the AQuA model are reported for the VS images of (e-g). (i-l) Zoomed-in images from another diagnostically relevant region, maintaining the same comparison format as in (e-h). The circled region corresponds to cell infiltrate, indicative of acute transplant rejection. The confidence scores predicted by the AQuA model are reported for the VS images of (i-k)."

3. Finally, to comprehensively show AQUAs capabilities, adding a similar case study on hallucinations that can alter the diagnosis in a good VS model and can be detected by AQUA would also be interesting.

In this work, we have not seen instances where hallucinations produced by a high-quality/good, well-trained VS model have altered final diagnoses. This observation actually indicates the success of the virtual staining framework and is consistent with the definition of "realistic hallucination" provided in our paper. **If a VS model generated images with hallucinations that do not appear in the actual tissue specimen and mislead the diagnosis, it would be already detected as a bad VS model by our AQuA pipeline.**

4. Regarding the newly added study on the TCGA data, it would be interesting to compare the results with the output of HistoQC.

We thank the reviewer for this suggestion. Before presenting the HistoQC testing results on the TCGA dataset, we would like to emphasize that:

- Our AQuA model is primarily designed to detect realistic and unrealistic hallucinations in VS images. Using it to identify artifacts and hallucinations in HS slides serves as an additional test bed. Furthermore, **the superior performance of AQuA compared to HistoQC has already been thoroughly demonstrated in Extended Data Figure 10, Supplementary Figure 4, and detailed analysis in the**

existing Discussion section, quote:

"To further demonstrate the effectiveness of AQuA, we also compared it against other HS slide quality control algorithms, including (1) a transformer-based architecture, TransMIL⁴⁷; and (2) an open-source quality control tool named HistoQC^{48,49} designed for digital pathology slides. When comparing AQuA with TransMIL, both of the models shared the same pre-trained ResNet-50 backbone⁵⁰ for feature extraction, and were trained on the same dataset of human lung HS images employing the same majority voting mechanism (C=5). **Extended Data Figure 9** summarizes the comparison of these two models on the same testing set of human lung HS images and reports the corresponding confusion matrices; the results show that TransMIL partially failed to separate the positive and negative populations with a low accuracy of 68.2%, whereas AQuA provides clear separation between the two populations and scored 100% accuracy. Additionally, the total number of trainable parameters of AQuA (~2.1 M) is less than TransMIL (~3.2 M), showcasing AQuA's efficiency. In addition to these, when comparing AQuA with HistoQC and TransMIL at the WSI-level, AQuA demonstrated a more robust concordance with pathologists' judgments in assessing the quality of the slide staining. **Extended Data Figure 10** illustrates the quality control detection results from the AQuA, HistoQC and TransMIL models for a poorly stained HS slide and a well-stained HS slide, as labeled by pathologists. As depicted in **Extended Data Fig. 10(a-f)**, AQuA consistently rejected all the regions from the poorly stained HS slide, while HistoQC and TransMIL only partially rejected some of these poorly stained regions. Although HistoQC and TransMIL successfully identified and rejected severely out-of-focus areas (**Extended Data Fig. 10(g-h)**), they both failed to accurately detect regions lacking hematoxylin staining, exhibiting small blurred areas, containing detached tissue fragments, and obscured lung architecture due to the presence of excessive RBCs, as shown in **Extended Data Fig. 10(i-l)**. As for the well-stained HS slide shown in **Extended Data Fig. 10(m-r)**, all three models, AQuA, HistoQC, and TransMIL, agreed on accepting most of the tissue regions, as desired. However, HistoQC erroneously rejected some regions with high-quality tissue staining (**Extended Data Fig. 10(u-v)**), while AQuA accurately accepted and preserved these stained areas. TransMIL also erroneously rejected several regions with high-quality staining, as depicted in **Extended Data Fig. 10(w-x)**, whereas AQuA accurately accepted these regions. Additional examples from another poorly stained HS slide as well as another well-stained HS slide, labeled by pathologists, are illustrated in **Supplementary Fig. 4**. Here, AQuA was able to accurately reject poor-staining regions aligning with the pathologists' assessments as shown in **Supplementary Fig. 4(a-f)**. However, HistoQC incorrectly accepted regions that contained out-of-focus areas, staining artifacts and exogenous pigments, as shown in **Supplementary Fig. 4(i-j)**, and TransMIL inaccurately accepted regions with air bubbles in **Supplementary Fig. 4(k-l)**. Furthermore, both HistoQC and TransMIL erroneously rejected

several regions that exhibited normal tissue structures and high staining quality, as depicted in Supplementary Fig. 4(m-r), which AQuA correctly identified and accepted. A similar performance advantage of AQuA over HistoQC was also observed during testing on the TCGA dataset."

In addition to these former results and extensive analyses described above, we further evaluated the performance of HistoQC on 470 bad-quality (positive) HS lung images and 519 good-quality (negative) HS lung images from the TCGA dataset and compared it with the performance of AQuA. Unlike AQuA, which directly outputs a confidence score for each test image patch, HistoQC only provides per-pixel results indicating whether each pixel passes the quality check or not, and thresholding based on this map to a patch- or slide-level assessment has not been thoroughly studied. To mimic the concept of a confidence score in AQuA, we used the percentage of quality accepted areas for each image (acceptance area percentage) as the classification metric. The image patches with over 50% of the total area passing the quality check are classified as negative (good-quality), and those below this threshold are identified as positive (bad-quality).

Figure: The comparison of AQuA and HistoQC's performance on the TCGA dataset. (a) Boxplot of confidence scores given by AQuA. (b) Confusion matrix of AQuA's testing results on the TCGA dataset. (c) Boxplot of acceptance area percentage given by HistoQC. (d) Confusion matrix of HistoQC's testing results on the TCGA dataset.

These new comparison results are shown in the figure above, where AQuA achieved an accuracy of 87.6%, a sensitivity of 79.1%, and a specificity of 95.2%, outperforming HistoQC, which achieved an accuracy of 84.6%, a sensitivity of 76.6%, and a specificity of 91.9%. **These findings further highlight the superiority of AQuA over HistoQC.**

We also revised the descriptions in the Discussion to include this conclusion, quote:

"...A similar performance advantage of AQuA over HistoQC was also observed during testing on the TCGA dataset."

5. Reporting the accuracy without transfer learning would also be helpful since are samples were HE.

The **new Extended Data Figure 13** was added to compare AQuA's performance on the TCGA dataset before and after the transfer learning step. The related analysis was also added in the subsection of "TCGA dataset" in the Methods section, quote:

"...Since the WSIs in the TCGA database originated from 49 distinct tissue sources, they have significant H&E color style variations which the initial AQuA model has never seen. Consequently, transfer learning is essential to maintain the high performance of the AQuA workflow. To highlight the importance of transfer learning, the testing results of the AQuA model with and without transfer learning are provided in Extended Data Fig. 13..."

"Extended Data Figure 13 Autonomous quality assessment of AQuA model before and after transfer learning on a subset of good HS images from the TCGA dataset. (a) Boxplot of confidence scores given by AQuA after transfer learning. (b) Confusion matrix of AQuA's testing result on TCGA dataset after transfer learning. (c) Same as (a), but using the AQuA model before transfer learning. (d) Same as (b), but using the AQuA model before transfer learning."

6. Would suggest to either support the following statement with some experiments or leave it out: "Additionally, the AQuA framework can be extended to some of the recent advances in stain-to-stain transformations and stain normalization methods^{25,59-61}. Through iterations between image domains of different stains, or between image domains of AF and multiple types of stains, AQuA can potentially provide autonomous quality and hallucination assessment for virtually transformed and normalized pathology images, which is especially important for cases where HS references are inaccessible."

This section was added upon the request of Reviewer 2 during the previous round of the revision. **It was placed in the Discussion section rather than the Results section to highlight the prospective future applications of the AQuA framework.** To enhance clarity

and prevent any potential misunderstandings, we have also revised our descriptions within the Discussion section, quote:

"..Additionally, although we did not provide results on this capability in this manuscript, the AQuA framework can potentially be extended to some of the recent advances in stain-to-stain transformations and stain normalization methods^{25,59-61}. Through iterations between image domains of different stains, or between image domains of AF and multiple types of stains, AQuA might provide autonomous quality and hallucination assessment for virtually transformed and normalized pathology images, which is especially important for cases where HS references are inaccessible."